# Engineering selective competitors for the discrimination of highly conserved protein-protein interaction modules

Charlotte Rimbault [1,2], Kashyap Maruthi[3,4], Christelle Breillat[1,2], Camille Genuer[1,2], Sara Crespillo[1,2], Virginia Puente-Muñoz[1,2], Ingrid Chamma[1,2], Isabel Gauthereau [1,2], Ségolène Antoine[1,2], Coraline Thibaut[1,2], Fabienne Wong Jun Tai[5], Benjamin Dartigues[5], Dolors Grillo-Bosch [1,2], Stéphane Claverol[6], Christel Poujol[7], Daniel Choquet [1,2,7], Cameron D. Mackereth [3,4]* & Matthieu Sainlos [1,2]*

Designing highly specific modulators of protein-protein interactions (PPIs) is especially challenging in the context of multiple paralogs and conserved interaction surfaces. In this case, direct generation of selective and competitive inhibitors is hindered by high similarity within the evolutionary-related protein interfaces. We report here a strategy that uses a semi-rational approach to separate the modulator design into two functional parts. We first achieve specificity toward a region outside of the interface by using phage display selection coupled with molecular and cellular validation. Highly selective competition is then generated by appending the more degenerate interaction peptide to contact the target interface. We apply this approach to specifically bind a single PDZ domain within the postsynaptic protein PSD-95 over highly similar PDZ domains in PSD-93, SAP-97 and SAP-102. Our work provides a paralog-selective and domain specific inhibitor of PSD-95, and describes a method to efficiently target other conserved PPI modules.

[1] Interdisciplinary Institute for Neuroscience, UMR 5297, Centre National de la Recherche Scientifique, F-33076 Bordeaux, France. [2] Interdisciplinary Institute for Neuroscience, University of Bordeaux, F-33076 Bordeaux, France. [3] Institut Européen de Chimie et Biologie, Univ. Bordeaux, 2 rue Robert Escarpit, F-33607 Pessac, France. [4] Inserm U1212, CNRS UMR 5320, ARNA Laboratory, Univ. Bordeaux, 146 rue Léo Saignat, F-33076 Bordeaux, France. [5] University of Bordeaux, CBiB-LaBRI, F-33000 Bordeaux, France. [6] Proteome Platform, Functional Genomic Center of Bordeaux, University of Bordeaux, F-33076 Bordeaux, France. [7] Bordeaux Imaging Center, UMS 3420 Centre National de la Recherche Scientifique, University of Bordeaux, US 4 INSERM, F-33076 Bordeaux, France. *email: cameron.mackereth@u-bordeaux.fr; sainlos@u-bordeaux.fr

Protein–protein interactions (PPIs) are involved in the complex and intricate cellular networks that dynamically govern processes such as transport, localization and signal transduction. Preventing specific interactions can provide insight into physiological role of each protein partner or reduce the deleterious effects of abnormal protein function. It is in the latter context that PPI inhibitors have seen increasing interest as potential drug targets[1,2]. Despite their promise, the study and targeting of PPI modules still represent a challenge, due in part by stronger evolutionary conservation of residues at the PPI interface compared to the rest of the protein domain[3–7]. Processes such as domain recombination[8] and gene duplication have led to paralogs, as well as distantly related proteins, that can share highly conserved interfaces with similar specificity. This tendency is exemplified by protein domains that bind short peptide sequences, such as the PDZ, SH3, SH2 and WW domains. Large-scale interactome studies on PDZ[9] and SH3[10] domains highlight shared binding preferences for protein family clusters[11,12]. The development of a selective inhibitor for a specific PPI must, therefore, avoid interaction with similar coexisting PPI interfaces, or risk adverse off-target effects.

The postsynaptic density protein 95 (PSD-95; also known as SAP90 or *DLG4*) is highly studied and one of the main post-synaptic scaffold proteins. PSD-95 plays an important role in the organization of the postsynaptic density[13,14], with the anchoring of key synaptic proteins such as ionotropic glutamate receptors (NMDA receptors and AMPA receptor complexes) and adhesion proteins. PSD-95 dysfunctions have therefore been associated with various central nervous system disorders[15,16] and constitutes a therapeutic target for stroke treatment[17]. Complicating its study and treatment is the fact that PSD-95 belongs to a family of four similar proteins essential for synaptic function that evolved from a common ancestor[18,19]. These proteins belong to the disk-large (DLG) subfamily of membrane-associated guanylate kinases (MAGUKs) and also include SAP97 (*DLG1*), SAP102 (*DLG3*) and PSD-93 (*DLG2*). All four members of this subfamily share the same structural domain organization of sequence-conserved PDZ (PSD-95, discs large, zona occludens 1) domain PPI modules[13,14]. Numerous studies indicate that each protein exerts different specific functions in synaptic plasticity and cognitive behaviours despite their strong homology[13,19–24]. However, the discrete role of each family member and the molecular mechanisms underlying their differences still remain unresolved.

This inability to clearly distinguish the precise role of each PSD-95-like DLG protein also limits targeting by potential therapeutics. Past methods to investigate the family members have typically relied on genetic approaches, which suffer from a low temporal resolution and hence allow compensation phenomena. In addition, no molecular tools have yet been reported that exhibit selective and acute modulating effect towards a single family member. We and others have previously developed and used synthetic competitive peptide-based ligands that target the repeated PDZ domains of DLG proteins[25–27]. However, while showing strong selectivity for the four DLG proteins, the high sequence conservation and binding specificity shared by the PDZ domains[9] prevents specific interaction with a single family member. New methods are required to study PSD-95 and member-specific functions with acute and highly selective modulation of endogenous PPIs in their native environment.

Here we describe a two-part strategy to create PPI modulators with high selectivity for a single member of a conserved family, in this case a protein module from PSD-95 that contains the first two PDZ domains. Our strategy consists in first isolating specific binders by phage display, which in a second step are combined with more degenerate peptides that block the PPI domain surface. The results are PPI inhibitors with designed selectivity at the level of a single family member, that enable the study of isolated PPIs amongst highly sequence-conserved and promiscuous interaction modules.

## Results

**Target analysis and selection strategy**. Numerous studies indicate that PSD-95 is composed of two supramodules: one composed of the first two PDZ domains, and the second containing the third PDZ domain, a SH3 domain and a non-functional guanylate kinase domain (Supplementary Fig. 1a). In designing our selection strategy, we focused on the entire first PDZ–PDZ supramodule instead of isolated PDZ domains to help avoid generating binders that recognize epitopes blocked in the full-length protein. Surface analysis shows that the binding groove and surrounding area of the first two PDZ domains of PSD-95 are highly conserved within the DLG paralog family (PSD-95, SAP97, SAP102 and PSD-93, Fig. 1 and Supplementary Fig. 1b). All PDZ domain inhibitors that have been reported to date operate exclusively by blocking the binding groove[28] but it is unlikely that rational or selection-based approaches solely against the binding groove would generate specific PSD-95 binders. In contrast, less-conserved patches can be found on the opposite sides of the binding grooves (Fig. 1b, c). We reasoned that by first targeting these patches, it will be possible to generate the higher selectivity required for family member-specific binding, which could in turn be exploited to engineer the final selective competitors by fusion of an element that directly interacts with the PDZ domain binding groove (for specificity vs selectivity precisions, see Supplementary Note 1).

We used a phage display strategy with a library of diversified [10]FN3 domains as a robust scaffold to yield convex surface binders[29,30] (Fig. 2a). The [10]FN3 domain has been previously used to obtain high-affinity binders of numerous protein domains with good stability, ease of production and the absence of disulphide bridges as found in $V_H H$ or variable regions of antibodies[30]. We used the sequence improved by the group of Koide[29,31]. Starting with a commercially available phagemid vector, we introduced a lac repressor (*LacIq* gene) to reduce the phagemid toxicity, swapped the PelB peptide signal sequence to a DsBA motif to rely on the SRP pathway[32] and inserted the [10]FN3 scaffold as a fusion to the g3p minor phage coat protein (Supplementary Fig. 2a). We next performed diversification of the [10]FN3 BC and FG loops using NNK degenerate codons by both varying all residues as well as the length of the two loops by the pFunkel method[33] (Supplementary Fig. 2b). This provided us with a library of about $10^{10}$ unique clones as estimated by the sequence analysis of 96 randomly picked colonies (Supplementary Fig. 2c). In parallel, we produced the biotinylated tandem PDZ domains of PSD-95, as well as the tandems of the other DLG family members by introducing a biotin acceptor peptide tag on their N-terminus and co-expressing the resulting modified gene with a plasmid encoding for the biotin ligase BirA in *Escherichia coli*. The targets were purified to homogeneity and biotinylation levels were above 75% as judged by gel assay.

Three rounds of selection were performed on streptavidin-coated magnetic beads functionalized with the biotinylated PDZ domains, using our phage library produced with the M13KO7 phage helper to favour the display of a single diversified domain per phage (Supplementary Fig. 2d). The target concentration was decreased between each round to increase the stringency (from 100, 50 to 25 nM). After the third round, 96 isolated colonies were randomly picked and analysed by phage-ELISA and sequencing. We identified 11 different clones that showed a strong ELISA response for PSD-95 with various levels of enrichment (Fig. 2b). No consensus sequence emerged from the sequence analysis,

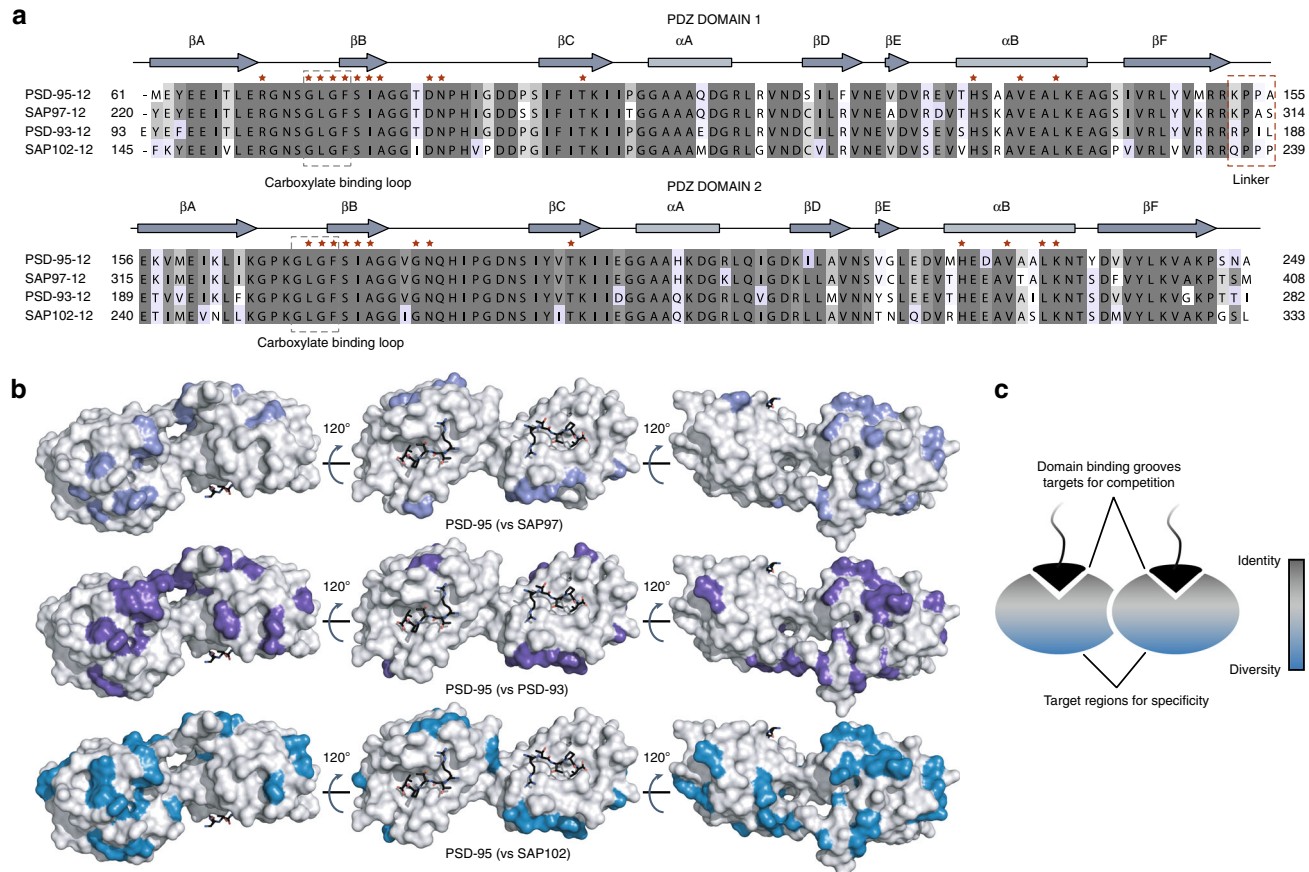

**Fig. 1** Analysis of PSD-95 tandem PDZ domains as a target for specific binders. **a** Sequence alignment of the first two PDZ domains of the PSD-95 paralog family (*Rattus norvegicus*). The red asterisks indicate residues directly involved in the binding of partner proteins. **b** Surface representations of PSD-95 tandem PDZ domains (PDB ID 3GSL, domain 1 on the left and domain 2 on the right) with ligand modelled in (RTTPV aligned from PDB ID 3JXT, black sticks) and with non-identical residues coloured in shades of blue according to the other tandem it is compared to (SAP97, PSD-93 or SAP102). **c** Scheme summarizing the location of highly conserved and more diverging regions on the surface of PSD-95 tandem PDZ domains with respect to other family members

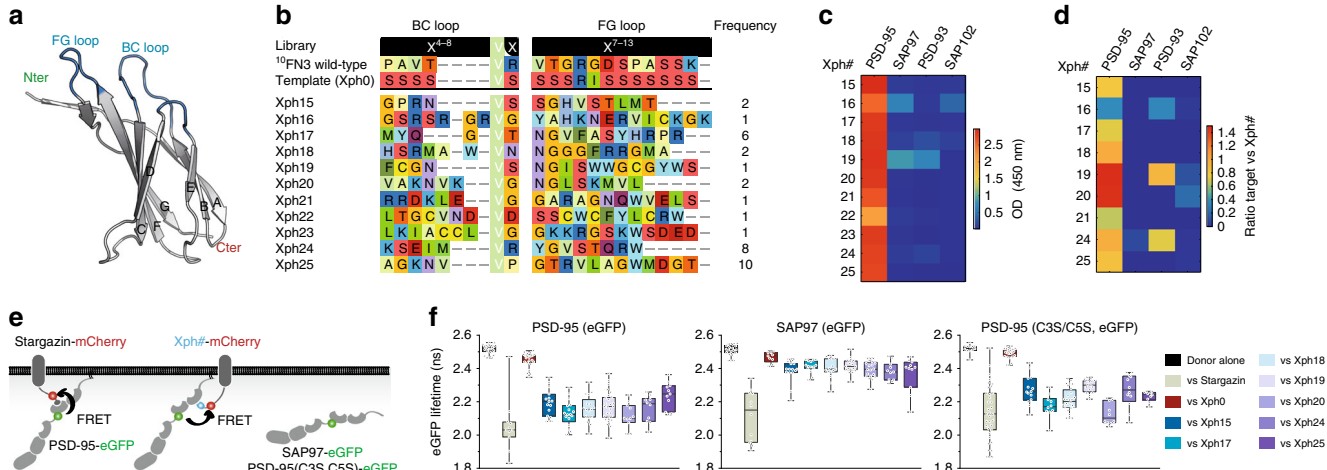

**Fig. 2** Isolation and selectivity of clones targeting PSD-95 tandem PDZ domains. **a** Structure of the protein scaffold used for selection (PDB ID 3K2M). The diversified loops (BC and FG) are represented in blue. **b** Library design and loop sequences of the isolated clones (Xph#) against PSD-95 first two PDZ domains. **c** Phage-ELISA of isolated clones against tandem PDZ domains from PSD-95 family. OD, optical density at 450 nm. **d** Pull-down assay of purified selected clones against tandem PDZ domains from PSD-95 family. Unprocessed original gels are shown in Supplementary Fig. 4. **e** Schematic of the FRET systems used for measurement of the donor lifetime (FLIM). **f** Lifetime of eGFP inserted in PSD-95, SAP97 and PSD-95 (C3S, C5S) in presence of the indicated acceptor-containing protein constructs. Box plots show median, first and third quartile, with whiskers extending to the minimum and maximum and all individual data points (each corresponding to a single cell) pooled from at least two independent experiments. Source data are provided as Source Data file

which can be explained by the much higher theoretical diversity of the designed library as compared to the actual diversity that can be handled with the phage display method.

**Clones characterization**. Specificity of the isolated clones was first evaluated by phage-ELISA response towards the tandem PDZ domains of PSD-95, as compared to the other DLG family members (Fig. 2c). Most clones showed specificity for PSD-95, which was remarkable given the absence of negative selection. Performing the same analysis with the isolated second PDZ domain of PSD-95 failed to show a strong response, suggesting that the isolated clones were either binding to the first PDZ domain or had epitopes on both domains (Supplementary Fig. 3). We excluded clones that contained multiple cysteines from further study, and confirmed the binding properties of the remaining clones with a pull-down approach on recombinant proteins (Fig. 2d and Supplementary Fig. 4). The evolved $^{10}$FN3 domain clones were produced in *E. coli* with a deca-His-tag, directly isolated from the lysates with Ni-NTA magnetic beads, and then incubated with purified tandem PDZ domains. The material left on the beads following the wash was eluted with imidazole and analysed by densitometric analysis of the colloidal blue-stained sodium dodecyl sulphate-polyacrylamide gel electrophoresis (SDS-PAGE) band intensity. The results were similar to measurements by phage-ELISA, indicating that recognition of PSD-95 tandem PDZ domains is indeed mediated by the evolved $^{10}$FN3 domains. To ensure that the binding capacities of the clones were preserved in a cellular environment, the seven best binders were further evaluated by a cell-based FRET/FLIM (Förster resonance energy transfer/fluorescence-lifetime imaging microscopy) assay. The FRET system was based on one previously developed to investigate divalent ligands[25] (Fig. 2e). The donor fluorescent protein, EGFP, was inserted after the second PDZ domain in PSD-95 or SAP97. The acceptor, mCherry, replaces the C-terminal PDZ domain-binding motif (PBM) of the transmembrane protein Stargazin, and is followed by a 20-amino-acid linker and the $^{10}$FN3 clone. All clones showed strong binding to full-length membrane-bound PSD-95 as indicated by reduction of the mean lifetime of the donor fluorescent protein to around 2.2 ns as compared to the lifetimes above 2.4 ns obtained with the donor alone or in presence of a naïve clone (Xph0; Fig. 2f). In contrast, only weak binding could be observed with SAP97 with mean lifetimes around 2.4 ns for all the clones we tested (Fig. 2f and similar results were obtained for PSD-93, Supplementary Fig. 5). Strong binding was also observed with a soluble mutant of PSD-95 (ref. [34]) that can be more directly compared to the cytosolic SAP97. Together these results indicate that the evolved $^{10}$FN3 domains we have selected are robust and specific binders of epitopes on the PSD-95 tandem PDZ domains.

**Epitope mapping**. Following the specificity evaluation, five final clones (Xph15, Xph17, Xph18, Xph20 and Xph25) stood out based on their relative binding strength and specificity. We selected three representative clones (Xph15, Xph18 and Xph20) to further investigate binding properties with a series of in vitro assays. To maximize the solubility and stability, we used two strategies: the first consisted of a fusion to the SUMO protein tag on the C-terminus of the clone (an N-terminal tag resulted in loss of binding), the second approach involved mutation of serine 65 into a lysine as previously reported by the group of Koide[35]. Both strategies improved our capacity to concentrate and freeze-store the proteins while maintaining homogeneity of the samples, as judged by analytical size exclusion chromatography (Supplementary Fig. 6). Thermal stability evaluation of the three S63K mutant clones (Supplementary Fig. 7) showed inflection in their

thermal unfolding curves around 69 °C for Xph18 and Xph20, and at 77 °C for Xph15, suggesting only partial loss of stability to that of wild-type $^{10}$FN3 (82.5–88 °C)[36,37]. Finally, nuclear magnetic resonance (NMR) spectroscopy was used to test the binding specificity of the recombinant clones towards $^{15}$N-labelled tandem PDZ domains of all four DLG family members (Supplementary Fig. 8). Binding was detected by the change in $^{15}$N-HSQC peak positions, and in keeping with the previous results, all three clones resulted only in significant changes to the PSD-95-12 spectra, with the exception of Xph20 and SAP97 for which partial binding could be observed in these conditions (>80 µM). Overall, these results indicate that the several recombinant forms of the selected clones are specific for PSD-95-12 and constitute solid candidates for further investigation and engineering.

To allow for precise engineering of subsequent tools based on Xph15, Xph18 and Xph20, detailed information on the mode of interaction and the precise epitopes of each clone are required. NMR spectroscopy was chosen to access residue-specific information, and therefore we first obtained chemical shift assignment of the PSD-95 tandem PDZ domain (Supplementary Fig. 9). A comparison of spectra for PSD-95-12 in the unbound form and bound to each clone reveals residue-specific changes in NMR crosspeak positions or signal disappearance due to broadening (Fig. 3; Supplementary Figs. 10–12). Quantification of the clone-dependent changes in the crosspeak positions shows that Xph15 and Xph20 interact with similar residues on PSD-95 situated on the opposite side of the binding groove of PDZ domain 1. In contrast, the Xph18 epitope involves both PDZ domains 1 and 2, as well as the connecting linker, but similarly encompasses a region distant from the binding grooves.

The epitope mapping experiments from Xph15 and Xph20 reveal a binding epitope nearly exclusive to the first PDZ domain. To probe this interaction in more detail, we repeated the NMR binding experiments with isolated PDZ1 and PDZ2 from PSD-95 (annotated reference spectra in Supplementary Fig. 13). Addition of Xph15 and Xph20 to $^{15}$N-labelled PDZ1 resulted in nearly identical chemical shift perturbation as for the tandem construct (Supplementary Figs. 14 and 15). Also similar to the previous binding studies, there were no significant changes, and thus no apparent interaction, for the isolated PDZ2 domain. Using similar protein concentrations for Xph18, neither isolated PDZ domain from PSD-95 displayed evidence of interaction (Supplementary Fig. 16). At higher protein concentrations, a specific interaction between Xph18 and PDZ1 could, however, be detected. Thus, for Xph18, it appears that a primary interaction still involves PDZ1, although the epitope extends across both PDZ1 and PDZ2 and the presence of both domains is a requisite for binding.

Based on the specific complexes formed between each of the clones and PSD-95, we decided to generate putative atomic models in order to better define key structural elements of the epitopes. To this end, we used the NMR-based interaction data to generate docking models for all three clones supplemented with additional information. For Xph15 and Xph20, we first collected reciprocal binding data from isotopically labelled clones in the unbound and PDZ1-bound forms (Supplementary Figs. 17 and 18). During the analysis, we also noted that there were two populations in the NMR spectra, but only for the unbound state. Assignment of backbone chemical shifts for the two equal populations of free Xph15, as well as the major and minor populations of free Xph20, localized the differences to the BC and FG loops, and the C-terminal β-strand (Supplementary Figs. 19 and 20). Further investigation revealed that these two populations were unaffected by temperature (283–308 K) and remained in equilibrium during the titration with Xph15 or Xph20, with a population exchange rate >1 s$^{-1}$ (Supplementary Fig. 21).

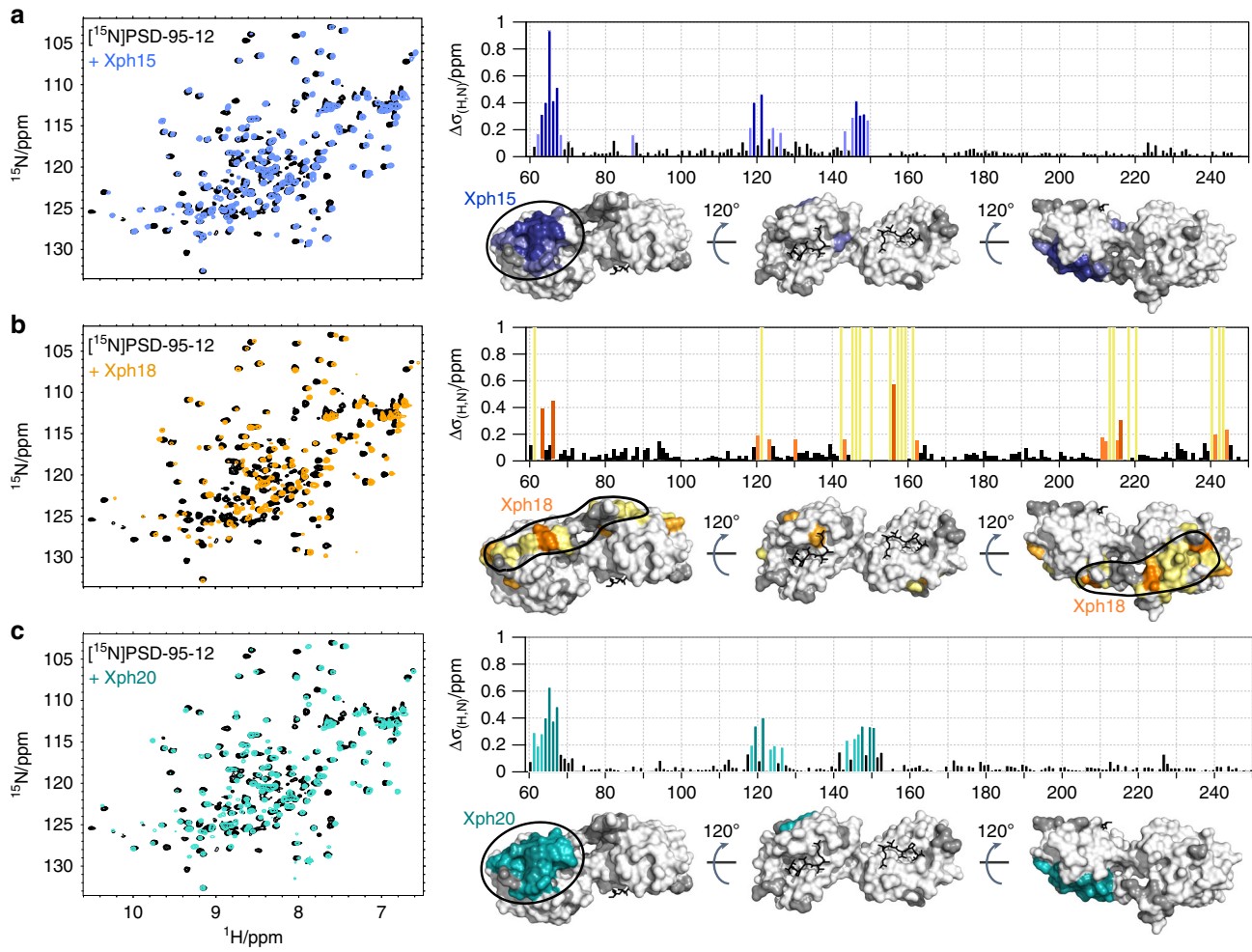

**Fig. 3** Epitope mapping. **a–c** $^{15}$N-HSQC NMR spectrum of unbound 80 μM [$^{15}$N]PSD-95-12 (black) with spectra following addition of 100 μM natural abundance Xph15 (blue) (**a**), Xph18 (orange) (**b**) or Xph20 (teal) (**c**). Histograms show the combined $^{1}$H$^{N}$, $^{15}$N chemical shift perturbation $\Delta\delta_{(H,N)}$ of the backbone amide crosspeaks for residues in PSD-95-12 that result from addition of the clones. Residues with $\Delta\delta_{(H,N)}$ values greater than 0.3 ppm (dark coloured bars) or 0.15 ppm (light coloured bars) are also highlighted on surface representations of PSD-95-12 (same orientations as in Fig. 1). Dark grey shading indicates residues with ambiguous or missing NMR data. Yellow bars in **b** indicate crosspeaks that are broadened upon addition of Xph18. In all three cases, addition of 1.2 molar equivalents of Xph15, Xph18 or Xph20 was sufficient to fully shift the [$^{15}$N]PSD-95-12 crosspeaks to the bound population, consistent with 1:1 stoichiometry for the complexes. Source data are provided as Source Data file

Using the identified binding surfaces on Xph15 and Xph20, as well as the corresponding interaction regions on PDZ1, we generated a series of docking models (Fig. 4a and Supplementary Fig. 22). For both clones, the model reveals that the bound Xph15 or Xph20 likely extends away from the ligand-binding groove, and is not expected to sterically hinder interaction to the C-termini of PSD-95 interaction partners. In addition, the bound clones are remote from the interdomain contacts to PDZ2, and we can reasonably speculate that the interaction has limited impact on the orientation of the two domains. This is further supported by the strong similarity of chemical shifts when comparing Xph15 and Xph20 binding to PDZ domain 1 alone and to the tandem. There should also be minimal impact on the dynamic interconversion of two major domain arrangements of the PDZ1 and PDZ2 domains that has been recently described[38].

For Xph18, we first created a docking model of Xph18 with PDZ1 (Supplementary Fig. 22c). Since we knew that PDZ2 also contributes to the interaction with Xph18, we collected residual dipolar coupling data on the Xph18:PSD-95-12 complex in order to fix the orientation of the PDZ2 domain relative to PDZ1, and to confirm that the two domains are relatively fixed in position relative to each other (Supplementary Fig. 22d, e). A final model

of the complex illustrates that this PDZ domain arrangement explains the previously observed extended interaction surface with Xph18 (Fig. 3b). Despite the fixed arrangement, the binding grooves on both PDZ1 and PDZ2 are still accessible to ligand binding.

Overall, these results demonstrate a primary requirement of PDZ1 and a specificity that is achieved by binding to regions remote from the binding groove. For all three clones, the origin of the specificity could be attributed principally to a single phenylalanine residue (F119) that is replaced by an arginine in all three other proteins from the DLG family (Supplementary Fig. 22g).

**Affinity determination**. Binding affinities of Xph15, Xph18 and Xph20 were evaluated by single-cycle kinetics surface plasmon resonance (SPR) and by isothermal titration calorimetry (ITC) (Fig. 5 and Supplementary Figs. 23 and 24). With both techniques, the affinity of the three clones were similarly ranked with Xph15 being the weakest binder and Xph20 the strongest. Xph15 presented a dissociation constant in the micromolar range with fast kinetics ($k_{on}$ $9.5 \times 10^4$ M$^{-1}$ s$^{-1}$ and $k_{off}$ 0.4 s$^{-1}$). In comparison, Xph20 presented a faster $k_{on}$ and a slower $k_{off}$,

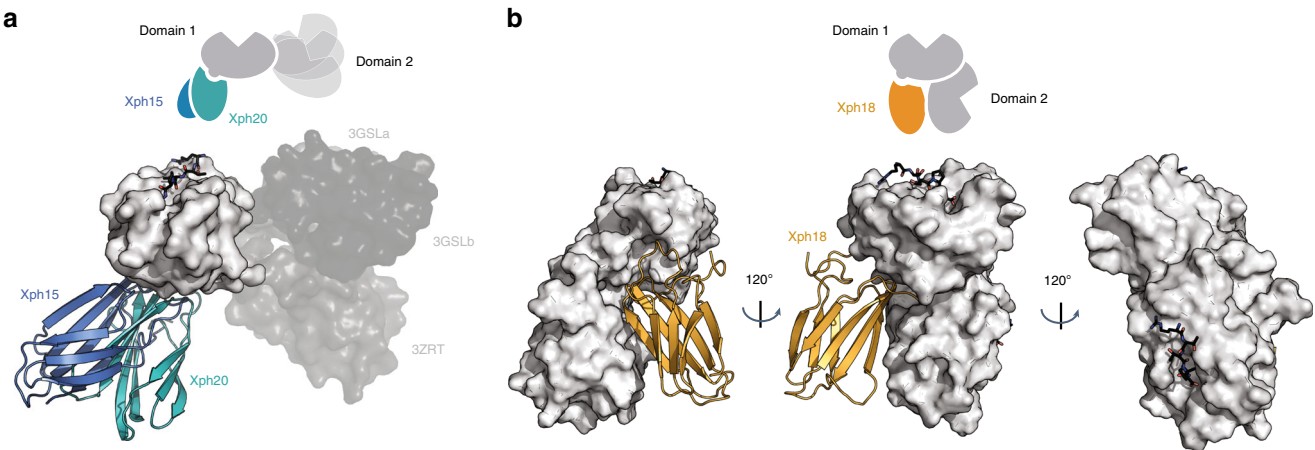

**Fig. 4** Docking models. **a** Superimposed lowest energy models of Xph15 (blue) and Xph20 (teal) bound to the surface of PSD-95-12 obtained by using Haddock2.2 with respective scores of −90.2 ± 2.7 and −90.5 ± 3.9 (mean ± s.d.). The models were calculated using only PDZ domain 1 and the resulting complexes are superimposed with three different PSD-95 tandem conformers obtained by X-ray crystallography (PDB 3GSL and 3ZRT). **b** Lowest energy model of Xph18 (orange) bound to the surface of PSD-95-12. A first model of PDZ domain 1 with Xph18 using Haddock2.2 results in a score of −115.9 ± 3.8, which was followed by the alignment of PDZ domain 2 by additional HADDOCK and residual dipolar coupling restraints resulting in a final score of −147.4 ± 1.3

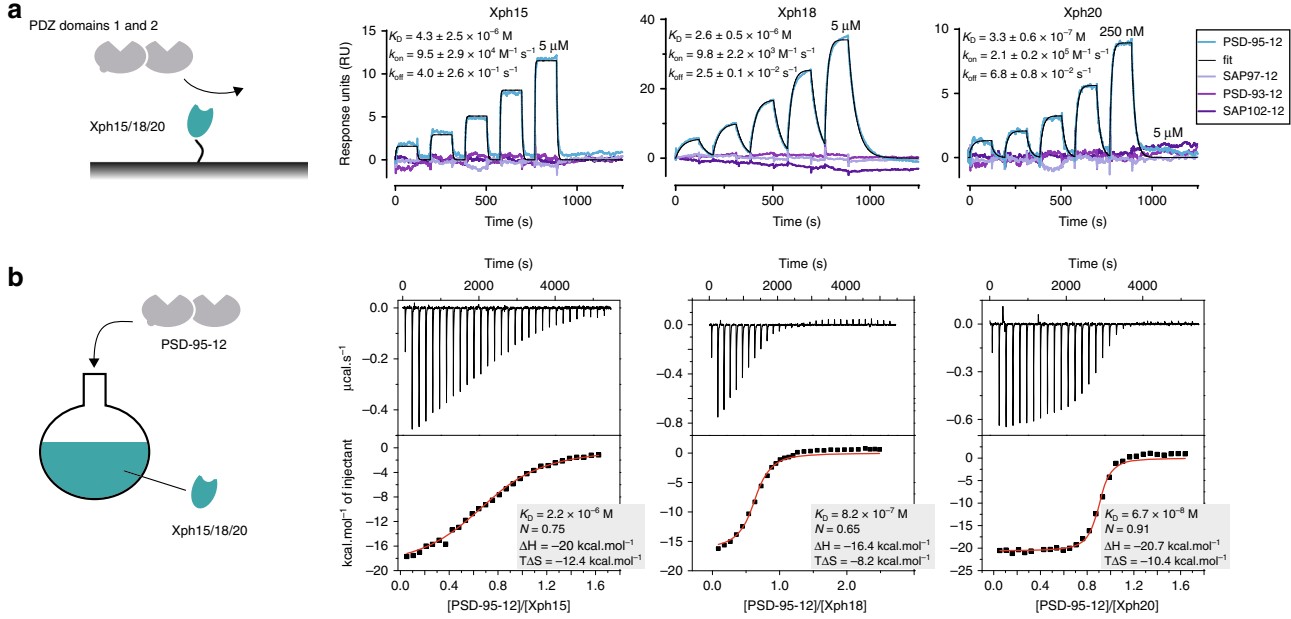

**Fig. 5** Binding affinities of the selected clones. **a** SPR sensorgrams obtained by single-cycle kinetics of tandem PDZ domains (analyte) against immobilized biotinylated Xph15/18/20 (ligand). The reported concentrations represent the highest concentration used on the final analyte injection, the previous four injections are a series of twofold dilutions. The immobilized ligand density for Xph15, Xph18 and Xph20 was 37, 74 and 24 RU, respectively. For each experiment, the theoretical $R_{max}$ (maximum analyte-binding capacity of the surface in RU) was compared to the experimental $R_{max}$ (theory/observed: 23/20 RU, 47/52 RU and 15/17 RU for Xph15, Xph18 and Xph20, respectively). The coloured curves represent measured data points and black lines represent the global fit obtained with a 1:1 binding model used for analysis. Kinetics values are the average and s.d. of three independent titrations. For rate plane with isoaffinity diagonals representation, see Supplementary Fig. 24. **b** Isothermal titration calorimetry thermographs and curve fits for titrations of Xph15, 18 and 20 into PSD-95 domains 1 and 2. $N$ represents the fitted stoichiometry. Source data are provided as Source Data file

leading to submicromolar dissociation constants of 330 nM and 67 nM by SPR and ITC, respectively. Finally, Xph18 showed a different kinetic profile with slower $k_{on}$ and $k_{off}$ yielding a micromolar to submicromolar dissociation constant. The slower association rate constant observed for this clone can be justified by the fact that its epitope is spread over both domains 1 and 2, which are to some extent mobile with respect to one another[38] and the two domains must adopt the appropriate orientation so that the epitope can be fully recognized. Compared to antibodies,

our binder affinities were at the lower limit ($K_D$ ranging from 10 μM to 100 nM) and display relatively fast dissociation rate constants (half-lives <1 min). Nevertheless, they are in the range of natural binders such as PSD-95 partners with PBMs that bind in the low micromolar range with fast rate constants. Of note, further investigation of the sole unspecific binding observed between Xph20 and SAP97, that was partially observed by NMR titration, resulted in a $K_D$ estimation over 150 μM (Supplementary Fig. 23).

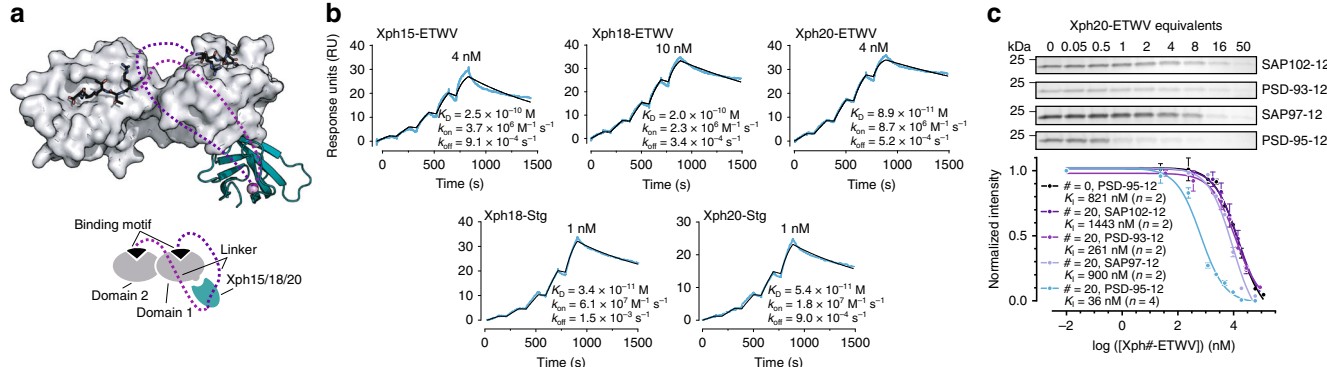

**Fig. 6** Engineering PSD-95 selective ligands. **a** Design of selective competitors against PSD-95 tandem PDZ domains. The purple ball corresponds to the Xph clone C-terminus (T94) that is fused to the linker and binding motif. The binding motif can in principle bind to each domain (overlay of the two situations). **b** Representative SPR sensorgrams obtained by single-cycle kinetics of tandem PDZ domains (analyte) against immobilized biotinylated Xph15/18/20-binding motif (ligand). The reported concentrations represent the highest concentration used on the final analyte injection, the previous four injections are a series of twofold dilutions. The coloured curves represent measured data points and black lines represent the global fit obtained with a 1:1 binding model used for analysis. For rate plane with isoaffinity diagonals representation, see Supplementary Fig. 24. **c** Competitive titrations of Xph20-ETWV against a complex of tandem PDZ domains and a divalent ligand. Top: colloidal blue-stained SDS-PAGE analysis of the competition against indicated PDZ domains (non-competed material). Bottom: quantification by densitometry of competition. Each data point represents the mean of two independent experiments and s.e.m. For uncropped gels, see Supplementary Fig. 25. Source data are provided as Source Data file

These results were further confirmed by using a different experimental configuration for the SPR (i.e., immobilization of the tandem PDZ domain instead of the clones); however, the biphasic behaviour of the clones prevented a quantitative analysis (Supplementary Fig. 23). Interestingly, single point mutations on PSD-95 and SAP97 tandems (F119R and R278F, respectively, in order to swap what appears as a key residue for Xph15, Xph18 and Xph20 specificity) significantly impaired binding of Xph20 to the PSD-95 mutant by SPR and NMR, while it allowed recognition and binding to the SAP97 mutant (Supplementary Fig. 25). This highlights the critical role of F119 in the generation of specific binders and further validates our structural studies.

**Selective competitor engineering and characterization.** In order to convert our specific binders (Xph15, Xph18 and Xph20) to selective inhibitors, it is necessary to add a peptide motif that interacts with the PDZ domain binding groove despite the fact that available PBMs are relatively non-selective for specific PDZ domains from PSD-95, SAP97, SAP102 or PSD-93 (Fig. 6a). One of the chosen peptides was derived from the AMPAR auxiliary subunit stargazin with higher affinity for the second PDZ domain[39], and the second peptide is a consensus sequence isolated from a peptide phage display investigation that binds both domains[9]. To preserve the terminal carboxylate group, which is key to PDZ domain binding, the inhibiting peptides must be inserted C-terminal to the Xph clones with sufficiently long connecting linkers.

The engineered constructs were first evaluated by SPR (Fig. 6b, Supplementary Fig. 26 and Supplementary Table 1). Fusion of the two elements dramatically changed the kinetic profiles as compared to each isolated element, most notably by slowing the dissociation rate constant. Dissociation constants were consequently also modified, with affinities in the subnanomolar range. Some level of interaction is observed for other tandem domains as expected from the non-specific binding capability of the peptide (Supplementary Fig. 26). However, similar levels of response in the sensorgrams were only obtained with over 100-fold higher analyte concentrations and resulting dissociation constants more than a 1000-fold weaker than for PSD-95 (Supplementary Table 1). Overall, all constructs that we generated

showed dramatically improved affinities while maintaining a strong selectivity.

The competing capacity of one of the most potent clones was next evaluated by a pull-down assay (Fig. 6c, Supplementary Fig. 27 and Supplementary Table 2). PDZ domain tandems for all four family members were first bound to streptavidin-coated magnetic beads functionalized via a biotinylated divalent peptide-based ligand[25] and the resulting PDZ domain complexes were then incubated in presence of increasing amounts of the competitors. The biomimetic divalent ligand was used here as a model for complex multivalent interaction systems such as those that can be found in cells. The affinities of the divalent ligand were first determined by fluorescence polarization for each tandem PDZ domains (Supplementary Fig. 28). The similarity of the $K_I$s that were measured illustrates the sequence identity of the binding grooves and its consequences even on complex ligands. Quantification of the competition was performed by loading on SDS-PAGE the material left on the beads and densitometric analysis of the colloidal blue-stained band intensity. The titration results were consistent with the SPR experiments, with lower concentrations of Xph20-ETWV required to compete off the divalent ligand from PSD-95 PDZ domains as compared to SAP97, SAP102 or PSD-93. Replacing the Xph20 half of the molecule by a naïve [10]FN3 scaffold (Xph0) abolished the increased competition capacity against PSD-95. The residual competition of Xph0-ETWV towards all DLG family members is equal to that observed for Xph20-ETWV binding to SAP97, SAP102 or PSD-93. The lower competition activity appears to be due solely to the ETWV motif, as further supported by measured $K_I$ values in the range expected for the isolated ETWV peptide (Supplementary Fig. 28). Efficient competition of Xph20-ETWV against PSD-95 therefore involves an essential combination of the targeting and competing elements.

Compared to the SPR measurements, we observe here a reduction of the $K_I$ difference between PSD-95 tandem PDZ domains and the other tandems (10-fold vs 1000-fold). We attribute this discrepancy to several factors. First, the binding modes are different as the divalent ligand can bind both PDZ domains, whereas our competitor only binds a single binding groove. Second, the divalent ligand presents a much weaker affinity compared to Xph20-ETWV, which prevents a robust

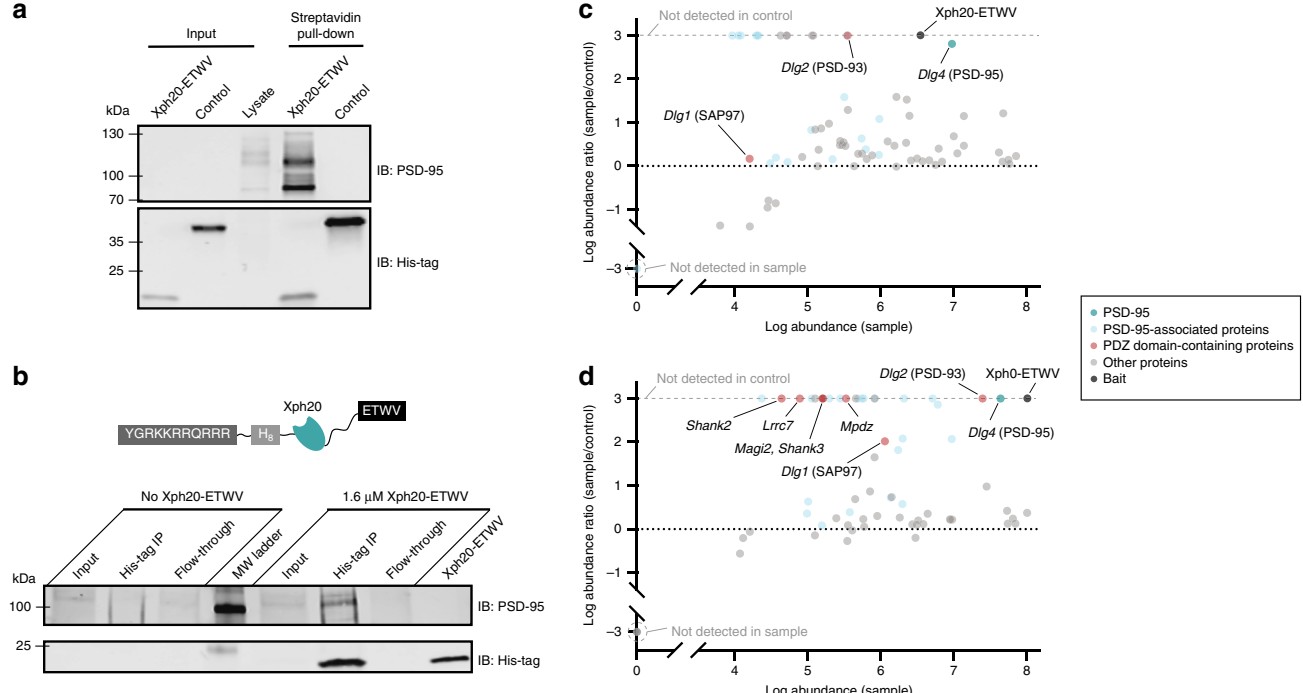

**Fig. 7** Binding to endogenous PSD-95. **a** Western blot analysis of pull-down material from a rat brain lysate using either biotinylated Xph20-ETWV or mScarlet-I as a control. IB stands for immunoblotting. **b** Western blot analysis of immunoprecipitated material from neuron culture using an anti-His tag antibody in presence or absence of TAT-functionalized Xph20-ETWV. **c** and **d** Proteomic analysis of pull-down material from a rat brain lysate using either biotinylated Xph20-ETWV for **c** or Xph0-ETWV for **d** using biotinylated mScarlet-I as a non-binding reference (control). The results correspond to triplicates for each series and include proteins identified with a minimum of two peptides. Source data are provided as Source Data file

quantitative determination of the inhibition constant. Finally, despite our efforts, some detectable amounts of proteolyzed ligand missing the PBM (Supplementary Fig. 29 and Supplementary Table 3) were present together with the competitor. These by-products reduce the effective concentration of the competing ligand and also act as a blocker of the binding site of Xph20 in the Xph20-ETWV construct.

In order to confirm that the binding properties of the engineered competitors are maintained with the full-length endogenous protein, we next evaluated the Xph20-ETWV capacity to recognize PSD-95 from rat brain lysates and primary hippocampal neuron culture. Using a biotinylated competitor and magnetic streptavidin-coated beads, we performed pull-downs on adult rat brain lysates. In comparison to a control protein (mScarlet-I), Xph20-ETWV was clearly able to efficiently bind endogenous PSD-95 as observed by western blot analysis (Fig. 7a). To further confirm binding to PSD-95 in its native environment, we appended a protein transduction domain from the HIV-1 TAT sequence[40] on the competitor, and attempted co-immunoprecipitation (co-IP) in neuron primary cultures. We first evaluated several PSD-95 antibodies for co-IP to avoid potential epitope overlap with Xph20 (Supplementary Fig. 30). Out of four antibodies, three presented an epitope on the first two PDZ domains. Unfortunately, the remaining antibody performed poorly as a PSD-95 immunoprecipitant and only a faint signal could be observed for Xph20-ETWV by western blot after incubation of the cell culture with the TAT-derived competitor. Nevertheless, co-IP with an anti-His tag antibody allowed us to clearly detect PSD-95 (Fig. 7b) thereby demonstrating that Xph20-ETWV can bind endogenous PSD-95 in its native environment.

Finally, we investigated whether our strategy could preserve the selectivity of Xph20-ETWV for PSD-95 in a complex cellular environment where a large number of PDZ domain-containing proteins can in principle interact with class I PBMs such as

ETWV. To this end, we performed a proteomic analysis of the pull-down material from a rat brain lysate using the biotinylated competitor by comparison to a neutral control protein (mScarlet-I). The results show a clear enrichment for PSD-95 (Fig. 7c) with only two other PDZ domain-containing proteins detected (PSD-93 and SAP97) albeit at much lower abundance. While the presence of these proteins could be the consequence of the ETWV motif, they could also be co-precipitates of PSD-95-associated proteins as previously observed[41]. The enrichment of other PSD-95-associated proteins not containing PDZ domain together with the recent report of supercomplexes composed of PSD-95, PSD-93 and NMDA receptors[42] are consistent with the latter possibility. Comparison of the results obtained in the same conditions with Xph0-ETWV (Fig. 7d), in which numerous PDZ-domain containing proteins are detected in addition to PSD-95, confirms that high selectivity for PSD-95 in complex cellular extracts is achieved by combining Xph20 and the ETWV PBM.

**Precise target identification**. As the PBMs we used can in principle interact with either of the two PDZ domains within the complex, we sought to determine if the first or second PDZ domain was principally targeted by our tools. For this purpose, we used a photocrosslinking approach (Fig. 8a). The unnatural amino acid *p*-azidophenylalanine (pAzF) photocrosslinker was introduced in the construct sequence by the amber suppression method[43]. We hence placed an amber stop codon at the -5 position of the PBM by site-directed mutagenesis. This position was chosen as it is not highly conserved in the consensus recognition motif of both domain 1 and 2, and we have previously shown that incorporation of aromatic residues at this site did not significantly modify the binding properties of partner sequences[44]. The pAzF-containing mutants of both Xph20-Stg and

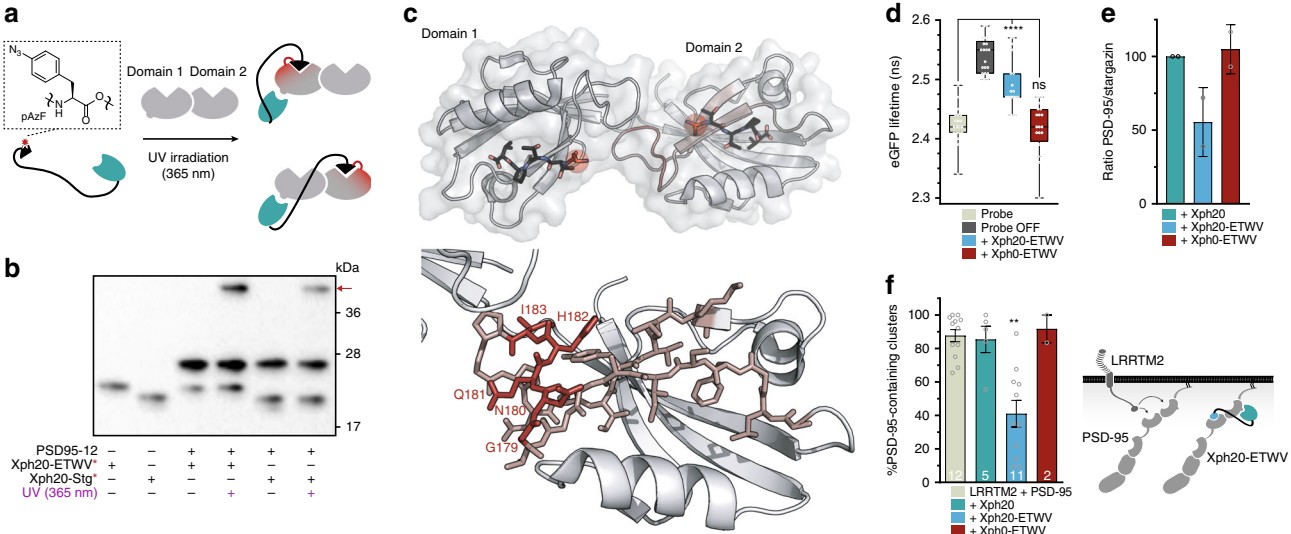

**Fig. 8** Selective blocking of PSD-95 PDZ domain 2 with engineered ligands. **a** Schematics of the photocrosslinking assay. Amber codon suppression method is used to insert *p*-azidophenylalanine (pAzF) into the competitor in the vicinity of PDZ domain-binding motif to create a covalent bond with the target domain in the region of interaction with the motif. **b** Photocrosslinking of Xph20 derivatives with PSD-95 tandem PDZ domain analysed by anti-His tag western blot. The red arrow indicates the photocrosslink products. For uncropped blots, see Supplementary Fig. 28. **c** Position of PSD-95 photocrosslinked fragment with Xph20-ETWV* and -Stg* identified by LC–MS/MS. Top: PSD-95 domains 1 and 2 structure (PDB ID 3GSL) with ligands modelled in (RTTPV, black sticks). The red spheres represent pAzF position in the binding motifs. Bottom: zoom in the identified region on PSD-95. The salmon and red sticks represent the identified fragment with the red highlighting the most likely candidate residues. **d** Competition in cellular environment measured by FRET/FLIM. The box plots show median, first and third quartile, with whiskers extending to the minimum and maximum of all individual data points (****$P <$ 0.0001, ANOVA test followed by a Tukey's Multiple Comparison Test 2 by 2.). Each data point represents a unique cell. **e** Co-immunoprecipitation of PSD-95 by stargazin in the presence of Xph20-ETWV and controls ($n = 2$ independent experiments). The bars represent the mean ± s.d. **f** Schematics of LRRTM2/PSD-95 interactions and bar graph of the percentage of LLRTM2 clusters containing PSD-95. The bars represent the mean ± s.e.m. and numbers in the columns indicate the number of cells (**$P < 0.01$, non-parametric ANOVA test followed by a Dunn's Multiple Comparison Test 2 by 2.). Source data are provided as Source Data file

Xph20-ETWV were produced in *E. coli* in the presence of an additional plasmid encoding an engineered pair of tRNA and aminoacyl-tRNA synthetase to direct the incorporation of the unnatural amino acid. Crosslinking was performed at 365 nm in the presence of the tandem PDZ domains, and generated a single additional band with a molecular weight consistent with a covalent crosslink with the tandem PDZ domains (Fig. 8b and Supplementary Fig. 31a). Selectivity of the modified ligands was first verified by performing the photocrosslinking in presence of a mixture of the tandem domains of PSD-95, SAP97, SAP102 and PSD-93 (Supplementary Fig. 31b, c). The nature of the resulting complexes could be distinguished by size, and we observed a predominant band that corresponds to the crosslink with PSD-95 with minor amounts of a crosslink with SAP97. Further analysis by liquid chromatography tandem mass spectrometry (LC/MS/MS) confirmed these results (Supplementary Fig. 31c) and support their interpretation as the consequence of a 1:1 complex between the tandem PDZ domains and the engineered competitor. Introduction of pAzF at the -5 position therefore did not significantly modify the selectivity of the ligands. We next excised the bands corresponding to the photocrosslinked products of Xph20-Stg* and –ETWV* with PSD-95. Analysis of the trypsin-digest fragments by LC/MS/MS identified the same fragment of PDZ2 for both ligands, specifically the βB and βC strands and connecting loop (Fig. 8c, Supplementary Fig. 32 and Supplementary Tables 4 and 5). Although specific residues in the direct crosslink could not be identified with precision, the loop that connects the two β strands is consistent with the expected position of the pAzF residue within the second PDZ domain-binding groove. The absence of observed fragments from PDZ1, together with robustness of the pAzF-PBM to generate photocrosslink adducts (as seen in Supplementary Fig. 31), strongly supports the

second PDZ domain as the main target of the two engineered competitors.

To determine if PDZ1 binding could be compromised by interaction with Xph20, we performed a crosslinking experiment using Xph20-ETWV* with just domain 1 and observed that the ETWV motif could indeed interact in this configuration with the binding groove (Supplementary Figs. 33 and 34a and Supplementary Table 6). Next we attempted to selectively strengthen the targeting to PDZ1 by using a described selective PBM for domain 1 from the GluK2 C-terminus[45] (-ETMA). In this case, the crosslinking experiment provided us with fragments that belonged to both binding grooves as well as the linker region (Supplementary Figs. 33 and 34b and Supplementary Table 7) suggesting that the motif exchanges between the two domains. These results suggest that the origin of the targeting to a domain could be the consequence of both the respective affinity of the PBM for each of the two domains and of the conformation(s) of the tandem that would lead to more efficient access to domain 2. Overall, we conclude that our selective targeting strategy is not only validated but also robust and easily amenable to further modification and improvement of linker length, targeting elements and competing motif.

**Competing properties in cellular environment.** Finally, the capacity of the engineered competitor to disrupt specific interactions was assessed in a cellular environment using three approaches (Fig. 8d–f and Supplementary Fig. 35). We first used a FRET/FLIM approach. As initial attempts with the stargazin/PSD-95 system did not yield conclusive results, we designed an intramolecular FRET system that only reports binding to the second PDZ domain. The system consists of the tandem PDZ

domains in which the donor protein (eGFP) was inserted on the N-terminus of PDS-95 PDZ1. The acceptor protein (mCherry) was inserted on the C-terminus of PDZ2 with a long linker to separate mCherry from the tandem and the donor. The C-terminus of mCherry was functionalized with the last 14 amino acids of stargazin to allow monitoring of PDZ domain competing binders. Additionally, to be able to exclusively focus on interactions mediated by the second PDZ domain, the PDZ1 was mutated (H130V) to impair its binding properties[25]. The reporter showed a shorter lifetime than soluble eGFP indicating proximity of the two fluorescent proteins and the lifetime was increased upon expression of a reporter deletion mutant lacking the stargazin PBM. A similar increase in lifetime was observed when the reporter was co-expressed with the Xph20-ETWV competitor, but not with the Xph0-ETWV construct, demonstrating that efficient competition can be achieved in cells and requires both Xph20 and the PBM. This result was confirmed by a co-IP assay (Fig. 8e and Supplementary Fig. 35d, e) between stargazin and a mutated PSD-95 in which only PDZ2 was left fully functional (for the reasons mentioned above). Stargazin was then immunoprecipitated from 293T cell lysates transfected with stargazin, the PSD-95 mutant and either Xph20-ETWV, Xph0-ETWV or Xph20. In these conditions, and in agreement with the FRET assay, only Xph20-ETWV was able to reduce the amount of co-precipitated PSD-95. Finally, the competing property of Xph20-ETWV was evaluated with a fully functional PSD-95 against a synaptic adhesion protein, LRRTM2, reported to be a PSD-95 partner that binds PDZ domains 1 and 2 (ref. [46]) (Fig. 8f and Supplementary Fig. 35f, g). In COS-7 transfected cells, recruitment of PSD-95 at LRRTM2 clusters was not detectably affected by co-expression of Xph20 or Xph0-ETWV. In contrast, expression of Xph20-ETWV led to more than a 50% decrease of recruited PSD-95 as a direct result of competition for PDZ2. These experiments together demonstrate the capacity of Xph20-ETWV to selectively and efficiently bind the PSD-95 second PDZ domain and to consequently block PDZ domain 2-mediated interactions in a cellular context.

## Discussion

We report here a semi-rational strategy used to design selective binders of a single PPI domain within a family of highly sequence-conserved PPI domains. By combining a non-selective minimal binding motif to a protein scaffold evolved to specifically recognize a target, we have dramatically increased the selectivity for the PSD-95 second PDZ domain. PDZ domains constitute a representative class of PPI module that is characterized by fast kinetic interactions in the micromolar range that depend on a minimal set of consensus residues (at 0 and −2 positions from the C-terminus). Designing and validating a methodology to generate molecular tools that selectively target and modulate the function of a single PDZ domain of PSD-95 addresses challenges that pertain both to the general conception of PPI modulators and to the investigation of one of the main synaptic scaffold proteins.

The selection that we used to evolve PDZ domain binders was not directed at any specific sites and in particular was not performed by blocking the PDZ domain binding grooves. It is interesting to note that, based on the phage-ELISA and pull-down results, most of the clones that we isolated with such an unbiased selection scheme showed strong specificity for PSD-95. Although epitopes were only mapped for three of the clones, we can reasonably speculate that given their specificity most bind to regions outside of the conserved binding grooves. This observation contrasts with the reported tendency of [10]FN3-derived binders to principally target functional sites (peptide-binding grooves as would be expected for PDZ domains)[30]. Our results suggest that

parts of the first PDZ domain of PSD-95 constitute hot-spots for PPI as they appear to be involved in the binding of all the isolated clones. This finding validates the benefit of choosing to keep intact the tandem PDZ domain supramodule for our selection. Specificity of the characterized clones is principally driven by a single residue, which indicates that even minimal surface variations can be exploited to generate specific protein- and more specifically [10]FN3-based binders. It is of interest that epitopes for the selected clones are conserved within numerous species (Supplementary Fig. 36) and are not involved in any reported post-translational modifications[47] (Supplementary Fig. 37). This property thus extends the scope of application beyond the species we have used for selection (*Rattus norvegicus*) and guarantees recognition of PSD-95 regardless of its cellular state.

An unexpected finding during structural characterization of the unbound clones was that they exist in two conformations, and this may constitute an inevitable trade-off for installing diversity on two of the main loops. Molecular dynamics studies also predict strand swapping at high temperature for these [10]FN3 terminal strands[48]. This structural plasticity of the clones may have consequences on their interaction with PSD-95 and result in a complex interaction mode. In our analyses, quantitation of the binding properties were performed using a simple 1:1 Langmuir binding model as we only observe a unique 1:1 complex for all the clones as investigated by NMR.

If we compare the [10]FN3 domain part of our modulators to some of the commercially available PSD-95-specific antibodies, epitopes are also found in fragments that encompass the same surface region targeted by Xph15, Xph18 and Xph20 (Supplementary Fig. 30b), suggesting shared epitopes that may rely on the same residues to achieve specificity. In comparison to antibodies, the small size and synthetic nature of our binders constitute an advantage by allowing extensive biophysical and biochemical characterization of the tools, although the monovalence and reduced binding surface of unappended Xph compared to antibodies can lead to reduced affinities. Only two other protein-based binders of PSD-95 have been developed by other groups for imaging purposes. One is derived from a single chain variable fragment (PF11)[49] that was obtained from a selection against the full-length palmitoylated form of PSD-95. The other molecule is an evolved [10]FN3 domain (PSD95.FingR)[50] that was selected against the SH3-GuK domains of PSD-95. While PF11 is described as recognizing only the palmitoylated form of PSD-95 (and not the one of PSD-93), PSD95.FingR also binds to SAP97 and SAP102. In both cases, the epitopes were not clearly established and the affinities not determined. In contrast, Xph15, Xph18 and Xph20 represent specific and synthetic small domains that bind to PDS-95 with precisely identified epitopes. Their high specificity and apparent non-hindrance of peptide ligand binding to PSD-95 also constitute a base for further engineering to develop imaging tools to monitor endogenous PSD-95 with minimal perturbation.

The combination of synthetic binders has already been described as a strategy to improve the affinity for a target; examples include fluorescent proteins[51], anti-EGFR $V_H$Hs[52], a PDZ domain binder[53,54], or even kinases with [10]FN3-derived binders[35]. Our aim here was different, although the final effect also relies on affinity improvement: we use the [10]FN3 moieties to increase the selectivity of a competing motif towards a protein of interest. The fusion of moderate to weak binders (100 nM–10 μM) created strong binders with affinities in the picomolar range, arising from slower $k_{off}$. Determination of the effective concentrations ($C_{eff}$)[55] resulting from the tethering of two binders via a linker, as a quantitative indicator related to the affinity enhancement, provided us with values mostly in the 10–50 mM range (Supplementary Table 8). These values are

overall higher than those reported for a bivalent system involving PDZ domain-mediated interactions[53]. While the high $C_{eff}$ values are a direct consequence of the strong affinity of the fusions, they may also reflect our particular case where the tethered PBM can bind to two different and also tethered PDZ domains. In this context, the deliberate use of weaker PBMs is a possible approach to improve the competitor selectivity by reducing the off-target effects. This increased selectivity would result from a loss of interaction with cellular concentrations of SAP97, SAP102 or PSD-93 at the expense of an acceptable loss of the competitor affinity. Other directions for further engineering are prompted by the characterization of the PDZ domain binding groove targeted by the two competitors (Xph20-ETWV and Xph20-Stg). In a first step, the linker length could be adjusted to closely match the distance between the epitope and peptide-binding groove. Secondly, in order to generate PDZ1 blockers, the nature of the peptide source could be varied to include PBMs more selective for this domain (such as attempted with GluK2 PBM) together with an adaptation of the linker length.

As an important validation step, we have shown that the competing properties and the selectivity of the clone Xph20-ETWV were preserved in cellular environments. Given the lack of chemical tools to selectively inhibit single PDZ domains within conserved protein families, our competitors constitute a unique opportunity to investigate the role of endogenous PSD-95 without impacting other paralog family members. Natural partners of the second PDZ domain of PSD-95 include essential synaptic proteins such as TARP-containing AMPAR receptors, NMDA receptors, LRRTMs and nNOS[13]. Even though some of these partners form multimeric protein complexes that rely on multivalent interactions to efficiently anchor to synaptic PSD-95, we have nevertheless demonstrated the capacity of Xph20-ETWV to compete against a divalent ligand. An additional advantage of the relatively small size of the protein-based competitors (~16 kDa) is that they can either be genetically encoded to control levels of expression and cell-type targeting, or exogenously produced as a cell-permeant version for acute effect, such as done for $V_HH$[56]. Finally, the pAzF-containing ligands could also constitute selective irreversible inhibitors of the second PDZ domain of PSD-95 after light-triggered crosslinking.

In summary, our two-step strategy allowed us to circumvent the challenges associated with direct generation of selective or specific competitive binders targeting a highly conserved binding groove within members of a same protein family. With the recent emergence of directed evolution techniques and development of small size synthetic binding domains[30,57–59], we foresee that our strategy will constitute an attractive and powerful method to address the design of selective inhibitors to target other conserved PPI surfaces.

## Methods

**Phagemid construction.** The plasmids generated and the primers used in this study are listed in Supplementary Tables 9 and 10, respectively.

The phagemid (referred to as pSEX84) was derived from the commercial pSEX81 Surface Expression Phagemid Vector (PROGEN). The pSEX81 vector EcoRI restriction site was modified to a HindIII site and the *LacI^q* transcriptional repressor gene (containing a single T → C mutation at the -35 position of the promoter region of *LacI*[60]) was inserted between XhoI restriction site followed by a strong terminator gene sequence (tHP) upstream of the lac promoter[61]. The sequence of the tenth subunit of human fibronectin type III repeat (10FN3) was designed as in Karatan et al.[62] and the gene was synthesized by GenScript. The synthesized gene, Xph0, was flanked by HindIII and BamHI restriction sites in 5′ and 3′, respectively. The BC, DE and FG loops were replaced by serines in place of most of the wild-type residues to generate an inert template. A ribosome binding site coding sequence (5′-agga) was added after the HindIII restriction site and before the methionine initiation codon (ATG). The DsbA signal sequence (MKKIWLALAGLVLAFSASA-) was also included N-terminally to Xph0 for the secretion of the fusion protein into the periplasmic space thus replacing the pelB signal peptide sequence present in the pSEX81 vector, to ensure a greater surface

display of 10FN3 (refs. [31,63]). Additional restriction sites (NotI and KpnI) were added into Xph0 sequence as well as His8 tag, a FLAG tag and the Amber stop codon at the end of the sequence after a short GGSGGS linker. The sequence was optimized for expression in *E. coli*. The resulting construct was subcloned into the pSEX84 vector between HindIII and BamHI restriction sites, in frame with the phage particle minor coat protein pIII (g3p), under the lac operon gene regulatory system.

**Vector construction.** To produce PDZ domains and Xph clones under various forms, the commercial vector pET-24a(+) (Novagen) was modified to generate pET-IG, pIGc, pET-SUMOc, pET-IG-TAT, pbIG and pIGb. In order to express proteins with N-terminal His10-tag, pET-24a(+) was modified between NdeI/XhoI sites to generate pET-IG that presents from 5′ to 3′, after the start codon and in frame, a His10-tag, the Tobacco Etch Virus (TEV) cleavage site (-ENLYFQG-) and a multiple cloning site (BamHI-XhoI). In order to express proteins with a C-terminal His10-tag, pET-24a(+) was modified between NdeI and HindIII sites to generate pIGc that presents from 5′ to 3′, after the start codon and in frame, a multiple cloning site (NdeI-XhoI) and a His10-tag immediately followed by a stop codon. In order to express proteins with a C-terminal SUMO fusion and His10-tag, pET-24a(+) was modified at the XhoI site to generate pET-SUMOc that presents from 5′ to 3′, after the start codon and in frame, the source vector T7 tag and multiple cloning site (BamHI-XhoI), a (-GSS-)2 linker, the TEV cleavage site, the SUMO (small ubiquitin-modifier) tag and a His10-tag immediately followed by a stop codon. In order to express proteins with an N-terminal TAT sequence, pET-24a(+) was modified between NdeI/XhoI sites to generate pET-IG-TAT that presents from 5′ to 3′, after the start codon and in frame, a HIV-1 TAT sequence[40] (YGRKKRRQRRR), a His10-tag, the TEV cleavage site and a multiple cloning site (BamHI-XhoI). In order to express proteins with an N-terminal biotinylation tag and a His10-tag, pET-24a(+) was modified between NdeI/XhoI sites to generate pbIG that presents from 5′ to 3′, after the start codon and in frame, a Biotin acceptor peptide (AP-tag or AviTag, -GLNDIFEAQKIEWHE-), a His10-tag, the TEV cleavage site and a multiple cloning site (BamHI-XhoI). In order to express proteins with a C-terminal biotinylation tag and a His10-tag, pIGc was modified at the XhoI site to generate pIGb that presents from 5′ to 3′, after the start codon and in frame, a multiple cloning site (NdeI-XhoI), a Biotin acceptor peptide and a His10-tag immediately followed by a stop codon. The pET-NO vector was previously described[25] and contains after the initial methionine a His10-tag, the TEV cleavage site (-ENLYFQG-) and a multiple cloning site (BamHI-XhoI). To improve biotinylation yields, the biotin ligase BirA (from the pDisplay-BirA-ER, gift from Alice Ting, Stanford University, Addgene plasmid #20856)[64] was cloned into a pACYC-duet-1 vector as a fusion to mCherry to generate pACYC-mCh-BirA.

**Plasmid construction.** The amino acid sequences of the main constructs are listed in Supplementary Table 11.

The cDNAs of PSD-95, SAP97, SAP102 and PSD-93 (gift from Nathalie Sans, University of Bordeaux) were all from the rat species. Boundaries used for the first two PDZ domains (tandem) are the following residues 61–249 for PSD-95 (PSD-95-12), 145–333 for SAP102 (SAP102-12), 220–408 for SAP97 (SAP97-12) and 93–282 for PSD-93 (PSD-93-12). Boundaries for the isolated PDZ domains 1 and 2 of PSD-95 were, respectively, residues 61–152 and 155–249. Xph0 was synthesized as described above (Genscript) and Xph clones were directly PCR-amplified from the isolated colonies obtained at the end of the selection after sequencing. The S63K mutation in Xph clones (the numbering corresponds to the one used for the wild-type 10FN3, PDB ID 1FNA), F119R in PSD-95 and R278F in SAP97 were inserted by site-directed mutagenesis.

PSD-95-eGFP and Stargazin-mCherry were previously described[25]. The palmitoylation site mutant of PSD-95 (C3S, C5S) was obtained from PSD-95-eGFP by site-directed mutagenesis. SAP97-eGFP was obtained by first cloning the entire SAP97 gene into a pcDNA3 vector, then inserting by site-directed mutagenesis XhoI and NheI sites at the 379 position (after the second PDZ domain) and cloning between these two sites and in frame PCR-amplified eGFP. PSD-93-eGFP was obtained by cloning in two steps the N-terminal and C-terminal parts of PSD-93 into the PSD-95-eGFP plasmid around the eGFP domain (replacement of PSD-95 by PSD-93) using HindIII and AgeI sites and BsrgI and XhoI sites for the N- and C-terminal parts, respectively. The eGFP domain was inserted after the second PDZ domain in position 282. TM-mCherry-Xph constructs were obtained by digesting and ligating the corresponding Xph from pET-IG (resulting in an N-terminal linker of 19 amino acids) into Stargazin-mCherry plasmid C-terminally to mCherry (replacement of Stargazin C-terminus) between NdeI and XhoI sites.

The PSD-95-12 FRET reporter was generated by first synthesizing (Genscript) a backbone incorporating various restriction sites, (GGS)n linkers to separate the elements and the last 18 amino acid from Stargazin. The synthetic fragment was cloned into an intermediary vector and the donor (eGFP), acceptor (mCherry) and PSD-95-12 were sequentially inserted in the sequence. Binding to PDZ domain 1 was impaired by a point mutation (H130V)[25] by site-directed mutagenesis. The final gene (eGFP-PSD-95-12[H130V]-linker-mCherry-linker-Stg) was transferred into the pcDNA3.1 vector. A control construct was produced similarly by omitting the last 18 amino acid from Stargazin. For expression of the competitors and related controls in eukaryotic cells, three fragments corresponding to mIRFP670-Nuc (fluorescent protein followed by three NLS sequences, synthesized by

Eurofins), TEV-TEVcs (TEV protease followed by the TEV cleavage site, synthesized by Eurofins) and the competitor (Xph20-ETWV [S63K], Xph0-ETWV, Xph20 [S63K], obtained by PCR amplification or digestion) were assembled into a pCAG vector.

The gene corresponding to Xph20 followed by a linker composed of 12 repeats of the GGS motif, the ETWV motif, and a stop was synthesized (Eurofins) and inserted into pET-IG and pbIG vectors between BamHI and XhoI sites. All other ETWV-fusion constructs were then obtained by using a classical ligation, inserting the desired gene sequence (Xph0, Xph15 or Xph18) between BamHI and NotI restriction sites in place of Xph20 sequence. The Xph20-ETMA construct was obtained by mutagenesis, replacing the last two amino acids of Xph20-ETWV by a methionine and an arginine. The fragment corresponding to a 26-amino-acid linker followed by Stargazin 13 last amino acids was synthesized (Genscript) and inserted into pIG-Xph15 after Xph15 sequence with a classical ligation using the XhoI restriction site. This plasmid was used to insert the other Xph sequences between the BamHI and XhoI sites. The constructs were then PCR amplified and transferred into pbIG using the BamHI and BlpI sites. For the amber suppression approach, the TAG amber codon was introduced in the competitors by site-directed mutagenesis at the -5 position. An arginine residue was also introduced by site-directed mutagenesis at -11 and -12 positions leading to Xph20-ETWV* and Xph20-Stg* constructs, respectively, in order to facilitate proteomics experiments (smaller resulting trypsin-digested fragments). For the TAT-Xph20-ETWV construct, Xph20-ETWV was PCR amplified and then inserted into the pET-IG-TAT vector between the BamHI and XhoI restriction sites by ligation.

**Standard protein expression and purification.** For expression of His-tagged and GST-fusion proteins, E. coli BL21 CodonPlus (DE3)-RIPL competent cells (Agilent, 230280) were transformed with the genes of interest inserted into an expression vector (pET-IG, pET-NO, pET-SUMOc or pGEX-4T-2). Transformed bacteria were selected on LB plate containing chloramphenicol (at 34.5 µg mL⁻¹, required to maintain the E. coli strain pACYC plasmid expressing codons for rare E. coli tRNAs) and the appropriate antibiotic for maintaining the expression plasmid (30 µg mL⁻¹ for kanamycin or 50 µg mL⁻¹ for carbenicillin). A total of 5 mL of LB starter cultures with appropriate antibiotics were inoculated into 300 mL of ZYM-5052 auto-inducing medium as described by Studier[65]. This medium contains minimal medium (ZY, Z for 1% N-Z-amine or tryptone, Y indicates 0.5% yeast extract) complemented with 0.2 × metals, 1 × M, 1 × 5052, 2 mM MgSO₄ and antibiotics for maintaining the expression plasmid. Cultures were grown in sterile glass vessels 5 h at 37 °C at 250 rpm. Target proteins were then expressed for ~20 h at 16 °C. After the expression, cultures were centrifuged for 15 min at 6000 × g at 4 °C. Supernatants were discarded; bacteria pellets were resuspended in 35 mL of NaCl 0.9% and transferred to 50-mL Falcon tubes. Bacteria were centrifuged for 30 min at 6000 × g at 4 °C and supernatants were discarded. The pellets were stored at −80 °C until purification.

For His-tagged proteins, the bacteria pellets were resuspended in 40 mL of lysis buffer composed of TA buffer (50 mM Tris.OAc pH 8.0), 1 mg mL⁻¹ lysozyme and 1:1000 dilution of Protease Cocktail III (Calbiochem) and were sonicated at 4 °C (Sonics VC505, pulsed mode with an amplitude of vibration of 40%, for a processing time total of 2 min with a 2-s on cycle and 1-s off cycle). Following 30 min of centrifugation at 10,000 × g at 4 °C, the clarified crude extract was collected and 3 mL of HIS-Buster Nickel Affinity gel (equilibrated beforehand with equilibration buffer (TA buffer with 20 mM imidazole)) was added. After 1 h of incubation at 4 °C with continuous rocking, the resin was centrifuged for 15 min at 3000 × g and the supernatant was discarded. The resin was then resuspended into 40 mL of equilibration buffer, centrifuged fpr 10 min at 3000 × g and the supernatant was discarded. The resin was then resuspended into 10 mL of equilibration buffer and transferred into chromatography columns. The column was washed with 20 mL of washing buffer (TA buffer with 40 mM imidazole) and the protein was then eluted with 10 mL of elution buffer (TA buffer with 500 mM imidazole). EDTA was added to the eluted sample at a final concentration of 10 mM and the proteins were dialyzed three times against 1 × PBS (5 mM Na₂HPO₄, 5 mM NaH₂PO₄, 150 mM NaCl, pH 7.4) at 4 °C.

For the TAT-Xph20-ETWV protein, the purification was made as described for His-tagged proteins, but the TA buffers were made with 2 M NaCl instead of 300 mM NaCl. After elution, the protein was centrifuged and filtered on a 0.22-µm filter and directly loaded onto a size exclusion chromatography column (HiLoad 16/600 Superdex 75 pg) to undergo a gel filtration. For GST fusion proteins, the cell pellets were resuspended in 40 mL of lysis buffer, sonicated as described above and DTT was added at a final concentration of 10 mM. Following 1 h of centrifugation at 14,000 × g at 4 °C, the clarified crude extract was collected and 3 mL of GST-Buster QF Glutathione resin (equilibrated beforehand with 1 × PBS, 60 mL) was added. After 1 h of incubation at 4 °C with continuous rocking, the resin was centrifuged for 5 min at 3000 × g and the supernatant was discarded. The resin was then resuspended and extensively washed with 100 mL of 1 × PBS. The protein was then eluted with 10 mL of elution buffer (50 mM Tris-HCl, pH 8.0 with 20 mM glutathione) and finally dialyzed three times against 1 × PBS at 4 °C.

Following dialysis, proteins (His-tagged or GST-fusion) were filtered on a 0.22-µm filter and loaded onto a size exclusion chromatography column (HiLoad 16/600 Superdex 75 pg) to undergo a gel filtration, using an ÄKTAprime or ÄKTApurifier chromatography system and 1 × PBS with 0.01% Tween 20 as running buffer.

Proteins were recovered, characterized by SDS-PAGE and concentrated using Amicon Ultra-15 Centrifugal Filter Devices (Merck Millipore) with a 10 kDa cut-off. The concentration was determined by absorbance at 280 nm and molar extinction coefficient predicted by ProtParam (web.expasy.org). The proteins were finally aliquoted and flash-frozen with liquid nitrogen for conservation at −80 °C.

**Biotinylated proteins production.** Tandem PDZ domains, Xph clones and Xph fusions to PDZ domain-binding motifs were cloned into pbIG or pIGb vectors by inserting the desired genes between BamHI and XhoI or NdeI and XhoI restriction sites, respectively. Home-made E. coli BL21 (DE3) competent cells (source Thermo Fisher Scientific, C600003) containing the pACYC-mCh-BirA plasmid were transformed with the gene of interest into pbIG or pIGb. Transformed bacteria were selected on LB plate containing chloramphenicol and kanamycin for maintaining the two expression plasmids. A total of 5 mL of LB starter cultures with appropriate antibiotics were inoculated into 300 mL of ZYM-5052 auto-inducing medium containing the appropriate antibiotics (30 µg mL⁻¹ for kanamycin and 34.5 µg mL⁻¹ for chloramphenicol). Cultures were grown in sterile glass vessels for 5 h at 37 °C at 250 rpm. Biotin (50 µM) was then added to the medium and the temperature was lowered to 16 °C for expression of the target proteins (~20 h). Proteins were purified as described for His-tagged proteins. Biotinylation yields were evaluated using either the Pierce Biotin quantitation Kit following the manufacturer's protocol (Thermo Fisher Scientific) or by an SDS-PAGE assay[66].

**Isotopically labelled proteins production.** For minimal media expression and purification of isotopically labelled proteins, isotopically labelled constructs were expressed in BL21 pLysY (New England Biolabs, C3010I) from LB plates containing 30 µg mL⁻¹ kanamycin. Cells pellets from overnight cultures in 10 mL of LB medium containing 30 µg mL⁻¹ kanamycin one used to start growth in 500 mL cultures of M9 minimal medium containing 1 g L⁻¹ ¹⁵NH₄Cl, with double-labelled samples also containing 2 g L⁻¹ ¹³C-glucose. ²H,¹³C,¹⁵N-labelled proteins were grown as for the double-labelled proteins but with ²H₂O instead of water. At an OD₆₀₀ of ~0.8, the cells were induced for protein expression with 0.5 mM IPTG, and incubated at 20 °C, 200 rpm for 12–16 h. The cells were collected by centrifugation for 15 min at 4500 × g and resuspended in sonication buffer (500 mM NaCl, 20 mM imidazole, 50 mM phosphate buffer, pH 7.5) and lysed by sonication (45 cycles with 10 s ON/10 s OFF at 45% Amplitude). The lysate was centrifuged at 20,000 × g for 30 min and the supernatant was loaded onto 2 mL of Nuvia beads (Bio-Rad) pre-equilibrated with sonication buffer. The beads were washed with 10 column volumes of sonication buffer and the recombinant protein was eluted with 1 mL aliquots of elution buffer (500 mM imidazole, 500 mM NaCl, 50 mM phosphate buffer, pH 7.5). The elute containing protein was pooled and the buffer exchanged to PBS by using a PD-10 column (GE Healthcare). Constructs with a cleavable N-terminal His-tag were incubated overnight with TEV protease at room temperature for 12–16 h, with the TEV protease, uncleaved protein and the cleaved His-tag removed by a second passage through the Nuvia beads. The purified sample was then concentrated to >100 µM by using Amicon 3000 MWCO centrifugal device (Merck Millipore), with the concentration determined by absorbance at 280 nm and molar extinction coefficient predicted by ProtParam (web.expasy.org).

For ¹³C/¹⁵N double-labelled PSD-95 PDZ1, PDZ2 and PDZ1–PDZ2, expression was performed in E. coli BL21 CodonPlus (DE3)-RIPL selected from LB plates containing 30 µg mL⁻¹ kanamycin as well as 34.5 µg mL⁻¹ chloramphenicol to maintain the BL21 pACYC plasmid. Starter cultures were first grown at 37 °C shaking at 300 rpm until it reached an OD₆₀₀ of 0.7, then the cells were pelleted by 30 min centrifugation at 5000 × g at 4 °C, washed twice with Minimal medium, and finally resuspended in 500 mL of Minimal medium for expression (with ¹³C glucose and ¹⁵NH₄Cl). The culture was placed back into the incubator at 37 °C for 1 h, shaking at 300 rpm before adding IPTG at 1 mM. Finally, the temperature was lowered to 16 °C for an overnight expression with shaking at 300 rpm. After the expression, cultures were centrifuged for 30 min at 6000 × g at 4 °C. Supernatants were discarded; bacteria pellets were resuspended in 35 mL of NaCl 0.9% and transferred to fresh 50-mL Falcon tubes. Bacteria were centrifuged for 40 min at 6000 × g at 4 °C and supernatants were discarded. The induced cells were then resuspended in 40 mL of lysis buffer and sonicated in a pulsed mode with an amplitude of vibration of 40%, a processing time of 2 min and a 2-s on cycle and 1-s off cycle. Following 1-h centrifugation at 14,000 × g at 4 °C, proteins were purified with a Ni-affinity chromatography as described above. After a dialysis against a TEV cleavage buffer overnight at 4 °C, in the presence of the TEV enzyme that was added directly into the SnakeSkin™ dialysis tubing, proteins were then loaded on a size exclusion chromatography column (HiLoad 16/600 Superdex 75 pg) to undergo a gel filtration, using an ÄKTAprime chromatography system. Proteins were recovered, characterized by SDS-PAGE and concentrated using Amicon Ultra-15 Centrifugal Filter Devices (Merck Millipore) with a 3- or 10-kDa cut-off according to the manufacturer's instructions. They were finally aliquoted and flash-frozen with liquid nitrogen for a −80 °C conservation.

**Unnatural amino-acid-containing proteins production.** The plasmids containing the amber codon were co-transformed in BL21(DE3) (Thermo Fisher Scientific, C600003) with the pEVOL-pAzF (a gift from Peter Schultz, Scripps Research

Institute, Addgene plasmid # 31186) coding for the orthogonal AzF RNA synthetase/AzF tRNA$^{CUA}$ pair derived from *Methanocaldococcus jannaschii*[43]. A total of 5 mL of LB starter cultures with appropriate antibiotics were inoculated into 500 mL of M9-minimal medium (45 mM $Na_2HPO_4$, 25 mM $KH_2PO_4$, 8.5 mM NaCl, 0.1 mM $CaCl_2$, 1 mM $MgSO_4$, 0.03 mg mL$^{-1}$ thiamine, 1 mg mL$^{-1}$ $NH_4Cl$, 2 mg mL$^{-1}$ glucose, 22 nM $FeCl_3$, pH 7.0) supplemented with kanamycin and chloramphenicol (30 μg mL$^{-1}$ for kanamycin and chloramphenicol 10 μg mL$^{-1}$). At an $OD_{600}$ of 0.6, AzF (H-4-Azido-Phe-OH, Bachem) was added to a final concentration of 1 mM and protein expression was induced for 16 h at 37 °C at 250 rpm with 1 mM IPTG and 0.02% arabinose. After the expression, cultures were centrifuged for 15 min at $6000 \times g$ at 4 °C. Supernatants were discarded; bacteria pellets were resuspended in 35 mL of NaCl 0.9% and transferred to fresh 50-mL Falcon tubes. Bacteria were centrifuged for 30 min at $6000 \times g$ at 4 °C and supernatants were discarded. The pellets were stored at −80 °C.

The auto-induced pellets were then lysed and purified as described in the His-tagged purification protocol using here Cobalt-Buster Nickel Affinity gel (Amocol) instead of HIS-Buster Nickel Affinity gel.

**Library construction**. The *E. coli* CJ236 *dut*⁻ *ung*⁻ strain (TaKaRa, E4141S), which lacks the dUTPase and uracil N-glycosylase enzymes, was used to generate uracilated single-stranded DNA template. The *E. coli* TG1 electrocompetent cells (Lucigen, 60502-1) were used to produce the diversified bacterial and phage libraries owing to their wild-type versions of dUTPase and uracil N-glycosylase, favouring the newly synthesized strand propagation and to their F' episome required for infection by M13 vectors. The *E. coli* XL1-Blue (Agilent, 200236) strain was used for all infection steps during panning.

For ssDNA production, the phagemid was transformed into *E. coli* CJ236 and the transformation product was plated onto a fresh Petri plate containing 2x YT medium (per liter: 16 g tryptone, 10 g yeast extract, 5 g NaCl, 0.75% agar (m/v), chloramphenicol (10 μg mL$^{-1}$), carbenicillin (50 μg mL$^{-1}$). After an overnight incubation at 37 °C, a single colony was picked, transferred into 1 mL of 2x YT medium supplemented with 50 μg mL$^{-1}$ carbenicillin, 10 μg mL$^{-1}$ chloramphenicol, 10$^{10}$ M13KO7 helper phages (New England BioLabs) and was incubated with shaking at 200 rpm at 37 °C for 2 h. Kanamycin (25 μg mL$^{-1}$) was added and the culture was incubated at 37 °C, and shaken at 200 rpm for 6 h. After the incubation, the 1 mL culture was transferred to 30 mL of 2x YT in the presence of 50 μg mL$^{-1}$ carbenicillin, 25 μg mL$^{-1}$ kanamycin, 0.25 μg mL$^{-1}$ uridine and bacteria were grown for 20 h at 25 °C (200 rpm). Cells were centrifuged at $33,000 \times g$ for 10 min at 4 °C to clarify the supernatant, which was transferred to a new tube and mixed with 1/5 volume of 20% (wt/vol) PEG8000/2.5 M NaCl (PEG/NaCl) solution and incubated for 5 min at room temperature to precipitate phages. Phages were recovered after a first 10-min centrifugation at $14,000 \times g$, a second brief centrifugation at $5000 \times g$ and subsequent discarding of supernatants. The pellet of phage particles was resuspended in 0.5 mL of phosphate buffered saline (PBS; 137 mM NaCl, 3 mM KCl, 8 mM $Na_2HPO_4$, 1.5 mM $KH_2PO_4$; pH 7.2) and transferred to a 1.5-mL microcentrifuge tube. This phage solution was centrifuged for 5 min at $16,200 \times g$ at 4 °C in a microcentrifuge to get rid of any insoluble precipitate and the liquid supernatant was transferred to a new 1.5-mL microcentrifuge tube. The ssDNA was then extracted and purified with the QIAprep Spin M13 kit (Qiagen) following the modified protocol of Tonikian et al.[67]. Next, 7 μL of MP buffer (Qiagen kit) was added to the 0.5 mL phages solution and mixed, then incubated at room temperature for at least 2 min. All the following steps are performed at room temperature and all the centrifugations were achieved in a microcentrifuge at $6200 \times g$ with a subsequent discard of flow-throughs. The sample was applied to a QIAprep Spin M13 column and centrifuged for 30 s. After discarding the flow-through, 0.7 mL of PB buffer (Qiagen kit) was added to the column and centrifuged for 30 s. A 2-min incubation with 0.7 mL of PB buffer was made and the column was centrifuged for 30 s allowing the DNA to be separated from the protein coat and to be adsorbed to the matrix. A total of 0.7 mL of PE buffer was added to the column and centrifuged for 30 s. After discarding the flow-through, the PB buffer incubation as well as the PE buffer step were repeated to remove residual proteins and salts. The column was centrifuged at $6200 \times g$ for 30 s in a new 1.5-mL microcentrifuge tube to remove residual PE buffer (Qiagen kit) and transferred to a fresh 1.5-mL microcentrifuge tube. For the elution step, 100 μL of buffer EB (Qiagen kit) was added to the column and incubated for 10 min. After a last centrifugation of 30 s, the eluant containing the purified dU-ssDNA was saved. The ssDNA concentration was determined with a Nanodrop spectrophotometer (Thermo Scientific) and the quality evaluated by electrophoresing 3 μL on a 0.8% TAE/agarose gel at 100 V for 1 h.

Combinatorial libraries in which the BC and FG loops of $^{10}$FN3 were diversified were constructed by using pFunkel mutagenesis[33], inspired by Kunkel mutagenesis. Five phosphorylated oligonucleotides were designed to introduce from 5 to 9 NNK random codons at position 25 in the $^{10}$FN3 gene and seven phosphorylated oligonucleotides were designed to introduce 7 to 13 NNK random codons at position 75 corresponding to the BC and the FG $^{10}$FN3 loops, respectively (see Supplementary Note 2 for the oligonucleotides sequence; the numbering corresponds to the one used for the wild-type $^{10}$FN3, PDB ID 1FNA).

The five BC-phosphorylated oligonucleotides were combined in equimolar amounts (1.5 pmol total; Eurogentec) and the seven FG-phosphorylated were combined in the same way (1.5 pmol of FG-oligonucleotides total), these

phosphorylated oligonucleotides were annealed to uracilated ssDNA template, at a molar ratio of 4 (oligonucleotides/ssDNA) in a 0.5-mL Eppendorf tube containing 1 × PfuTurbo Cx hotstart DNA polymerase buffer (Agilent Technologies), 10 mM dithiothreitol (DTT), 0.5 mM NAD$^+$ (New England BioLabs), 0.2 mM dNTPs, and 4% DMSO in a total volume of 94 μL. The annealing was performed by heating to 95 °C for 3 min, then at 60 °C for 3 min. After the annealing step, 200 units of cohesive end Taq ligase (New England BioLabs) and 2.5 units of PfuTurbo Cx hotstart DNA polymerase were added bringing the total volume to 100 μL. Extension and ligation of the mutant strand was performed at 65 °C for 15 min and 45 °C for 15 min. A total of 3.8 pmol of phosphorylated oligonucleotide (5′-P-ggtctggtgctggcattctc) was added and one more cycle of 95 °C for 30 s, 55 °C for 45 s, 65 °C for 10 min and 45 °C for 15 min was performed. Ten units of UDG (uracil-DNA glycosylase, New Englands BioLabs) and 30 units of ExoIII (Exonuclease III, New Englands BioLabs) were added and the mixture was incubated at 37 °C for 1 h followed by an inactivation step for 20 min at 70 °C. Before purification, 1 μL of the pFunkel solution was transformed into 10 μL of *E. cloni* 10G chemically competent cells (Lucigen, 60106-2) and were plated onto 2x YT medium agar-plate with 50 μg mL$^{-1}$ carbenicillin, 10 mM glucose and incubated at 37 °C overnight. The number of colonies obtained generally depends on pFunkel efficiency and the sequence diversity of the DNA library was estimated by performing Sanger sequencing of 96 randomly picked colonies from the obtained transformants. The DNA was then purified using Amicon Ultra-0.5 Centrifugal Filter Devices (Merck Millipore) with a 30-kDa cut-off according to the manufacturer's instructions. The typical volume obtained was 20 μL. The purified DNA was immediately used for electroporation into TG1 electrocompetent cells (Lucigen).

In four pre-chilled 0.1 cm cuvettes (BioRad), the DNA library was electroporated into 250 μL of TG1 cells at 1800V with an electroporator (BioRad MicroPulser) and then incubated with prewarmed Recovery medium (Lucigen) for 1 h at 37 °C with shaking at 250 rpm. A total of 10 μL of the electroporated cells were kept for serial dilutions and plating on 2x YT agar-plate with 50 μg mL$^{-1}$ carbenicillin and 10 mM glucose to determine the electroporation efficiency and estimate the library size. The entire remaining volume was then plated on dishes (Greiner bio-one, 145 mm × 145 mm × 20 mm) containing 2x YT agar-plate with 50 μg mL$^{-1}$ carbenicillin, 10 mM glucose and incubated overnight at 37 °C. Each plate was recovered with 4 mL of 2x YT medium and scraped to recover library-containing TG1 cells. Two more washes with 2 mL of 2x YT medium were made to ensure maximal library recovery. Bacteria were centrifuged at $5000 \times g$ for 10 min at 4 °C and resuspended in fresh 2x YT medium for two times. Bacteria were then flash-frozen with liquid nitrogen after the addition of 20% sterile glycerol and were kept at −80 °C for later use.

**Sequencing analysis software**. SynDivA is a tool implemented in Python for analyzing multiple sequencing results in the context of directed evolution projects using diversified gene libraries. SynDivA works as follows. First, SynDivA pre-processes the sequencing input data using BioPython[68]. It determines the orientation of the sequences and their reading frame, translates the nucleotide sequences into protein sequences and then filters the sequences by removing the ones that do not have pre-defined restrictions sites or contain stop codons in variable regions. Pairwise alignment is performed on the valid sequences using NW-align (http://zhanglab.ccmb.med.umich.edu/NW-align). Sequences are clustered using MCL algorithm[69] and multiple sequence alignment is done using Clustal Omega[70]. SynDivA outputs all the results in an HTML report.

SynDivA is available for use on the instance of Galaxy hosted on CBiB's server via https://services.cbib.u-bordeaux.fr/galaxy/?tool_id=fibronectin&version=1.0.

**Biopanning procedure**. For the phage library preparation, an aliquot of frozen bacterial library was resuspended in 500 mL of 2x YT in the presence of 50 μg mL$^{-1}$ carbenicillin. During the exponential phase of growth ($OD_{600} = 0.4$–0.6), bacteria were infected with M13KO7 helper phage particles (multiplicity of infection or MOI = 20) for 60 min at 37 °C. A total of 2.5 μM IPTG and 30 μg mL$^{-1}$ kanamycin were added and bacteria were grown overnight at 37 °C at 200 rpm. The culture was centrifuged at $16,000 \times g$ for 10 min at 4 °C, the supernatant was transferred to a new tube, mixed with 1/5 volume of PEG/NaCl solution and incubated for 5 min at room temperature. The precipitated phages were centrifuged at $16,000 \times g$ for 10 min at 4 °C and the supernatant was discarded. Following another quick centrifugation, the pellet was resuspended in 15 mL of 50 mM $Na_2HPO_4$, 50 mM $NaH_2PO_4$, 150 mM NaCl, 0.05% Tween 20, 0.5% BSA, pH 7.4 (PBT buffer). To pellet insoluble precipitates, a 5-min centrifugation at $27,000 \times g$ at 4 °C was carried out and the supernatant was transferred in a clean tube. Phage particle concentration was estimated spectrophotometrically ($OD_{268} = 1.0$ for a solution of $5 \times 10^{12}$ phage per mL).

The panning procedure was implemented as follows. The biotinylated target was first pre-incubated with streptavidin-beads (Dynabeads® M-280 Streptavidin, Thermo Fisher Scientific) for 10 min at room temperature. Beads were washed three times and blocked for 1 h at room temperature on a shaker with 1 mL of PBS, 0.2% Tween 20, 1% BSA (PBT2 buffer). Phages were then incubated with the bead-target mix for 15 min at room temperature while shaking. Beads were then washed six times with 1 mL of PBS, 0.2% Tween 20 (PT2 buffer) and two steps of 5 min incubation with PT2 were performed. The washing procedure was ended with three washes with 1 mL of PBS. Elution was performed with 100 μL of 100 mM glycine

pH 2.7 for 10 min with shaking, and the eluate was neutralized by adding 7 μL of 100 mM NaHCO$_3$ pH 9.0. Three consecutive panning rounds were conducted wherein the biotinylated PSD-95-12 target concentration was decreased (round 1: 100, round 2: 20, round 3: 4 nM). The performance of our selection was characterized by an enrichment parameter defined as the ratio of the phage count recovered from the target sorting compared to the count of background binding phages in the negative control (biotin sorting). Neutralized phagemid solution was added to 1 mL of actively growing *E. coli* XL1-Blue (OD$_{600}$ = 0.4–0.6) and incubated for 1 h at 37 °C with shaking. After taking out 10 μL of the infected bacteria for further serial dilutions and plating on 2x YT medium agar-plate with 50 μg mL$^{-1}$ and 10 mM glucose to determine the enrichment ratio, 50 μg mL$^{-1}$ carbenicillin and 10$^{10}$ phage mL$^{-1}$ were added to the medium. After 1 h incubation at 37 °C, the solution was transferred to a 250-mL baffled flask containing 25 mL of 2x YT medium, supplemented with 50 μg mL$^{-1}$ carbenicillin, 25 μg mL$^{-1}$ kanamycin, 2.5 μM IPTG and bacteria were grown overnight at 37 °C at 200 rpm. Phages were isolated by precipitation with PEG/NaCl solution (as in the phage library preparation step) and resuspended in 1.3 mL of PBT2. Their concentration was estimated spectrophotometrically as described above. Plates used for determining the enrichment ratio were also used to select single colonies for phage-ELISA.

**Phage-ELISA.** Phage-ELISAs were performed against the target (biotinylated PSD-95-12) and against other biotinylated tandem PDZ domains to assess the specificity. For binding determination, 96 colonies were randomly picked from the agar plates resulting from the last round of panning. The individual colonies were cultured overnight at 37 °C while shaking in a microplate deep well (Porvair Sciences) containing 450 μL of 2x YT medium, 50 μg mL$^{-1}$ carbenicillin and 10$^{10}$ M13KO7 helper phage per well. The microplate was centrifuged at 1700 × *g* for 10 min and 400 μL of supernatant containing Xph-displaying-phages was transferred to a new microplate deep well containing 400 μL of PBT and used in screening phage-ELISA.

Nunc-Immuno MaxiSorp 96-well plates (Thermo Fisher Scientific) were coated overnight at 4 °C with a mix of 2 μg mL$^{-1}$ of streptavidin. Wells were then blocked with 200 μL of blocking buffer (PBS, 0.5% (wt/vol) BSA) for 1 h at 37 °C. After removing the blocking buffer, the wells were washed four times with PT with an automatized plate washer, and 1.2 equivalents (per total biotin binding sites) of biotinylated targets in 50 μL of PT buffer per well were added and incubated for 15 min at room temperature. Thereafter, wells were washed four times with PT buffer and 100 μL of rescued phage supernatant was added to the ELISA plates and incubated on a rocking shaker for 1 h. Following eight washes with PT, 100 μL of horseradish-peroxidase-conjugated anti-M13 monoclonal antibody (GE Healthcare, diluted at 1:5000 in PBT) was added to each well. Samples were incubated on a rocking shaker for 30 min, and were then washed six times with PT and four times with PBS. Next, 100 μL of 1-Step Ultra (3,3′,5,5′-tetramethylbenzidine) TMB-ELISA was added and the reaction was stopped with 100 μL of 1 M H$_3$PO$_4$. Absorbance was read at 450 nm on a POLARstar Omega plate reader.

The sequence of each clone was determined by performing Sanger sequencing of the 96 selected colonies. Analysis of the sequences was performed using SynDivA software.

**GST pull-down assay.** Plasmids containing Xph clones in a pET-IG vector were transformed into BL21-CodonPlus (DE3)-RIPL competent cells and overnight precultures started from a single colony were used to inoculate 60 mL of LB. The cells were grown at 37 °C until an OD$_{600}$ of 0.6 was reached. Induction was performed by adding 1 mM of IPTG and the cells were grown for an additional 16 h at 16 °C. The cells were then harvested by centrifuging 50 mL of the cultures at 3000 × *g* for 30 min at 4 °C. The pellets (about 0.8 g each) were stored at −80 °C until lysis. Lysis was performed by thawing the cells and resuspending them in 20 mL of cold TA buffer (50 mM Tris.OAc, 150 mM NaCl, pH 8.0) containing 1 mg mL$^{-1}$ of lysozyme and 1 × protease inhibitors (Protease Cocktail III). The mixture was kept under gentle agitation for 15 min at 4 °C and lysed by sonication (Sonics VC505, pulsed mode with an amplitude of vibration of 40%, for a processing time total of 2 min with 1 s pulses and 1 s intervals). The lysates were cleared by centrifugation at 9000 × *g* for 30 min at 4 °C and used immediately for the binding assays.

The His-tagged Xph clones lysates were added to 40 μL of His Mag Sepharose excel magnetic beads (GE Healthcare) that had been first washed twice with 1 × PBS with 0.05% Tween 20. The mixtures were incubated at 4 °C under agitation for 1 h. The beads were collected with a magnet and washed eight times with wash buffer composed of 1 × PBS with 0.05% Tween 20, 500 mM NaCl and 10 mM imidazole (2 × 10 mL, then 6 × 2 mL). One-fifth of the recovered beads were incubated with 20 μM of purified tandem PDZ domains GST fusions in 200 μL of 1 × PBS with 0.05% Tween 20 for 1 h at room temperature under agitation. The beads were collected with a magnet and washed six times with wash buffer. The proteins were then eluted with 20 μL of 500 μM imidazole solution in TA buffer. The eluates were mixed with a 2 × sample buffer, heated for 5 min at 75 °C and ran (5 μL per lane) on a 4–20% Bis-Tris precast gel. The gel was stained with colloidal blue and analysed with a ChemiDoc MP imager (Bio-Rad). The amount of the various Xph clones were similar and binding was calculated by the determining the ratio between the intensity of the band corresponding to the pulled-down GST

fusion protein (tandem PDZ domains) vs the intensity of the band corresponding to the Xph clone.

**FRET/FLIM.** A day before transfection, 30,000 COS-7 cells (ECACC-87021302) per 12-well-plates were plated on glass coverslips. Cells were maintained with DMEM medium supplemented with Glutamax and 10% FBS. Cells were then transfected using a 2:1 ratio X-treme GENE HP DNA transfection reagent (Roche) per microgram of plasmid DNA with a total of 0.5 or 0.25 μg DNA per well. All direct binding experiments were performed using a 1:1 molar ratio between the donor- and acceptor-containing plasmids. All competition experiments were performed using a 1:5 molar ratio between the FRET reporters and the competitor (or control) plasmids, respectively. After 30 min incubation at room temperature, the DNA-Xtreme complex solution was added dropwise in each well. Experiments were performed after 24 h of expression. For FLIM measurements, coverslips were transferred into a ludin chamber filled with 1 mL of fresh Tyrode's buffer (20 mM glucose, 20 mM HEPES, 120 mM NaCl, 3.5 mM KCl, 2 mM MgCl$_2$, 2 mM CaCl$_2$, pH 7.4, osmolarity around 300 mOsm kg$^{-1}$ and pre-equilibrated in a CO$_2$ incubator at 37 °C).

For the time domain analysis (time-correlated single photon counting (TCSPC)) method, the confocal microscope was a Leica DMR TCS SP2 AOBS on an inverted stand (Leica Microsystems, Mannheim, Germany), using objectives HC Plan Apo CS 20X dry NA 0.70 and HCX Plan Apo CS 40X oil NA 1.25 and HCX Plan Apo CS 63X oil NA 1.32 and HCX Plan Apo CS 100X oil NA 1.40. The lasers used were argon (458, 476, 488, 496, and 514 nm), green Helium-Neon (543 nm) and red Helium-Neon (633 nm). The microscope was equipped with a galvanometric stage in order to do *z* acquisition.

FLIM was performed with the TCSPC method on a multiphoton SP2 AOBS Leica confocal system using a HC Plan Apo CS 63X oil NA 1.32 objective (Leica Microsystems, Mannheim, Germany). The pulsed light source was a tunable Ti: Sapphire laser (Chameleon, Coherent Laser Group, Santa Clara, CA, USA). The laser was used at 900 nm in order to excite GFP. The laser repetition frequency was 80 MHz which gave a 13-ns temporal window for lifetime measurements. The system was equipped with the TCSPC from Becker and Hickl (Berlin, Germany) comprising a PMC-100 detector characterized by a transit time spread of ~150 ps and SPC 830 photon counting and timing electronic card. A 510/40 band-pass filter (Chroma Technology, Rockingham VT, USA) was used to specifically detect the donor fluorescence. Fluorescence decay curves were obtained using single spot mode of SPCM software (Becker and Hickl). Experiments corresponding to Fig. 2f were obtained using this method.

For the frequency domain analysis (LIFA) method, the spinning disk microscope was a Leica DMI6000 (Leica Microsystems, Wetzlar, Germany) equipped with a confocal Scanner Unit CSU-X1 (Yokogawa Electric Corporation, Tokyo, Japan) using objectives HC PL Fluotar 10X dry NA 0.3, HCX PL Apo 40X oil NA 1.25, HCX PL Apo CS 63X oil NA 1.4 and HCX PL Apo 100X oil NA 1.4 and an Evolve EMCCD camera (Photometrics, Tucson, USA) or a HQ2 CCD camera (Photometrics, Tucson, USA). The diode lasers used were at 408, 491, 561 and 638 nm. The 37 °C atmosphere was created with an incubator box and an air heating system (Life Imaging Services, Basel, Switzerland). This system was controlled by MetaMorph software (Molecular Devices, Sunnyvale, USA).

The FLIM measurements were done with the LIFA fluorescence lifetime attachment (Lambert Instrument, Roden, Netherlands), which allows the generation of lifetime images by using the frequency domain method. This system consisted of a modulated intensified CCD camera Li2 CAM MD, a modulated light excitation light source, a modulated GenIII image intensifier. For widefield epi-illumination, modulated LED (Light-Emitted-Diode) were used at 451 nm (3 W), 477 nm (3 W) or 523 nm (3 W). Using the LIFA with the spinning-disk, the excitation source was a 60-mW modulated laser at 488 nm (Omicron, Rodgau-Dudenhofen, Germany).

Both the LED and the intensifier were modulated at frequency up to 100 MHz. A series of 12 images was recorded for each sample. By varying the phase shifts (12 times) between the illuminator and the intensifier modulation, we could calculate the phase and modulation for each pixel of the image. Consequently, we could determine the sample fluorescence lifetime image using the manufacturer's software LI-FLIM software. Lifetimes were referenced to a 1-μM solution of fluorescein in Tris-HCl (pH 10) or a solution of erythrosin B (1 mg mL$^{-1}$) that was set at 4.00 ns lifetime (for fluorescein) or that was set at 0.086 ns (for erythrosin B)[39]. Experiments other than Fig. 2f were obtained using this method. For competition experiments, only cells presenting a high level of expression of the competitor or control as measured by mIRFP670 fluorescence were taken into consideration. Datasets were analysed by an ANOVA test followed by a Tukey's Multiple Comparison Test 2 by 2.

**NMR spectroscopy.** NMR spectra were recorded at 298 K using Bruker Avance III 700 MHz or 800 MHz spectrometers equipped with a triple resonance gradient standard probe or cryoprobe, respectively. Topspin versions 2.1, 3.2 and 3.5 (Bruker BioSpin) were used for data collection. Spectra processing used NMRPipe[71] followed by analysis with Sparky 3 (T.D. Goddard and D.G. Kneller, University of California) or CARA (RLJ. Keller, ETH Zurich).

**Backbone chemical shift assignment of PSD-95.** Spectra for the assignment of backbone $^1H^N$, $^{13}C'$, $^{13}C^\alpha$, $^{13}C^\beta$ and $^{15}N^H$ nuclei of the first PDZ domain of rat PSD-95 (residues 61–152 with an N-terminal Ser-Gly-Ser- remaining after cleavage by TEV protease) were collected on a 345 μM $^{13}C$,$^{15}N$-labelled sample in PBS with 10% $D_2O$ added for the lock. NMR assignment used 2D $^{15}N$-HSQC, 3D HNCO, 3D HNCACO, 3D HNCA, 3D CBCACONH and 3D HNCACB spectra. The backbone assignment is nearly complete, except for missing assignments for the initial Ser-Gly- remaining from the His-tag, the backbone carbonyl for Asp91, and the amide $^1H^N$ and $^{15}N^H$ chemical shifts for Asn72, Ser73, Leu75 and Glu122. Also assigned are sidechain $^{15}N^{\delta2}$, $^1H^{\delta2}$ and $^{13}C^\gamma$ nuclei from Asn72, Asn85, Asn114 and Asn121, as well as sidechain $^{15}N^{\epsilon2}$, $^1H^{\epsilon2}$ and $^{13}C^\delta$ nuclei from Gln107. Chemical shift assignments for PSD-95 PDZ1 were deposited in the Biological Magnetic Resonance Data Bank (BMRB) as entry 27309. An annotated $^{15}N$-HSQC is included in Supplementary Fig. 13a.

Spectra for the assignment of backbone $^1H^N$, $^{13}C'$, $^{13}C^\alpha$, $^{13}C^\beta$ and $^{15}N^H$ nuclei of the second PDZ domain of rat PSD-95 (residues 155–249 with an N-terminal Ser-Gly-Ser- remaining after cleavage by TEV protease) were collected on a 196-μM $^{13}C$,$^{15}N$-labelled sample in PBS with 10% $D_2O$ added for the lock. NMR assignment used 2D $^{15}N$-HSQC, 3D HNCO, 3D HNCACO, 3D HNCA, 3D CBCACONH and 3D HNCACB spectra. The backbone assignment is nearly complete, except for missing assignments for the initial Ser-Gly-Ser- remaining from the His-tag, all nuclei for Pro16, and the amide $^1H^N$ and $^{15}N^H$ chemical shifts for Lys168, Leu170 and Ser217. Also assigned are sidechain $^{15}N^{\delta2}$, $^1H^{\delta2}$ and $^{13}C^\gamma$ nuclei from Asn180, Asn187, Asn216, Asn234 and Asn248, as well as sidechain $^{15}N^{\epsilon2}$, $^1H^{\epsilon2}$ and $^{13}C^\delta$ nuclei from Gln181 and Gln207. Chemical shift assignments for PSD-95 PDZ2 were deposited in the Biological Magnetic Resonance Data Bank (BMRB) as entry 27310. An annotated $^{15}N$-HSQC is included in Supplementary Fig. 13b.

Spectra for the assignment of backbone $^1H^N$, $^{13}C'$, $^{13}C^\alpha$, $^{13}C^\beta$ and $^{15}N^H$ nuclei of the tandem PDZ1-PDZ2 domains of rat PSD-95 (residues 61–249 with an N-terminal Ser-Gly-Ser- remaining after cleavage by TEV protease) were collected on an 80 μM $^2H$,$^{13}C$,$^{15}N$-labelled sample in PBS with 10 % $D_2O$ added for the lock. NMR assignment used 2D 15N-TROSY, 3D TROSY-HNCO, 3D TROSY-HNCACO, 3D TROSY-HNCA, 3D TROSY-CBCACONH, and 3D TROSY-HNCACB spectra. Assignments are missing for all nuclei in Pro153, Pro154, Lys157, Pro167 and the initial Ser-Gly- remaining from the His-tag. Amide $^1H^N$ and $^{15}N^H$ chemical shifts are missing for Asn72, Ser73, Leu75, Ser217, Asn121, Glu122, Ala155, Lys168 and Leu170, and the backbone carbonyl is missing for Asn72, Asp91, Asn121, Ile183, His225 and Glu226. Residues His225, Glu226, Asp227, Leu232 and Tyr236 lack an assigned $^{13}C^\alpha$, and residues His225, Glu226, Asp227, Ala228 and Lys233 lack an assigned $^{13}C^\beta$. Chemical shift assignments for PSD-95 PDZ1-PDZ2 were deposited in the Biological Magnetic Resonance Data Bank (BMRB) as entry 27308. An annotated $^{15}N$-HSQC is included in Supplementary Fig. 9.

**Chemical shift assignment for PSD-95 bound to Xph15 or Xph20.** Titration of [$^{15}N$]PSD-95−12 with 1.2 molar equivalents of natural abundance Xph15 or Xph20 resulted in large changes for numerous crosspeaks in the slow exchange regime, such that it was not possible to unambiguously assign the bound forms from the 2D spectra. However, it was clear that crosspeaks corresponding to PDZ2 were unaffected by the addition of Xph15 or Xph20. To simplify the assignment process, backbone assignments were conducted first with the single PDZ domain in PSD-95-1 bound to Xph15 or Xph20. Assignment used a combination of 3D HNCA and 3D HNCO spectra and using the $^{13}C^\alpha$ and $^{13}C'$ chemical shifts of the free PSD-95-1 for comparison. Assignments of the bound form of PDZ1 were then used to assign the bound from of the full tandem domain, with assignments confirmed by comparing the 3D HNCA and 3D HNCO spectra of the bound form of PSD-95-12 with those of PSD-95-12.

**Chemical shift assignment for PSD-95 bound to Xph18.** Titration of [$^{15}N$]PSD-95-12 with 1.2 molar equivalents of natural abundance Xph18 resulted in numerous changes to the $^{15}N$-HSQC spectrum but with poor signal-to-noise. It was not possible to assign all crosspeaks in the bound form due to the size of the chemical shift changes coupled with a slow exchange regime. In order to circumvent the poor sensitivity of the spectra, a sample of perdeuterated PSD-95-12 was prepared in complex with 1.2 molar equivalents of natural abundance Xph18. Several more assignments were completed by comparison between 3D TROSY-HNCO and 3D TROSY-HNCA spectra on the complex with 3D HNCO and 3D HNCA of the free form of PSD-95-12. However, several peaks were still undetected due to peak broadening and are indicated in yellow in Fig. 3b.

**Backbone $^1H$, $^{13}C$, $^{15}N$ chemical shift assignment of Xph15.** Resonance assignment for the free form of Xph15 used a sample of 400 μM [$^{13}C$,$^{15}N$]Xph15 in PBS with 10% (v/v) $D_2O$ added for the lock. Spectra were collected at 298 K and included 2D $^{15}N$-HSQC, 3D HNCA, 3D HNCACB, 3D CBCA(CO)NH, 3D HNCO and 3D HN(CA)CO spectra. Near complete $^1H^N$, $^{15}N$, $^{13}C^\alpha$, $^{13}C^\beta$ and $^{13}C'$ backbone resonances of Xph15 have been assigned except for some nuclei within the N-terminal six amino acids, residues of the AB loop, and the C-terminal deca-histidine tag. It was clear during the assignment process that several residues are

represented by two distinct amide crosspeaks of equal intensity. The two equal populations likely arise from two slowly exchanging populations. Due to their equal intensity, it is not possible to unambiguously assign resonances to a specific conformation. The annotated $^{15}N$-HSQC is included in Supplementary Fig. 19, and the identity of residues that display two conformations are indicated.

In contrast to the free form, addition of 1.2 molar equivalents of natural abundance PSD-95-1 results in only one set of crosspeaks for Xph15, corresponding to a single conformation. Due to the extensive chemical shift perturbation of nearly all residues, the bound form of Xph15 also required complete chemical shift assignment. NMR spectra for assignment included 2D $^{15}N$-TROSY, 3D TROSY-HNCA, 3D TROSY-HN(CO)CA, 3D TROSY-HNCACB, 3D TROSY-HN(CO)CACB, 3D TROSY-HNCO and 3D TROSY-HN(CA)CO. As in the free form, backbone assignment is nearly complete except for some nuclei within the N-terminal six amino acids, residues of the AB loop, and the C-terminal decahistidine tag. The annotated $^{15}N$-TROSY is included in Supplementary Fig. 17.

**Backbone $^1H$, $^{13}C$, $^{15}N$ chemical shift assignment of Xph20.** Resonance assignment for the free form of Xph20 used a sample of 480 μM [$^{13}C$,$^{15}N$]Xph20 in PBS with 10 % (v/v) $D_2O$ added for the lock. Spectra were collected at 298 K and included 2D $^{15}N$-HSQC, 3D HNCA, 3D HNCACB, 3D CBCA(CO)NH, 3D HNCO, 3D HN(CA)CO and 3D HNHA spectra. Near complete $^1H^N$, $^{15}N$, $^{13}C^\alpha$, $^{13}C^\beta$ and $^{13}C'$ backbone resonances of Xph20 have been assigned except for some nuclei within the N-terminal six amino acids, residues of the AB loop, and the C-terminal deca-histidine tag. Most $^1H^\alpha$ resonances have also been assigned. It was clear during the assignment process that several residues are represented by two distinct amide crosspeaks representing a major conformation and minor conformation, in a 3:1 ratio based on peak volumes. The two populations likely arise from two slowly exchanging populations. Fortunately, the unequal peak volumes allow for clear distinction for resonances from each population. The annotated $^{15}N$-HSQC is included in Supplementary Fig. 20, and the identity of residues that display two conformations are indicated.

Addition of 1.2 molar equivalents of natural abundance PSD-95-1 results in only one set of crosspeaks for Xph20, corresponding to a single conformation. Due to the extensive chemical shift perturbation of nearly all residues, the bound form of Xph20 also required complete chemical shift assignment. NMR spectra for assignment included 2D $^{15}N$-TROSY, 3D TROSY-HNCA, 3D TROSY-HN(CO)CACB, 3D TROSY-HNCO and 3D TROSY-HN(CA)CO. As in the free form, backbone assignment is nearly complete except for some nuclei within the N-terminal six amino acids, residues of the AB loop, the C-terminal decahistidine tag, and also nuclei within Trp25, Ala27, Val34-Tyr37, Ala62, Ser84 and Ser91. The annotated $^{15}N$-TROSY is included in Supplementary Fig. 18.

**Chemical shift perturbation analyses.** Combined chemical shift perturbations ($\Delta\delta_{N,H}$) were calculated from the unbound and bound 2D $^{15}N$-HSQC spectra using the following equation:

$$\Delta\delta(N, H) = \sqrt{\left\{ \left[ 0.14 \left( \Delta\delta_{15_N} \right)^2 \right] + \left[ \left( \Delta\delta_{1_{HN}} \right)^2 \right] \right\}} \tag{1}$$

where $\Delta\delta_{15N}$ and $\Delta\delta_{1HN}$ are the changes in backbone amide chemical shifts for $^1H^N$ and $^{15}N$, respectively, between the free and bound form of the protein.

**Residual dipolar coupling (RDC).** The sample contained 160 μM [70%-$^2H$,$^{15}N$] with 180 μM natural abundance Xph18 in PBS. Reference spectra to measure isotropic $^1J_{H-N}$ used interleaved spin state-selective TROSY experiments at 700 MHz and 298 K, averaged from two separate measurements. Anisotropy was introduced by using the gel-stretch method[72] by using the 6 mm to 4.2 mm gel-stretch kit from New Era Enterprises. The gel was prepared by mixing 125 μL of 40% 19:1 acrylamide:bisacrylamide with 860 μL of 5 × TBE and 0.1% (w/v) ammonium persulfate, and initiating polymerization with 0.05% (v/v) TEMED. After 1 h in the gel chamber, the gel was dialyzed for 48 h at room temperature in distilled water. The dialyzed gel was then cut into two pieces of equal length, and dried overnight on parafilm at room temperature. One of the dried gel pieces was placed in the gel chamber to which was added 500 μL of the same [70%-$^2H$,$^{15}N$] PSD-95-12:Xph18 complex used for the isotropic NMR spectra. After the sample had entered the gel over 24 h at +4 °C, the excess liquid was removed and the gel was squeezed into an NMR tube open at both ends by using the assembled gel press. The top and bottom plugs were inserted and the $^1J_{H-N}$ + $^1D_{H-N}$ values were measured by again using interleaved spin state-selective TROSY experiments, averaged from two separate measurements. Subtraction of average $^1J_{H-N}$ from average $^1J_{H-N}$ + $^1D_{H-N}$ yielded $^1D_{H-N}$ values for 60 residues (24 from PDZ1, 36 from PDZ2).

**Docking-based models.** Models were generated by using HADDOCK[73] as implemented in the HADDOCK2.2 webserver[74] made available by WeNMR[75]. For Xph15, an homology model was first generated by using SWISS-MODEL with the template PDB entry 3RZW, and the PDZ1 domain of PSD-95 was taken from the PDB entry 3GSL. The HADDOCK protocol used the WeNMR GRID-enabled docking server with the Easy interface and 1000 calculated structures. Active

residues were identified by chemical shift perturbation analysis (Supplementary Figs. 14 and 17) and predicted solvent accessibility, with passive residues automatically generated. Active residues included residues 28–31, 33, 55, 56, 58 and 79–82 from Xph15, and residues 62–65, 67, 118, 119, 121, 124, 145, 147 and 149 from PSD-95–1. From the 1000 calculated structures, 200 structures were further refined in explicit water, and 186 of these final structures (93.0%) were clustered into six clusters, from which the top cluster based on Z-score (−1.8) was taken as the predicted ensemble. For Xph20, the protocol was similar and used an homology model for Xph20 based on PDB entry 3K2M. Active residues were again selected based on chemical shift perturbation (Supplementary Figs. 15 and 18) coupled with solvent accessibility. Active residues on Xph20 included residues 29–31, 33, 57, 58, 60, 81 and 83–88, and active residues from PSD-95-1 included 62–65, 67, 118, 119, 121, 124, 126, 143, 145, 147, 149 and 150. From the 200 water-refined structures, 174 (87.0%) were clustered into 11 clusters, with the top cluster selected as the predicted ensemble (Z-score of −1.7). For Xph18, the homology model was based on PDB entry 3RZW but the model calculation strategy was carried out in two steps. Unlike Xph15 and Xph20, initial chemical shift perturbation studies using Xph18 with isolated PSD-95-1 and PSD-95-2 did not reveal binding to either domain. Upon repeating the study with doubled protein concentrations, numerous $^1$H,$^{15}$N crosspeaks for PSD-95-1 displayed perturbation or peak broadening, with still no observed binding to PDZ2 (see Supplementary Fig. 16). Therefore, an initial HADDOCK docking used Xph18 and PDZ1, with active residues for PSD-95-1 including residues 61, 63, 65, 67, 90, 91, 119, 124, 127, 145, 147, 149 and 151. Active residues for Xph18 comprised solvent accessible residues from the BC, DE and FG loops (residues 28, 29, 30, 31, 32, 33, 34, 35, 57, 58, 59, 80, 81, 82, 83, 48, 85, 86, 87, 88 and 89). The main cluster represented 123 of the 200 calculated models, with a HADDOCK score of −115.9 and a Z-score of −2.5. The lowest energy model from this cluster was then used to guide structure calculation within ARIA2.3/CNS1.2 for the complete PSD-95-12:Xph18 complex, by maintaining the structure of Xph18:PDZ1 complex with synthetic NOE distances derived from the HADDOCK model. Additional NOE distances and dihedrals were obtained for all non-hydrogen atoms in PDB entry 3GSL, while leaving linker residues 151 to 155 free of any restraints. HADDOCK-type restraints were included for contacts between PDZ2 and Xph18. Active residues for PDZ2 included 155, 156, 157, 15, 159, 211, 213, 214, 216, 218, 220, 240, 242 and 244, with no specification for a particular region on Xph18. Finally, 60 RDC values for the Xph18-bound conformation of PSD-95-12 were included, with rhombicity of 0.4 and magnitude −16. The lowest energy structures formed a cluster which was taken as the final ensemble of the Xph18:PSD-95-12 complex. HADDOCK docking statistics also appear in Supplementary Fig. 22.

**Surface plasmon resonance.** SPR measurements were performed on a BIAcore X100 or a BIAcore T200 instrument with analysis temperature set to 25 °C. CAP Sensor chips (GE Healthcare) that allow the reversible capture of biotinylated ligands as an immobilization system were used. During the experiments, reagents were kept at room temperature or at 6–10 °C for the X100 and T200, respectively. Two experimental configurations were used for the study of Xph clones, either immobilized biotinylated Xph clones to monitor the interaction with tandem PDZ domains or the opposite. Immobilization levels were optimized to reflect a compromise between minimal surface density of the ligand to avoid rebinding effects and generation of exploitable sensorgrams when possible.

Sensorgrams were collected using single cycle kinetics as Xph clones or tandem PDZ domains were injected at various concentrations (using two-fold dilutions unless otherwise stated) in PBS containing 500 mM NaCl (pH 7.4), supplemented with 0.1% Tween 20 at a flow rate of 30 μL min$^{-1}$. For T200 experiments, contact time was set to 120 s followed by a dissociation phase of 70 s. After the last injection was performed, the dissociation time was extended to 600 s. For X100 experiments, settings only differed for the dissociation phase (170 s). Regeneration of the functionalized surface was achieved with the standard regeneration solution supplied with the Biotin CAPture kit. All sensorgrams were double referenced prior to analysis by first subtracting data from a reference flow cell functionalized with the capture reagent but on which no ligand was attached and then subtracting a blank cycle where buffer was injected instead of the protein sample. The kinetic data were analysed using a 1:1 Langmuir binding model of the BIAevaluation software with the bulk refraction index (RI) kept constant and equal to 0 and the mass transfer constant ($t_c$) kept constant and equal to $10^8$. Affinity from steady-state (equilibrium) was obtained by fitting a plot of the response at equilibrium against the concentration with the one site binding (hyperbola) equation from GraphPad Prism 7.04.

**Isothermal titration calorimetry.** ITC experiments were performed in PBS 1× buffer pH 7.4 for Xph15 and Xph20 and in PBS 1× pH 7.4 with 0.01% Tween 20 using a MicroCal iTC200 (GE Healthcare). The titration sequence included a single 0.5-μL injection followed by 29 injections, 1 μL each, with a spacing of 180 s between the injections.

Concentrations of the protein used for the titrations (Xph) were in the range of 18–40 μM, while the ligand in the syringe (PSD-95-12) was in the range of 0.25–0.44 mM. All experiments were performed at 25 °C. As a blank, an independent experiment with only buffer in the calorimeter's cell was performed with the same ligand solution of PSD-95-12 in the syringe to determine the

corresponding heats of dilution. The experimental thermograms were baseline corrected, blank subtracted and the peaks were integrated to determine the binding heats produced by each ligand injection with the OriginLab Software (OriginLab, Northampton, MA). Finally, each heat was normalized per mole increase in the total ligand concentration. The resulting binding isotherm was fitted using an independent binding model of $n$ independent and identical sites, allowing the determination of the dissociation constant, $K_D$, and the binding stoichiometry, $N$, for each interaction. Note that the deviation from an ideal value of 1 for $N$, especially for Xph18, likely reflects a limited ability for the purified Xph to oligomerize and exist in two structural populations, as also noted during NMR spectroscopy on the isolated Xph samples.

**Peptide synthesis.** Peptides were synthesized at 0.05 mmol scale. Amino acids were assembled by automated solid-phase peptide synthesis on a CEM μwaves Liberty-1 synthesizer (Saclay, France) following standard coupling protocols. The divalent ligand [Stg$_{15}$]$_2$ was obtained by using copper-catalyzed click chemistry on resin harbouring a mix of sequences functionalized by azide and alkyne groups as described previously[25]. Briefly, a 7:3.5 mixture of Fmoc-Lys(N$_3$)-OH and pentynoic acid was manually coupled to the deprotected N-terminal amino group of elongated peptides on resin followed by copper(I)-catalyzed azide-alkyne cycloaddition in DMF/4-methylpiperidine (8:2) with CuI (5 eq), ascorbic acid (10 eq) and aminoguanidine (10 eq). N-free peptide resins were derivatized with acetyl groups or biotin. Peptides were purified by RP-HPLC with a semi-preparative column (YMC C$_{18}$, ODS-A 5/120, 250 × 20 mm) and characterized by analytical RP-HPLC and MALDI-TOF. Peptides were lyophilized and stored at −80 °C until usage.

**Fluorescence polarization.** For direct titrations, the stargazin peptides (monovalent and divalent) were coupled to fluorescein as previously described[39]. Briefly, following the synthesis of the peptides, the N-terminal amino group was manually and sequentially coupled to a PEG linker (Fmoc-TTDS-OH, 19 atoms, Iris Biotech, FAA1568) and fluorescein isothiocyanate. The peptides were purified by RP-HPLC and lyophilized until usage. The fluorescein-labelled stargazin peptide (50 nM) was titrated against a range of increasing concentrations of the different recombinant PDZ domains in a 100-μL final volume. Fluorescence polarization was measured in millipolarization units (mP) at an excitation wavelength of 485 ± 5 nm and an emission wavelength of 520 ± 5 nm using a POLARstar Omega (BMG Labtech) microplate reader. Titrations were conducted at least in duplicate and measured twice. To determine the corresponding affinities (apparent $K_D$), curves were fitted using a nonlinear regression fit formula[76] with GraphPad Prism v7.04 after normalizing the values of each protein series between the initial unbound and the saturating states.

For competitive titrations, experiments were designed such that the starting polarization value represent 75% of the maximal shift of the direct titrations. For the divalent stargazin ligand, PSD-95-12, SAP97-12, PSD-93-12 and SAP102-12 were used at respective concentrations of 206, 800, 118 and 982 nM. Tandem PDZ domains, bound to the fluorescein-labelled stargazin divalent peptide (50 nM), were titrated against a range of increasing concentrations of acetylated stargazin divalent ligand in a 100-μL final volume. For the monovalent ligands, the –ETWV peptide was synthesized as a biotinylated peptide by Proteogenix and the acetylated Stargazin peptide was synthesized as previously described[25]. PSD95-2 (at a concentration of 24 μM), bound to the fluorescein-labelled stargazin monovalent peptide (50 nM), was titrated against a range of increasing concentrations of ETWV or Stargazin peptides in a 100-μL final volume. Titrations were conducted as above at least in duplicate and measured twice. To determine the corresponding inhibition constant ($K_I$), curves were fitted using a competition formula[77] with GraphPad Prism v7.04 after normalizing the values of each protein series between the initial unbound and the saturating states.

**Competitive pull-down assay.** The competitive pull-down assays and control experiments were performed using a KingFisher Duo system in 96 deep-well plates (see Supplementary Note 3 for a detailed protocol). Streptavidin-coated magnetic beads (50 μL of Dynabeads M-280 Streptavidin per well) were first equilibrated in PT buffer with 0.5% BSA (1 × PBS with 500 mM NaCl, pH 7.4 + 0.1% Tween 20) and incubated with 120 pmol of biotinylated [Stg$_{15}$]$_2$ or biotin in PT buffer. The beads were washed with PT buffer with 0.5% BSA and incubated next with 180 pmol of purified tandem PDZ domains. The tandem PDZ domains-ligands complex on the beads were next titrated with various molar ratio of the competitor (Xph20 or Xph0 fused to the ETWV motif) in PT buffer with 0.5% BSA. The beads were washed in PT buffer and transferred to 2 × sample loading buffer. The samples were heated for 5 min at 75 °C and ran on a 4–20% Bis-Tris precast gel. The gels were stained with colloidal blue and imaged with a ChemiDoc XRS + imager (Bio-Rad). Uncropped gels are provided in the Supplementary Information. The gels were then analysed with the ImageLab software (lanes and bands detections) and the intensity of the band corresponding to the tandem PDZ domains without competitor was taken as the 100% intensity. The relative intensities were reported in GraphPad Prism 7.02 software in function of the logarithm of the competitor concentration and the data were fitted to the competition binding equation "One site – Fit Ki" with constrained values for the concentration of labelled ligand (tandem PDZ domains, in nanomolar) and for the equilibrium

dissociation constant of the labelled ligand ($K_I$ for $[Stg_{15}]_2$ determined by fluorescence polarization in nanomolar).

**Pull-down on brain lysates.** Adult rat brain lysates were produced as previously described[25]. Brains were obtained from adult (2–3 months old) Sprague–Dawley rats raised in the animal facility of Bordeaux University B 33 063 917. Animals were killed by decapitation after isoflurane anaesthesia (5%, 3 min), in accordance with the European 2010/63/EU directive and approved by the Bordeaux University Ethics Committee (CE50). Briefly, frozen brains (2 × ~1.5 g) were thawed in 20 mL of ice-cold modified RIPA buffer (50 mM Tris pH 7.5, 150 mM NaCl, 0.1% SDS, 0.5% sodium deoxycholate, 1% NP-40, 1 mM EDTA) containing a protease inhibitor mixture (1:1000; Protease Inhibitor Cocktail set III; Calbiochem) for about 5 min and cut into small pieces. The tissues were homogenized using a glass/teflon homogenizer. Homogenates were centrifuged at $7500 \times g$ for 25 min at 4 °C to remove cell debris. The supernatant was aliquoted and stored at −80 °C until the affinity-based isolation (pull-down) experiments were performed.

For Western blot analysis, the pull-down assay was performed using a KingFisher Duo system in 96-deep-well plates. Streptavidin-coated magnetic beads (50 μL of Dynabeads M-280 Streptavidin per well) were first equilibrated in RIPA buffer with 0.5% BSA. In the meantime, 150 pmol of biotinylated Xph20-ETWV (or biotinylated mScarlet-I as a control) were incubated with 400 μL of brain lysate in RIPA-BSA buffer for 10 min at room temperature. Beads were then incubated with the Xph20-ETWV–lysate pre-mix and were washed twice with 1 mL of RIPA-BSA and twice with 1 mL of RIPA without BSA and finally transferred to 2 × sample loading buffer. The samples were heated for 5 min at 75 °C. Western blot analyses were performed as described for the anti-His coIP.

For mass-spectrometry analysis, the pull-down assay was performed using a KingFisher Duo system in 96 deep-well plates. As a preclearing step, brain lysate (200 μL per point) was first incubated with 50 μL of streptavidin-coated magnetic beads at room temperature in 1.5 mL low-binding tubes. After 10 min of incubation, the magnetic beads were captured and the cleared brain lysate supernatant was recovered and placed in new low binding tubes into which biotinylated proteins were added at a concentration of 60 nM (12 pmol of b-mScarlet, bXph20-ETWV and bXph0-ETWV). During the 10 min incubation of the lysate and the biotinylated proteins, 40 μL of Dynabeads M-280 Streptavidin were equilibrated in RIPA buffer with 0.5% BSA. Beads were then incubated with the proteins–lysate pre-mix and were washed three times with 1 mL of RIPA–BSA and nine times with 1 mL of RIPA without BSA and finally transferred to 2 × sample loading buffer. The samples were heated for 5 min at 75 °C.

**Immunoprecipitation on hippocampal neuron culture.** For IP experiments, hippocampal neurons (E18) were plated at a density of 500,000 cells/well in 6-well culture plates in Neurobasal medium supplemented with SM1 supplement (Stemcell), 2 mM Glutamine and 10% horse serum. At 3 DIV, the horse serum was removed and 5 μM Ara-C (Sigma-Aldrich) was added. From 6 DIV onwards, the culture medium was exchanged every 3 days by replacing half of the supplemented neurobasal medium by BrainPhys neuronal medium (Stemcell).

At 13–14 DIV, neurons were incubated for 45 min with or without 1.6 μM of TAT-Xph20-ETWV in 700 μL of maintenance medium per well. After two washes with cold tyrode's buffer (20 mM Glucose, 20 mM HEPES, 120 mM NaCl, 3.5 mM KCl, 2 mM MgCl₂, 2 mM CaCl₂), neurons were lysed with 350 μL of lysis buffer (HEPES 25 mM, NaCl 125 mM, 1% NP40, 1 × protease inhibitor cocktail (Calbiochem) and 1 × Halt Phosphatase Inhibitor Cocktail (Thermo Fisher Scientific)) per well during 20 min on ice. Neuron lysates were collected, homogenized with a micropestle and spun down at $15,000 \times g$ for 5 min. Protein concentration of each lysate was quantified using BCA reagent (Thermo Fisher Scientific).

For anti-histidine-tag co-IP, 200 μg of proteins per condition were incubated overnight at 4 °C with 20 μL of Dynabeads protein-A (Thermo Fisher Scientific, 10002D) pre-coated with 2 μg of anti-His antibody (Abcam, ab18184) during 1 h at room temperature. The following day, the co-IPs were washed three times with lysis buffer containing 0.01% Tween 20 and eluted in SDS-PAGE loading buffer. For anti-PSD-95 co-IP, the procedure was the same except for the use of Dynabeads protein-A pre-coated with 6 μL of rabbit anti-N-ter PSD-95 antibody (Cell Signalling, 2507).

For the anti-His-tag coIP, the starting material (25 μg) and all Co-IP elution were separated by SDS-PAGE and transferred to a nitrocellulose membrane. The dried membrane was blocked with Odyssey blocking buffer cut horizontally around the 50 kDa ladder band). The upper part was incubated overnight with a mouse anti-PSD-95 (Millipore-Merck, MAB1596) 1/1000 and bottom part with a mouse anti-Histidine tag (Sigma-Aldrich, H1029) 1/1000. Following incubation with secondary antibodies, respectively, anti-mouse IRDye-680LT (LI-COR, 926–68020) (1/15,000) and anti-mouse IgG, Fcγ fragment specific Alexa Fluor 790 (Jackson ImmunoResearch, 115-655-071) 1/15,000; blots were washed, dried and imaged using an Odyssey Imaging System.

For the anti-PSD-95 coIP, 10 μg of the starting material and all Co-IP elution were loaded on SDS-PAGE, transferred to nitrocellulose membrane. Dried membrane was blocked with Odyssey blocking buffer and incubated overnight with a mouse anti-PSD-95 (Millipore-Merck, MAB1596) 1/1000 and with a mouse anti-Histidine tag (Sigma-Aldrich, H1029) 1/1000. Following incubation with secondary

antibody anti-mouse light chain IRDye-800 (Jackson ImmunoResearch, 115-655-174) (1/15,000), the blots were washed, dried and imaged using an Odyssey Imaging System.

Analysis was done using the Odyssey Image Studio Lite Ver5.2 software and quantification of the co-immunoprecipitated Xph20-ETWV fraction normalized to the IP fraction of PSD-95 on the same lane was performed using the average intensity of each single band on linear range values.

**Photo-crosslinking experiments.** For determining the selectivity, photo-crosslinking reactions were performed in PBS buffer at room temperature. Xph20-ETWV* (15 μM) was mixed with one, two or three tandem PDZ domains protein (s) in a large excess (45 μM each) in a final volume of 20 μL, in 0.2-mL clear polypropylene PCR tubes (Bio-Rad). Long-wave UV irradiation ($\lambda = 365$ nm) was performed in a Vilber Lourmat™ Biolink™ BLX UV Crosslinker for 10 min (0.120 J cm⁻²). Samples were analysed by SDS-PAGE using 4–20% gradient Miniprotean TGX Precast gels (Bio-Rad) and colloidal blue staining. The gel image was obtained with a ChemiDoc XRS + imager (Bio-Rad).

For proteomic analysis of the region of interaction of the binding motif, photo-crosslinking reactions were performed in PBS buffer at room temperature. Each competitor (15 μM of Xph20-Stg*, –ETWV*, or -ETMA* fusion, 100-fold excess compared to $K_D$) were mixed with PSD-95 tandem PDZ domains (20 μM) in a final volume of 20 μL, in 0.2-mL clear polypropylene PCR tubes (Bio-Rad). Samples UV irradiation and SDS-PAGE analysis were conducted as described above.

For proteomic analysis of the selectivity, photo-crosslinking reactions were performed in PBS buffer at room temperature. Xph20-ETWV* at a final concentration of 10 μM was mixed with the four DLG tandems (20 μM each) in a final volume of 20 μL, in 0.2-mL clear polypropylene PCR tubes (Bio-Rad). Samples UV irradiation and SDS-PAGE analysis were conducted as described above.

**Proteomic analyses.** Protein sample were solubilized in Laemlli buffer and were deposited onto SDS-PAGE. After colloidal blue staining, bands were cut out from the gel subsequently cut in 1 mm × 1 mm pieces. Gel pieces were destained in 25 mM ammonium bicarbonate 50% acetonitrile (ACN), rinsed twice in ultrapure water and shrunk in ACN for 10 min. After ACN removal, gel pieces were dried at room temperature, covered with the trypsin solution (10 ng μL⁻¹ in 50 mM NH₄HCO₃), rehydrated at 4 °C for 10 min, and finally incubated overnight at 37 °C. Spots were then incubated for 15 min in 50 mM NH₄HCO₃ at room temperature with rotary shaking. The supernatant was collected, and an H₂O/ACN/HCOOH (47.5:47.5:5) extraction solution was added onto gel slices for 15 min. The extraction step was repeated twice. Supernatants were pooled and concentrated in a vacuum centrifuge to a final volume of 30 μL. Digests were finally acidified by addition of 1.2 μL of formic acid (5%, v/v) and stored at −20 °C.

Peptide mixture was analysed on an Ultimate 3000 nanoLC system (Dionex, Amsterdam, The Netherlands) coupled to an Electrospray mass spectrometer.

For the analysis of pull-down material, 10 μL of peptide digests were loaded onto a 300-μm-inner diameter × 5-mm C₁₈ PepMapTM trap column (LC Packings) at a flow rate of 10 μL min⁻¹. The peptides were eluted from the trap column onto an analytical 75-mm-inner diameter × 50-cm C₁₈ Pep-Map column (LC Packings) with a 4–40% linear gradient of solvent B in 105 min (solvent A was 0.1% formic acid and solvent B was 0.1% formic acid in 80% ACN). The separation flow rate was set at 300 nL min⁻¹. The mass spectrometer operated in positive ion mode at a 1.8-kV needle voltage. Data were acquired on an Orbitrap Fusion™ Lumos™ Tribrid™ mass spectrometer (Thermo Fisher Scientific, San Jose, CA) using Xcalibur 4.1 software in a data-dependent mode. MS scans ($m/z$ 375–1500) were recorded at a resolution of $R = 120,000$ (at $m/z$ 200) and an AGC target of $4 \times 10^5$ ions collected within 50 ms. Dynamic exclusion was set to 60 s and top speed fragmentation in HCD mode was performed over a 3-s cycle. MS/MS scans with a target value of $3 \times 10^3$ ions were collected in the ion trap with a maximum fill time of 300 ms. Additionally, only +2 to +7 charged ions were selected for fragmentation. Others settings were as follows: no sheath nor auxiliary gas flow, heated capillary temperature, 275 °C; normalized HCD collision energy of 30% and an isolation width of 1.6 $m/z$. Monoisotopic precursor selection (MIPS) was set to Peptide and an intensity threshold was set to $5 \times 10^3$.

For photocrosslinking experiments, 10 μL of peptide digests were loaded onto a 300-μm-inner diameter × 5-mm C₁₈ PepMapTM trap column (LC Packings) at a flow rate of 30 μL min⁻¹. The peptides were eluted from the trap column onto an analytical 75-mm-inner diameter × 25-cm C₁₈ Pep-Map column (LC Packings) with a 4–40% linear gradient of solvent B in 108 min (solvent A was 0.1% formic acid in 5% ACN, and solvent B was 0.1% formic acid in 80% ACN). The separation flow rate was set at 300 nL min⁻¹. The mass spectrometer operated in positive ion mode at a 1.8-kV needle voltage. Data were acquired on an Electrospray Q-Exactive quadrupole Orbitrap benchtop mass spectrometer (Thermo Fisher Scientific, San Jose, CA) using Xcalibur 2.2 software in a data-dependent mode. MS scans ($m/z$ 350–1600) were recorded at a resolution of $R = 70,000$ (at $m/z$ 200) and an AGC target of $3 \times 10^6$ ions collected within 100 ms. Dynamic exclusion was set to 30 s and top 12 ions were selected from fragmentation in HCD mode. MS/MS scans with a target value of $1 \times 10^5$ ions were collected with a maximum fill time of 100 ms and a resolution of $R = 17,500$. Additionally, ions with 2–7 charges were selected for fragmentation. Others settings were as follows: no sheath nor auxiliary

gas flow, heated capillary temperature, 250 °C; normalized HCD collision energy of 27% and an isolation width of 2 $m/z$.

For the pull-down material analysis, data were searched by SEQUEST through Proteome Discoverer 2.3 (Thermo Fisher Scientific Inc.) against three bait protein sequences imbedded in the Rattus norvegicus Reference Proteome Set (from Uniprot 2019-05; 29938 entries). Spectra from peptides higher than 5000 Da or lower than 350 Da were rejected. The search parameters were as follows: mass accuracy of the monoisotopic peptide precursor and peptide fragments was set to 10 ppm and 0.6 Da, respectively. Only b- and y-ions were considered for mass calculation. Oxidation of methionines (+16 Da) and protein N-terminal Acetylation (+42 Da) were considered as variable modifications and carbamidomethylation of cysteines (+57 Da) as fixed modification. Two missed trypsin cleavages were allowed. Peptide validation was performed using Percolator algorithm[78] and only high confidence peptides were retained corresponding to a 1% false-positive rate at peptide level. Peaks were detected and integrated using the Minora algorithm embedded in Proteome Discoverer. Proteins were quantified based on unique peptides intensities. Normalization was performed based on total protein amount. Protein ratios were calculated based protein abundancies. A statistical test (ANOVA) was calculated based protein individual values. Quantitative data were considered for proteins quantified by a minimum of two peptides, fold changes above 2 and a statistical $p$-value < 0.05.

For the photocrosslinking experiments, data were exported as mgf files, and StavroX 3.6 (Michael Götze, University of Halle-Wittenberg, Department of Biochemistry & Biotechnology, General Biochemistry)[79] was used for the detection of crosslinks between ligands incorporating pAzF and every possible amino acids in a database consisting of PDZ domains of PSD-95 embedded in ten yeast protein sequences. Stavrox analysis was performed with the following settings: proteolytic cleavage: C-terminal at Lys and Arg with 3 and 2 missed cleavages, respectively, minimum peptide length: 5, static modification: alkylation of Cys by IAA, variable modification: oxidation of M, cross-linker: AzidoPhe (-N$_2$), precursor and fragment mass accuracy: 10 ppm, signal-to-noise ratio: 2, precursor mass correction activated, prescore cut-off at 10% intensity, FDR cut-off: 5%.

**Immunoprecipitation assay.** 293T cells (ECACC-12022001) were plated at a density of 300,000 cells per well in 6-well culture plates in DMEM medium supplemented with 2 mM glutamax and 10% FBS. The day after plating, co-transfection of PSD-95 H130V, H372V eGFP [+253] and HA-stargazin-mCherry-21aa[39] and either Xph20, Xph0-ETWV or Xph20-ETWV was performed using X-treme GENE HP DNA transfection reagent (Roche) as per the manufacturer's instructions. A 1:1:5 molar ratio between PSD-95, Stargazin and the ligands was used.

After 18–20 h of expression, cells were lysed on ice with 200 µL per well of lysis buffer (125 mM NaCl, 25 mM HEPES, 1% NP40, 1 × protease inhibitor cocktail (Calbiochem) and 2 × Halt Phosphatase Inhibitor Cocktail (Thermo Fisher Scientific)). Cell lysates were collected, homogenated with a micropestle and spun down at 15,000 × g for 10 min. Proteins concentration of each lysate was quantified using BCA reagent (Thermo Fisher Scientific). For co-IP, 50 µg of protein per condition were incubated overnight at 4 °C with 20 µL of Dynabeads protein-G (Invitrogen, 10004D) pre-coated with 2 µg of rat anti-HA antibody (Sigma-Aldrich, 1 867 423) during 1 h at room temperature. The following day, the immunoprecipitations were washed three times with lysis buffer and eluted in SDS-PAGE loading buffer.

A total of 5 µg of start material and all IP elution were loaded on SDS-PAGE, transferred to nitrocellulose membrane and blocked with Odyssey blocking buffer. The membrane was incubated overnight with anti-GFP (1/5000, Abcam, ab290) and anti-mCherry (1/1000, Abcam, ab167453). Following incubation with an anti-rabbit IgG, light chain specific IRDye-800W secondary antibody (1/15,000, Jackson Immunoresearch, 211-652-171), blots were imaged using an Odyssey Imaging System. Analysis was done using the Odyssey software and quantification of the IP fraction of PSD-95 co-immunoprecipitated normalized to the IP fraction of Stargazin on the same lane was performed using the average intensity of each single band on linear range values.

**Clustering assay.** COS-7 cells were cultured in DMEM (GIBCO/BRL) supplemented with 10% FBS, 100 units mL$^{-1}$ penicillin and 100 mg mL$^{-1}$ streptomycin. Cells were electroporated with PSD-95-eGFP [+253], AP-LRRTM2[67], BirA$^{ER}$ (gift from Alice Ting, Stanford University, Addgene plasmid #20856) and either Xph20-ETWV, Xph0-ETWV or Xph20 (DNA ratio: 1, 1, 1, 3 for 1.5 million cells) with the Amaxa system for cell suspension (Lonza). Electroporated cells were plated on 18-mm glass coverslips (50,000–100,000 cells per coverslip), and observed 24 h after electroporation.

Cells expressing AP-LRRTM2 (where AP stands for biotin-acceptor peptide or AviTag) were labelled at room temperature for 5 min with monomeric streptavidin (mSA)[80] conjugated to STAR635P, rinsed twice and mounted in Tyrode solution (15 mM D-glucose, 108 mM NaCl, 5 mM KCl, 2 mM MgCl$_2$, 2 mM CaCl$_2$ and 25 mM HEPES, pH 7.4) containing 1% biotin-free BSA (Carl Roth) and placed in an open Inox observation chamber (Life Imaging Services, Basel, Switzerland). The chamber was placed on an inverted microscope (Nikon Ti-E Eclipse) equipped with a Prime CMOS Camera (Photometrics, Tucson, Arizona, USA) and a thermostatic box (Life Imaging Services) providing air at 37 °C.

Cells expressing AP-LRRTM2 were detected using mSA-STAR635P and kept on the setup in Tyrode's medium at 37 °C for <1 h. The nuclear miRFP670 was used to identify co-transfected cells and constructs expression level. Images of LRRTM2 and PSD-95 were acquired using the MetaMorph software (Molecular Devices) under the same acquisition parameters. Analysis was performed using a custom-made macro on MetaMorph. Briefly, images of AP-LRRTM2 and PSD-95-eGFP were segmented using a wavelet-based segmentation method to identify clusters. Segmented images from the AP-LRRTM2 signal were used to draw regions of LRRTM2 clusters. These regions were transferred onto the segmented images of PSD-95 to determine the percentage of LRRTM2 clusters containing PSD-95 (apposed and colocalized clusters). Datasets were analysed by a non-parametric ANOVA test followed by a Dunn's Multiple Comparison Test 2 by 2.

**Molecular graphics.** All figures depicting the protein structures were generated with PyMOL (Version 2.0.7 Schrödinger, LLC).

**Reporting summary.** Further information on research design is available in the Nature Research Reporting Summary linked to this article.

## Data availability

Backbone $^1$H, $^{13}$C and $^{15}$N chemical shift assignments for PSD-95-12, PSD-95-1 and PSD-95-2 were deposited in the Biological Magnetic Resonance Data Bank (BMRB) as entries 27308, 27309 and 27310, respectively. The mass spectrometry proteomics data have been deposited to the ProteomeXchange Consortium via the PRIDE partner repository with the dataset identifiers PXD015313 (cellular target identification) and PXD015366 (photocrosslinking). Key plasmids constructed in this study are available directly from Addgene at https://www.addgene.org/Matthieu_Sainlos/. The authors declare that the data supporting the findings of this study are available within the paper, its Supplementary Information file and Source Data file. Additional raw data and other materials are available from the corresponding authors upon reasonable request. The source data underlying Figs. 2c, d, f, 3a–c, 5a, b, 6b, c, 7c, d, and 8d–f and Supplementary Figs. 3, 5b, 14–20, 23b, 26c, 28a–d and 31c are provided as a Source Data file.

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

## Acknowledgements

This research was financially supported by grants from the Centre National de la Recherche Scientifique, the Conseil Régional de la Nouvelle Aquitaine, the National Infrastructure France BioImaging (grant ANR-10INBS-04-01), the Agence Nationale de la Recherche (CheMoPPI, ANR-13-BS07-0019-01) to C.M., C.P. and M.S., the European Research Council advanced grants nano-dyn-syn (232942) and ADOS (339541) to D.C., the Labex BRAIN (ANR-10-LABX-43) to C.R. and a Marie-Curie postdoctoral training grant to D.G. B. (neuroCHEMbiotools, FP7-PEOPLE-2010-IEF project #273817). We also thank the Biochemistry and Biophysics Core Facility of the Bordeaux Neurocampus funded by the Labex BRAIN (ANR-10-LABX-43) and J.M. Blanc and Y. Ruffin for technical assistance as well as the Structural Biophysico-Chemistry plateform (UMS3033/US001) of the Institut Européen de Chimie et Biologie (Pessac, France) for access to the T200 Biacore instrument and Laetitia Minder for technical assistance and the IINS cell culture facility and Emeline Verdier for technical assistance. Financial support from the IR-RMN-THC Fr3050 CNRS for conducting the research is gratefully acknowledged. The FP7 WeNMR (project# 261572), H2020 West-Life (project #675858) and the EOSC-hub (project #777536) European e-Infrastructure projects are acknowledged for the use of their web portals, which make use of the EGI infrastructure with the dedicated support of CESNET-MetaCloud, INFN-PADOVA, NCG-INGRID-PT, TW-NCHC, SURFsara and NIKHEF, and the additional support of the national GRID Initiatives of Belgium, France, Italy, Germany, the Netherlands, Poland, Portugal, Spain, UK, Taiwan and the US Open Science Grid.

## Authors contribution

C.R., C.D.M. and M.S. designed the research and wrote the article. C.R. generated the library and performed phage display selections and biophysical experiments with the help of C.B., C.G., I.G., S.Cr., S.A., C.T. and D.G.B. C.D.M. and K.M. designed the NMR experiments, performed all NMR experiments and analysed the data. C.B., V.P. and C.P. performed the FRET/FLIM experiments. I.C. developed and performed the cellular LRRTM2 binding assay. C.B. performed the immunoprecipitation experiments. F.W.J.T. and B.D. developed the sequencing analysis tool. D.G.B. synthesized the divalent ligands. S.Cl. performed the proteomic analysis. D.C. provided intellectual input, material and financial support. C.D.M. and M.S. coordinated and oversaw the research project. All authors discussed the results and commented on the manuscript.

## Competing interests

The authors declare no competing interests.

## Additional information

**Peer Review Information** *Nature Communications* thanks Yves Nominé, and other anonymous reviewer(s) for their contribution to the peer review of this work. Peer reviewer reports are available.

