## [Peer Review File · Nature Communications]

Reviewers' comments:

Reviewer #1 (Remarks to the Author):

The manuscript by Rimbault et al. address the very important problem of specific pharmaceutical targeting of one isoform within a family of homologous isoforms specifically the PSD-95 family of scaffolding proteins . In particular, the work focuses on PDZ domains, which have proved very challenging to target specifically due to their degenerate specificity and the high sequence conservation among PDZ domains within the PSD-95 family. The authors describe a highly novel strategy to generate a bivalent molecule containing an inhibitor fused to an evolved protein interaction domain. The manuscript contains an exceptional amount of work that is very well presented. The work is all of high technical quality. The strength of the work is the strategy of targeting regions outside the PDZ active site to achieve isoform specificity and then linking this to a canonical active site inhibitor. The extensive characterization of the inhibitor itself and the mapping of binding regions provide a detailed picture of the results of the phage selection. The main weakness of this report is that there is not enough quantitative assessment of the inhibitor both in terms of inhibition constants but also in terms of the relative specificity. The assays used to measure function are novel but perhaps not as clear cut for assessing pharmaceutical constants. My sense is that the inhibitor is not that effective or selective but still provides a great starting point for future development. The manuscript should focus mostly on the strategy and use less powerful language to describe the effectiveness unless more quantitative comparison is possible.

MAJOR POINTS

The abstract is too strongly worded given the degree to which the designed protein was able to function as an inhibitor and the degree to which selectivity of inhibition was actually realized or demonstrated. The abstract claims specific targeting of a single PDZ domain but the text as written makes it sound like the goal was inhibition of both domains but only inhibition of PDZ2 was possible with this strategy. The manuscript makes this sound like happenstance rather than specific targeting by design. Based on the results, I am still not clear how effective or specific these inhibitors are.

The crystal structure may not be representative of the conformation of the PDZS tandem in solution. There was no validation of this model, which required unusual peptide fusions to obtain crystals. NMR produced a wholly different model. The utility of the docked models based on this structure is unclear.

The Xph18 interaction modality is not clear. The discontinuous nature of the contact sites along with the statement that NMR encountered "significant difficulties due to unfavorable size and behavior of the bound complex" would suggest that the interaction is perhaps not 1:1.

What is the evidence that PDZ “domain 1 and 2 are mobile with respect to one another especially in the isolated tandem form”? The NMR by the group of M Zhang suggested a relatively fixed orientation with limited dynamics as did a fluorescence study.

The SPR experiments (e.g. shown in SI fig. 24 and 26) are carried out at concentrations well below the KD. This makes fitting difficult as the curves have not even reached the inflection point. In my experience, this makes the fitting challenging as the curves may fit equally well to a linear function. Given the high concentrations possible in NMR experiments, the rationale for not collecting more of the binding curve is not clear. Solution polarization measurements may be more amenable to the high concentrations required if SPR is a limitation.

The affinity of the inhibitors for the different isoforms along with the fitting parameters should be explicitly reported in the text. It is difficult to assess specificity without direct quantitative comparison. The text makes reference to the “measured K_i values” but I am not finding these values. Similarly, “specificity” has a formal mathematical definition based on the KDs for each isoform and the therapeutic concentration necessary for inhibition. These values should be added to the text to facilitate assessment of the magnitude of these parameters, and allow for a quantitative assessment of the claims made in the abstract regarding performance.

It is not clear what is to be taken from the differences in the magnitude of the SPR response (e.g. SI Frig 24 SAP-102). Is this taken as indicating a lower extent of binding? The curves show saturation at low inhibitor concentration suggesting a similar dissociation constant.

The inhibition assay involving a small peptide moiety binding to beads via streptavidin and biotin seems convoluted and the interpretation of the outcome is not clear to me. The peptide ligand is much smaller than the enzyme to which it is attached. Is the affinity the same for the SA bound peptide? Is it not possible to do a simple fluorescence polarization assay using labeled peptide, which would yield more traditional measures of K_i . The material remaining on beads may be adsorbed. The gel figure shows BSA and the Xph constructs are on the beads. Why would these components remain on beads if not for non-specific adsorption? Shouldn't competition allow the eluate from the beads to be assayed directly?

The reported KDs for the divalent peptides binding to PDZ tandems were ~micromolar, which is similar to the affinity for monovalent peptides. The previous literature reports on bivalent peptides binding to tandem PDZ domains have reported nanomolar affinities. What is the origin of this discrepancy? If the SGN peptide does not interact with both PDZ domains then it is not a bivalent ligand and this discussion should be revised.

The interpretation of the crosslinking results, in terms of a metric for competition, is not clear to me. To my eye, it appears from SI fig. 28 that all PDZ tandems formed the same amount of the lower electrophoretic mobility species (panel b). This suggests similar binding. The analysis of the “competition” experiment is not clear to me. Is this a qualitative assessment by eye or was some metric used? Would not mass spec be better suited to assess which peptides (from other PDZ tandems) are present in the shifted bands? Given that all PDZ tandems showed crosslinks, were the other tandem crosslinks analyzed by MS?

It is surprising that only PDZ2 is targeted by the ETWV peptide, which was chosen based on its similar affinity for both domains. Given that Xph20 binds PDZ1, this would be the closer binding site. PDZ domains are notoriously allosteric. Is it possible that the Xph20 binding is affecting PDZ1 affinity for this peptide directly? Is there some steric reason that the linker may prevent this interaction? The manuscript claims that Xph allows “non-hindrance of peptide ligand binding to PSD-95.” I am not clear as to how this was shown? Allosteric interactions on the surface of PDZ domains can have allosteric effects.

The manuscript refers to the “high efficiency of the pAzF-binding motif to generate photocrosslink adducts.” However, the SI Fig 28 visually appears to show that most of PDZ tandem is not crosslinked or shifted in mobility. How was the extent of crosslinking assessed?

It was hard to interpret the success of the cellular inhibition experiments shown in Fig 7. The constructs are far from natural involving fusion of multiple FPs and the experiment required mutations to disrupt PDZ1 and 3. The inhibitor was expressed in the cell so the effective concentration is not known. The manuscript also mentions that an alternate assay failed but the reasons are not clear. All of these caveats suggest that a very specific condition was chosen to maximize performance. This raises questions in my mind about how effective the inhibitors actually are.

The manuscript states that “Although epitopes were only mapped for three of the clones, it seems likely that given their specificity most bind to regions outside of the conserved binding grooves.” Is screening 3 isolates sufficient to make this claim? Is this statement necessary?

The discussion refers to these reagents as “imaging tools.” This is not clear to me. They may be useful modulators for many kinds of experiments but, given that they are unlabeled, how are they specifically of benefit to imaging?

The extent of the Supplementary Information is excessive for a paper of this length. There is 4 times the number of SI figures than main text figures while the only figure in the introduction is in the SI. I would suggest condensing the NMR figures similar to that shown in SI figure 8. My opinion is that SI Figs. 1, 6, 7 and 31 may not be necessary as these are not major points that affect the narrative.

MINOR POINTS

The Xph constructs are referred to as having “small size.” This seems incorrect given that they are nearly equal in size to the PDZ domains.

AP peptide used in biotin experiments is not defined.

TYPOS

P5 “All PDZ domains inhibitors that have been reported to date” should read “All PDZ domain inhibitors”

SI Fig 26 Legend: “(derived from the last 15 residues of stargazing)” should read “stargazin”

The panel numbering and legend are off for SI Fig. 2.

Reviewer #2 (Remarks to the Author):

The authors present here a strategy to block intracellular Protein-Protein interaction by combining engineered FN3 domains with peptides selective to the archetypical PDZ domain PSD95. Overall the experiments are well executed and also provide the necessary information to follow the major conclusions of the manuscript. The overall idea is an extension or new twist to the concept of affinity clamps introduced by the Koide lab in 2009 where an engineered FN3 domain is combined with a peptide recognition domain to bind selectively C-terminal peptides in cells. The authors stress in their paper that it is required to selectively target individual PDZ domains in multi-PDZ scaffold proteins to better understand their biological roles in cells.

In general, I think this is a good approach but I have several major concerns:

1. More care should be given to the fact that the selected FN3 domain are already specific to the PSD95 domain without counter selection. In practice this would allow to generate also FN3 binders that directly target the binding site of the PDZ and avoids the combination with peptides.

2. If the technology is such a great advance why didn't the authors provide engineered FN3 domains with selectivity to all three PDZs in the PSD95 assembly and show then that the strategy allows to target individual PDZ in the array. I am sure that by using phage display the authors can select in parallel several PDZ specific FN3 binders.

3. Can the authors provide a better assessment of intracellular specificity? For example, it is fairly common by others that are generating intracellular tools to check intracellular specificity by coupled co-IP mass spectrometry. Does the FN3-peptide bind endogenous PSD95 in cells?

4. It is confusing that the peptide apparently only binds to domain 2 although the domain 1 is recognized by the FN3. Is this a function of linker length? Have several linkers been tried?

5. Is there a cellular phenotype when the interaction of PSD95 is blocked that can be linked to the role of PSD95 in cells?

In general, the approach is a nice idea however I believe that it hasn't met the high standards of Nature Communication and should be published in a more specialized journal.

Reviewer #3 (Remarks to the Author):

The manuscript by Rimbault et al represents a huge piece of work. It includes a very thoughtful planning, several demanding experimental steps very carefully executed and analyzed, and rock-solid results that appear to fulfill all aspects of the planned aims. A very wide panel of techniques have been used, justifying the important list of authors that contributed different expertises. Finally, the paper is extremely well written; the reader is very well guided through the impressive amount of data.

The initial question was, how can we make selective inhibitors of close paralog proteins comprising very similar, yet not identical, peptide-binding domains? The "paralog model" is here the DLG protein family. Four very similar proteins, containing three PDZ domains, among which two are closely associated within a tandem unit. All these PDZ domains are known to bind C-terminal PDZ-Binding Motifs ("PBMs") with overlapping binding specificities. How to target one paralog protein and not the three other ones? The proposed strategy has been to build a library with a scaffold domain called "10FN3" bearing engineered variable loops. This generated a library of "10FN3bodies" which were then screened by Phage Display to identify specific binders of one particular paralog (here PSD-95). Several clones were selected, that bound selectively to PSD-95 in the submicromolar-nanomolar range while binding only weakly, if at all, to the other paralogs. The authors then built protein fusion constructs combining one of the selected binders to PBM peptides able to bind to one of the PDZ domains of PSD-95. This resulted in a bivalent protein inhibitor of PSD-95 that binds in the subnanomolar - picomolar range. The authors then developed an impressive series of approaches to characterize the thermodynamic, kinetic and structural aspects of binding, including NMR, SPR-BIAcore, ITC, cross-linking, mass spectrometry, semi-experimental structural modeling and pull-down and cellular assays. Among other findings, they could show that the "10FN3bodies" bound to regions distal from the "PBM-binding grooves of the PDZ domains. They could map these regions quite precisely thanks to NMR experiments.

Comments

1) Nomenclature.

The authors should also mention that these four proteins are also very frequently called DLG-1,2, 3 and 4. Actually, it is somehow confusing that in the manuscript the authors use the same name, i.e. PSD-95, to name both the entire four-member family and the particular member of that family that they focus on. Why not call it the DLG family rather than the "PSD-95-like" family? Then the confusion between PSD-95 as a protein or as a family would be avoided and the reader would be less confused with all these names...

2) Main data vs supplemental data.

In principle, the figures or tables in the main text should focus on those results that are most convincing, reliable and essential to the main line of the paper. Supplemental figures should rather deal with additional "details" (or repeats of an observation shown in the main figures). Furthermore, in such a solid experimental paper, any model or hypothesis figure should be rather kept for the

discussion at the end of the paper to clearly differentiate the data from hypothesis-models in the information flow.

In this regard the steps of the phage display, a very essential part of this work, might deserve to be further described in the main text, either in a figure panel or a table.

By contrast: being derived from modeling, do the molecular details shown in fig. 4 really bring much to the work, considering their uncertain nature? To take an instance, there is no doubt from the NMR data that the Phe residue is involved and that it contacts the indicated regions of the 10FN3bodies. However, the exact conformation of the Phe side chain, and its precise position relative to the bound domain, cannot be known with the atomic precision shown on the plots. Resolving experimentally the conformation of the Phe would require structure solving by crystallography or NMR. Resolving experimentally the relative positions of the two proteins in solution would also require such approaches, or (less reliably) SAXS and/or analytical ultracentrifugation. In the absence of such data, the surface imprints on the DLG tandem shown in Fig 3 are sufficient to describe all the information provided by the NMR data. This figure actually lacks the complementary plots, which would show the surface imprints of the 10FN3bodies (currently shown in suppl fig 17 and 18). Conversely, the plots shown in Fig 4 would be more appropriate in the supplementals.

It is not only that figure 4, from the "experimentally supported reality" viewpoint, may represent the weakest result of this otherwise extremely careful experimental article. It is also that it does not bring crucial information to the story. The critical information is that the phe is strongly involved. The conformation of the Phe in the complex is not a critical information. Same remark for the relative spatial positions of the two protein constructs: what matters for this story is their interfaces, not their precise relative positions.

Therefore my proposed option for these models would either to show them in the supplemental figures, or to use them in a conclusive figure supporting the discussion.

3) Bifunctional inhibitors - avidity & affinity - CeFF.

Combining two low affinity binders that interact with distinct regions of the target protein to build up an inhibitor displaying higher affinity and specificity has been used in diverse occasions in the PDZ field. The authors could cite more papers that used this strategy in the PDZ field. See for instance papers with the following PMIDS:

26014966 ; 25797137 ; 22343531.

These articles also cite, in turn, further examples of multivalent ligands displaying increased affinity and specificity.

In one of these papers (ref 26014966) a very simple mathematical formula was used to compute and discuss the "effective concentration" "Ceff" created by tethering the two sites A and B. The higher this "effective concentration", the more efficient the avidity effect created by the combination of the two binding sites on the single inhibitory molecule. Whenever one creates a bifunctional molecule (here, a 10FN3-PBM fusion) the Ceff is a useful quantitative parameter to describe the efficiency of the "tether" (here, the linker sequence) connecting the two functional moieties. It would be interesting if the authors would apply here the same formula to their own data, to compute for each Xph-PBM fusion the Ceff parameter and discuss it.

4) Dual conformations of 10FN3bodies

Page 9

"During the analysis, we also noted that there were two populations in the NMR spectra, but only for the unbound state."

That is quite interesting. Indeed the text is not 100% clear whether this is observed equally for the three constructs. Can the authors discuss this point, propose possible theoretical explanations and also the possible impact of this behavior on the the binding mode ? Indeed, if there are distinct conformations in the unbound state, the binding model may be complex. The derived values of KD values and kinetic constants are more likely to be incorrect. The best way to keep cautious is to describe the constants as "tentative values assuming a simple binding model". Then the reader is warned that these values are to be taken with caution.

5) SPR sensograms profiles

Fig 5. Maybe the authors should comment more extensively and precisely the differences of shape of the different sensorgrams. These shape differences are very striking visually, so any reader will spot them. Explaining and interpreting these shape differences in more detail would be useful (and pedagogic) for readers not experienced with SPR, and will probably also satisfy the experienced users...

6) "Natural" range of protein-protein binding affinities

Page 10

"Affinities were overall on the lower end of natural binders such as antibodies with dissociation constants ranging from 10 μ M to 100 nM and relatively fast dissociation rate constants (half-lives shorter than 1 min)".

Are antibodies the best instance of "natural binders" ?... Antibodies are binders of a very specific nature and tend to bind in the subnano-picomolar range: they belong to "another planet" as compared to the typical biological binders, as most biological interactions are generally reversible... Indeed, domain-motif interactions (such as PDZ-PBM interaction) are typically in the KD range mentioned by the authors (from 10 μ M to 100 nM with fast dissociation rate). Therefore it rather seems that "10FN3bodies" are already on the "higher end" of natural binders, if one excludes antibodies.

"Xph20 presented a faster k_a and a slower k_d , leading to nanomolar dissociation constants of 330 nM and 67 nM by SPR and ITC respectively."

330nM is rather "submicromolar" (a bit lower than 1 μ M) than nanomolar (around 1-10nM). 67 nM is intermediate. By contrast, later in the manuscript, the affinities of the Xph-ETWV fusions are described as "subnanomolar" when some of them almost approach the picomolar range. The fact, that adding appropriate PBMs to the tail of the Xph binder allows to multiply by almost 10.000 fold the affinity is impressive. An additional reason to calculate the "Ceff" for each bivalent construct as proposed before.

7) Full amino acid sequence details of main constructs

Since this article is mainly describing the interaction between protein constructs (DLG fragments and Xph derivatives) it would be very useful if the authors would provide an additional suppl table displaying the full aminoacid sequences of all DLG constructs, Xph (at least 15, 18 and 20) and Xph-PBM fusions. At the moment, to get these sequences it is necessary to look for them in different places, or sometimes try to reconstruct them from fragmental information in different parts of the ms.

In particular, concerning the two "ultimate inhibitors" Xph20-ETWV and Xph20-stg, it would be fantastic to know exactly their sequences. Luckily, I did find the sequence of Xph20-ETWV (suppl figure 27). By contrast I searched everywhere for the exact amino-acid sequence of the Xph20-stg fusion and could not find it. To understand how this construct was built I had to go to the following paragraph (mat&meth p 21-22):

"The fragment corresponding to a 26 amino acid-linker followed by Stargazin 13 last amino acids was synthesized (Genscript) and inserted into pIG-Xph15 after Xph15 sequence with a classical ligation using the XhoI restriction site. This plasmid was used to insert the other Xph sequences between the BamHI and XhoI sites. The constructs were then PCR amplified and transferred into pblG using the BamHI and BlnI sites. For the amber suppression approach, the TAG amber codon was introduced in the competitors by site-directed mutagenesis at the -5 position. An arginine residue was also introduced by site-directed mutagenesis at -11 and -12 positions for Xph20-ETWV and Xph20-Stg constructs respectively, in order to facilitate proteomics experiments (smaller resulting trypsin-digested fragments)."

Sure, one can try to reconstitute the construct using these explanations, provided that one can find the right stargazin sequence on the web, make the proper mutation at the -11 and -12 positions, reconstitute properly the sequence of the linker etc... But this is really complex when instead the authors can provide the sequence of Xph20-Stg in a common table, together with that of Xph20-ETWV, those of Xph15, 18, 20, and those of the main DLG constructs.

8) Discussion.

-Considering the huge team effort and human and material resources that have obviously been invested in this work, it might be worth discussing at least in a small paragraph the possible strategies that now may be envisioned to allow the in vivo targeting of such inhibitor in the animal's or person's body at the right localization where it should perform its inhibitory action. Indeed, in our own team we have previously generated, a recombinant bifunctional fusion protein that acted as a potent inhibitor of a viral oncoprotein. However, when trying to transfer our finding, it appeared that the targeting of a recombinant protein inhibitor in an intracellular context remains very difficult to develop. Maybe the situation is evolving and the authors are aware of innovative strategies? This would be interesting to discuss.

-Discussion page 15. "Although epitopes were only mapped for three of the clones, it seems likely that given their specificity most bind to regions outside of the conserved binding grooves.". This is a very interesting remark. It would be tempting to speculate, that the fact that PBM-binding groove is not a good "attractor" for "10FN3bodies" is related to the fact that the PBM-binding groove does not bind strongly and specifically to PBMs either. In other words, the stereochemical properties of the PBM binding groove would not make it a candidate for strong and/or specific protein-protein interactions, in general. And, by the way, this weak capacity for high-affinity and/or high binding specificity might well result from evolution and thereby respond to a particular requirement in the cell. It is well known that PDZ-PBM interactions are rather weak and promiscuous.

9) Detected typos (very few)

-Page 3

"One of its main functionS"

-page 15

"one of the main synaptic scaffold proteinS."

-Page 17 "at the price of an acceptable loss of the competitor affinity" -> "at the expense of an acceptable loss of the competitor affinity"

Reviewer #4 (Remarks to the Author):

The manuscript of Rimbault and colleagues reports on the design of specific ligands to specifically target a PDZ domain tandem of PSD95. For this purpose, the authors combined two molecules within a single bivalent chimera: one has been selected by phage display using library built with the so-called 10FN3 scaffold domain; the second molecule is a peptide containing a PDZ-binding motif capable to bind to one of the PDZ domains of PSD95.

The authors accumulated a huge amount of data using highly diverse approaches (phage display, NMR, SPR, ITC, MS, pull-down, ...) to characterize the binding of their molecules.

The manuscript shows a very beautiful and original work, is highly pleasant to read and the supplementary file section, although very large (it contains 32 suppl. figures, 6 tables and 2 notes) is highly useful for readers who are interested to go deeper in the details. Results are very interesting, not only in the PDZ topic, but also more generally in inter-domain interaction research area.

I strongly support the publication of this manuscript if the authors can address the four main aspects detailed below.

M1/ Specificity vs selectivity.

These two words are often employed in the manuscript. But it would be a plus whether the authors would define the meaning of these two words in the manuscript, specially the difference between the two. Indeed, there are several ways to define specificity or selectivity depending on the scientific context (chemistry, biology, by comparing to a few members of a family, or to the full list of the family members, ...).

M2/ Page 7: "Based on the specificity evaluation, none of the final clones clearly stood out and therefore we selected three representative clones (Xph15, Xph18 and Xph20)".

I do not fully agree with that sentence and the reasoning proposed by the authors to justify the choice of Xph20. To me, several clones clearly stand out based on the Phage-ELISA (2.c) and pull-down assays (2.d): Xph15, Xph18, Xph21 and Xph25. Indeed, more than the high intensity of the ratios calculated from the pull-down assay, I consider the absence of signal for the 3 other PDZ tandems (dark blue) compared to PSD95-12 as more important and relevant of the specificity; this way of thinking would exclude the Xph20 clone.

Furthermore, by looking at the 1H-15N HSQC (Supp. Fig. 7), I also have some concerns regarding the Xph20 clone since the superimposition of the HSQC spectra for SAP-97 shows quite significant changes as it is for PSD-95 (in both cases, blue and green peaks become visible). This observation calls into question the specificity of this clone.

Therefore, can the authors explain why they select Xph20, rather than Xph21 or Xph25?

M3/ SPR concerns

Usually SPR people decide to use single cycle experiments when they face some troubles to regenerate their surface, or when regeneration is detrimental to the ligand.

Regeneration might be an issue specially when the dissociation rate constant is very slow. However, this doesn't seem to be the case in the present work according to the KD values that the authors have reported at least at the beginning of the manuscript.

All together, these information would suggest that the authors faced some difficulties to develop this assay. In this regard, the authors should check several issues and discuss them in the manuscript:

* in figure 5a, it is striking to me that the overall shapes for Xph15 and Xph18 recorded by single cycle SPR experiment are so different when the affinities are so similar (4.3 uM compared to 2.6 uM). What does it mean?

* in the same figure 5a, the experimental plateau observed for the highest analyte concentration is significantly different from sample to sample (from ~10 RU for Xph15 to about 35 for Xph18), although the highest concentrations for Xph15 and Xph18 are the same and the affinities for Xph15 and Xph18 differ only by a factor of 2 (4.3 uM compared to 2.6 uM). Is it possible for the authors to provide some explanations?

* Rmax describes the maximum binding capacity, and can be calculated knowing the level of immobilized ligand at the surface as well as the molecular weights of analyte and ligand. It would be good to compare the theoretical maximal value with the experimental one obtained during the runs.

* Is there any explanation for the biphasic behavior observed when the ligand is one of the PDZ tandem (Supp. fig. 22), whereas a mono-exponential behavior is observed for Xph clones as a ligand (Fig. 5)?

M4/Kinetic and steady-state analysis

In link with the previous remarks, an other point would deserve to be addressed: as said before, the authors observed a mono-exponential or a biphasic behavior depending on the choice of the ligand for SPR experiment. They also observed by NMR that free Xph20 and Xph15 displayed two populations (not equally distributed between the two samples), likely arising from two slowly exchanging populations.

Despite this, they used 1:1 kinetics or 1:1 steady-state model to analyze their data, more likely because more complicated models would fit the data anyway.

Taking all these observations together, I would rather temper the analysis and the conclusions regarding numerical values of binding kinetic and affinity constants since the fitted constants are probably biased in some way.

This point should at least be discussed in the manuscript.

Next, some other Points are given below that can be easily addressed by the authors:

P1/ Sup. fig. 10:

According to the legend, crosspeaks that shift upon addition of Xph15 are annotated. However, I have the feeling that several annotations are missing (for instance the peaks at 10.3;123 ppm or 7.8;111 ppm).

P2/ An other way to represent the shift upon binding would be to plot the $\Delta\delta(H,N)$ vs. the sequence (as actually the authors did in suppl. fig. 14 to 20). It would seem appropriate to systematically provide this kind of plot for every figure displaying superimposed NMR data.

P3/By the way, it would be valuable, in order to help the reader, to superimpose on these $\Delta\delta(H,N)$ plots not only the position of the BC, DE and FG loops, but more generally information about secondary structures elements (as strands beta A to D).

P4/ PSD95-12 and addition of the different Xph by NMR (Suppl. figures S10-S12): why a surface representation to map the residues affected upon binding is not always provided as it is in Suppl. figures S14-S20?

P5/ Fig. 5: why aren't enthalpy changes (ΔH) and entropy changes ($-T\Delta S$) provided as usually done for ITC data?

Finally, a few Remarks on the text:

R1/ In the M&M section for SPR, a reference is missing for the sentence " biotinylated peptide was synthesized as previously described ".

Meanwhile, there is no information about what is attached to the reference flow cell.

R2/ To avoid any confusion between affinity (KD) and kinetic rates (kd), I would suggest to use kon and koff rather than ka and kd for kinetics association and dissociation rate constants, respectively.

R3/ Similarly to Xphxx-ETWV constructs, I guess that Xphxx_Stg corresponds to a construct containing the Xph clone followed by a linker and the 13 last residues of stargazin. However, unless I missed it, there is no clear definition of Xph-Stg in the text.

R4/ Supp. fig. 26, panel b: in order to improve the visibility of the plot showing the equilibrium analysis, I would suggest to use a log scale on X axis rather than the actual linear scale. Differences between the curves will be easier to observe.

A few Typos:

T1/ Figure 7, legend of panel d/: "minimum and maximum OF all individual data points".

T2/ Page 11: "some level(S?) of interaction is(ARE?) here observed ..."

T3/ End of page 14: "In contrast, expression of Xph20-ETWV lead(S) to more than a 50% decrease ..."

T4/ Page 17: "The fusion of moderate to weak binders (100 nM-10 uM) thus resulted in strong binders with affinities in the pM range arising from slower dissociation rate constants, Kd." Is it not kd rather than Kd? And therefore koff according to one of my previous comments.

T5/ Page 23: "Cultures were grown in sterile glass vessel(S) 5 h at 37 °C at 250 rpm(.)" The period is missing.

Point-by-point response for the reviewers

Reviewer #1 (Remarks to the Author):

The manuscript by Rimbault et al. address the very important problem of specific pharmaceutical targeting of one isoform within a family of homologous isoforms specifically the PSD-95 family of scaffolding proteins. In particular, the work focuses on PDZ domains, which have proved very challenging to target specifically due to their degenerate specificity and the high sequence conservation among PDZ domains within the PSD-95 family. The authors describe a highly novel strategy to generate a bivalent molecule containing an inhibitor fused to an evolved protein interaction domain. The manuscript contains an exceptional amount of work that is very well presented. The work is all of high technical quality. The strength of the work is the strategy of targeting regions outside the PDZ active site to achieve isoform specificity and then linking this to a canonical active site inhibitor. The extensive characterization of the inhibitor itself and the mapping of binding regions provide a detailed picture of the results of the phage selection. The main weakness of this report is that there is not enough quantitative assessment of the inhibitor both in terms of inhibition constants but also in terms of the relative specificity. The assays used to measure function are novel but perhaps not as clear cut for assessing pharmaceutical constants. My sense is that the inhibitor is not that effective or selective but still provides a great starting point for future development. The manuscript should focus mostly on the strategy and use less powerful language to describe the effectiveness unless more quantitative comparison is possible.

MAJOR POINTS

The abstract is too strongly worded given the degree to which the designed protein was able to function as an inhibitor and the degree to which selectivity of inhibition was actually realized or demonstrated. The abstract claims specific targeting of a single PDZ domain but the text as written makes it sound like the goal was inhibition of both domains but only inhibition of PDZ2 was possible with this strategy. The manuscript makes this sound like happenstance rather than specific targeting by design. Based on the results, I am still not clear how effective or specific these inhibitors are.

We have modified the abstract.

The crystal structure may not be representative of the conformation of the PDZS tandem in solution. There was no validation of this model, which required unusual peptide fusions to obtain crystals. NMR produced a wholly different model. The utility of the docked models based on this structure is unclear.

Our primary goal with the models is to provide a basis by which to judge how the binders will associate with the tandem PDZ1-PDZ2 domains from PSD-95. In addition to defining binding regions on each component, the models help to identify any potential steric concerns that could hinder the second step in our approach (i.e. the addition of appended PDZ domain peptide ligands), and to allow for distance measurement between the association interface and the canonical PDZ ligand binding site to ensure sufficient number of residues that link the appended peptide to the Xph binders. To emphasize these aspects of the models – and to avoid over-interpretation of any high-resolution details – we have provided a new main text Fig. 4 that better illustrates these models. We now highlight that both Xph15 and Xph20 bind in similar ways to only the PDZ1 domain, leaving PDZ2 free to move, and also show that the PDZ peptide binding regions are fully accessible to ligands. For Xph18, we highlight that the PDZ1 and PDZ2 domains are now locked into a complex in which PDZ1 and PDZ2 have very limited mobility. This situation is supported by the fact that Xph18 binds to regions on both PDZ1 and PDZ2, and importantly that a single main series of residual dipolar coupling data defines the interdomain orientation between PDZ1 and PDZ2. As with Xph15 and Xph20, the Xph18-bound complex also does not inhibit access to the canonical peptide binding regions on PDZ1 or PDZ2.

The Xph18 interaction modality is not clear. The discontinuous nature of the contact sites along with the statement that NMR encountered “significant difficulties due to unfavorable size and behavior of the bound complex” would suggest that the interaction is perhaps not 1:1.

The Xph18 interaction with the tandem is the most complex as it spans over 2 domains. Nevertheless, the titration data of [70%-²H, ¹⁵N]PSD-95-12 shows that the ¹H^N-¹⁵N TROSY crosspeaks with 1.5 and 2 equivalents of Xph18 show no further perturbation as compared to a 1:1 complex. As a result, the data support a 1:1 complex formed between PSD-95-12 and Xph18. In relation to the difficulties encountered with NMR spectroscopy, these mainly relate to the fact that unlike the complexes formed by Xph15 or Xph20, the complex formed with Xph18 displays reduced intermolecular mobility. For Xph15 or Xph20, the main complex is formed only with PDZ1, with the linker and PDZ2 remaining mobile. This situation is highlighted in the model presentation in the new main text Fig. 4. This arrangement allows more freedom of movement for PDZ2 and thus reduces the effective size of the complex. For Xph18, both PDZ1 and PDZ2 contribute to the complex and this three-domain complex has a larger effective size and the increased correlation time of the complex leads to subsequent reduction in NMR signal intensity. We have partially countered this effect by perdeuteration of PSD-95-12. More significant, we were unable to provide the same level of NMR characterization for either the bound or unbound Xph18 molecule itself. The free form of Xph18 starts to aggregate at levels above 100 μM such as required for chemical shift assignment experiments. This is exacerbated by the fact that unbound Xph18, as with unbound Xph15 and Xph20, exist in two conformation populations that further complicate the assignment process. Without this important reference assignment, we were also unable to assign the bound form of Xph18 with its reduced signal for the reasons described above.

What is the evidence that PDZ “domain 1 and 2 are mobile with respect to one another especially in the isolated tandem form”? The NMR by the group of M Zhang suggested a relatively fixed orientation with limited dynamics as did a fluorescence study.

Our claim that domain 1 and 2 are mobile with respect to one another results from the absence of a rigid linker between the two domains and the lack of evidence for strong interactions between the two domains that would stabilize a single conformation. Furthermore, all experimental studies (NMR, crystallography and single molecule TIRF FRET microscopy) on the tandem structure so far have provided different conformations (Long et al., J. Mol. Biol., 2003; Wang et al., J. Am. Chem. Soc., 2009; Sainlos et al., Nat. Chem. Biol., 2011; McCann et al., Structure, 2011). Overall, this indicates that the domains exhibit some mobility with respect to one another, but does not exclude the existence of more stable conformations. The most recent study on the tandem structure identified two major conformations separated by a low energy barrier that allows constant interconversion (Yanez Orozco et al. Nat Commun., 2018). We have added this reference to the text. In light of this interdomain mobility, we have decided to clarify our models by adding several representative PDZ2 orientations to the structures presented in main text Fig. 4.

The SPR experiments (e.g. shown in SI fig. 24 and 26) are carried out at concentrations well below the KD. This makes fitting difficult as the curves have not even reached the inflection point. In my experience, this makes the fitting challenging as the curves may fit equally well to a linear function. Given the high concentrations possible in NMR experiments, the rationale for not collecting more of the binding curve is not clear. Solution polarization measurements may be more amenable to the high concentrations required if SPR is a limitation.

We have repeated the titrations of SI Fig 24 (now S26) and 26 (now S28, for which we have used fluorescent polarization), for which the maximal concentrations were indeed low.

The affinity of the inhibitors for the different isoforms along with the fitting parameters should be explicitly reported in the text. It is difficult to assess specificity without direct quantitative comparison. The text makes reference to the “measured Ki values” but I am not finding these values. Similarly, “specificity” has a formal mathematical definition based on the KDs for each

isoform and the therapeutic concentration necessary for inhibition. These values should be added to the text to facilitate assessment of the magnitude of these parameters, and allow for a quantitative assessment of the claims made in the abstract regarding performance.

The “measured K_i values” appear both in Fig 6 and SI Table 2. The term “specificity” was only used for the Xph clones alone, which do not present inhibitory activity. We have clarified our use of “specificity” and “selectivity” in a supplementary note.

It is not clear what is to be taken from the differences in the magnitude of the SPR response (e.g. SI Fig 24 SAP-102). Is this taken as indicating a lower extent of binding? The curves show saturation at low inhibitor concentration suggesting a similar dissociation constant.

The experiment has been repeated at higher concentrations (now SI Fig 26).

The inhibition assay involving a small peptide moiety binding to beads via streptavidin and biotin seems convoluted and the interpretation of the outcome is not clear to me. The peptide ligand is much smaller than the enzyme to which it is attached. Is the affinity the same for the SA bound peptide? Is it not possible to do a simple fluorescence polarization assay using labeled peptide, which would yield more traditional measures of K_i . The material remaining on beads may be adsorbed. The gel figure shows BSA and the Xph constructs are on the beads. Why would these components remain on beads if not for non-specific adsorption? Shouldn't competition allow the eluate from the beads to be assayed directly?

Affinity of the peptide-based ligand on the beads to the tandems should be comparable to that measured in solution or by SPR as we use a PEG linker between the biotin and the functional part of the divalent ligand. Furthermore, the same system (streptavidin plus biotinylated ligand) was employed for the initial SPR experiments used to determine the K_D of the ligand for the tandems.

BSA was used here as a blocking agent after binding of the divalent ligand to block any surface of the beads that could lead to non-specific adsorption. This has been specified in the legend. Its presence in the gels is therefore expected. We have added controls that confirm that PDZ domains are only bound to the beads via specific binding interactions with the ligand (SI Fig 27h). Likewise, the presence of the Xph-ETWV is only occurring as a result of the tandem PDZ domain binding (SI Fig 27h). In this later case, we have not been able to fully rationalize the presence of the competitor on the beads for PSD-95 and SAP97.

Assaying the eluate should provide similar results, however it was more straightforward in our hands to assay the material left on the beads as we can achieve higher concentrations compatible with Coomassie staining and as we can demonstrate that there is no detectable non-specific binding of the species of interest.

The reported K_D s for the divalent peptides binding to PDZ tandems were ~micromolar, which is similar to the affinity for monovalent peptides. The previous literature reports on bivalent peptides binding to tandem PDZ domains have reported nanomolar affinities. What is the origin of this discrepancy? If the SGN peptide does not interact with both PDZ domains then it is not a bivalent ligand and this discussion should be revised.

As rightfully noted earlier by the same reviewer, the titrations leading to these K_D s were not optimally performed with respect to the concentration range used leading to some uncertainty on the reported values. These experiments were performed again using here a fluorescence polarization assay (SI Fig 28). Nonetheless, it is important to note that the previously reported values for the Stg divalent ligands (Sainlos et al, Nat Chem Biol, 2011) were obtained with a different technique and different protein constructs, which probably accounts for the differences. We have used numerous biophysical techniques (environment-sensitive probes, fluorescence polarization, SPR, NMR for the most part unpublished) and always observed a strong difference between monovalent and divalent ligands consistent with simultaneous binding of the two binding motifs for the divalent Stg constructs.

The interpretation of the crosslinking results, in terms of a metric for competition, is not clear to me. To my eye, it appears from SI fig. 28 that all PDZ tandems formed the same amount of the lower electrophoretic mobility species (panel b). This suggests similar binding. The analysis of the “competition” experiment is not clear to me. Is this a qualitative assessment by eye or was some metric used? Would not mass spec be better suited to assess which peptides (from other PDZ tandems) are present in the shifted bands? Given that all PDZ tandems showed crosslinks, were the other tandem crosslinks analyzed by MS?

Our initial analysis was only qualitative and based on electrophoretic mobility. In the revised version of the manuscript, we have performed a LC-MS/MS analysis (SI fig 31c).

It is surprising that only PDZ2 is targeted by the ETWV peptide, which was chosen based on its similar affinity for both domains. Given that Xph20 binds PDZ1, this would be the closer binding site. PDZ domains are notoriously allosteric. Is it possible that the Xph20 binding is affecting PDZ1 affinity for this peptide directly? Is there some steric reason that the linker may prevent this interaction? The manuscript claims that Xph allows “non-hindrance of peptide ligand binding to PSD-95.” I am not clear as to how this was shown? Allosteric interactions on the surface of PDZ domains can have allosteric effects.

The ETWV peptide was chosen from the work of Tanekian et al., which identified the sequence as the most efficient in binding each domain but not necessarily with a similar affinity (this parameter was not evaluated in the mentioned study). In our hands most peptide sequences that bind domain 2 present a weaker affinity for domain 1 (as is the case for stargazin, see Hafner et al, Neuron 2015).

In order to address these questions, we have performed two sets of additional photocrosslinking experiments (SI33 and 34). In the first, we show that Xph20-ETWV can interact with the binding groove of PDZ domain 1 when using this domain alone, which indicates that Xph20 binding is not preventing interaction of binding motifs to domain 1. In the second, we show that by replacing the binding motif with one more prone to interact with domain 1 (the C-terminus of the GluK2 subunit), we obtain here a mix of fragments from both the binding grooves of domain 1 and 2. Overall, these results suggest that the initial preferential binding to domain 2 is a result of both the relative affinity for each domain and conformation(s) of the tandem domains that would favor in our context (Xph20+linker+binding motif) interactions with domain 2.*

Finally, with respect to possible allosteric effects upon Xph20 binding to domain 1, we have performed NMR titrations of PSD-95-12 with peptide ligands in presence and absence of Xph20 and not observed any significant differences. These results will be used in another article currently in preparation.

The manuscript refers to the “high efficiency of the pAzF-binding motif to generate photocrosslink adducts.” However, the SI Fig 28 visually appears to show that most of PDZ tandem is not crosslinked or shifted in mobility. How was the extent of crosslinking assessed?

We have modified “high efficiency” by “robustness”, which is more appropriate. However, of note, the ratio of tandem PDZ domains, as described in the Material and Methods section, is at minimum threefold to avoid any interference of protein constructs in which the pAzF was not incorporated and that cannot be separated from the expected construct as the affinity tag is N-terminal. The large excess of tandem PDZ domains results in most PDZ domain (>2/3) not being crosslinked as observed on the gels.

It was hard to interpret the success of the cellular inhibition experiments shown in Fig 7. The constructs are far from natural involving fusion of multiple FPs and the experiment required mutations to disrupt PDZ1 and 3. The inhibitor was expressed in the cell so the effective concentration is not known. The manuscript also mentions that an alternate assay failed but the reasons are not clear. All of these caveats suggest that a very specific condition was

chosen to maximize performance. This raises questions in my mind about how effective the inhibitors actually are.

The aim of these experiments is to show that inhibition of PDZ domain 2 is achieved in a cellular context. As most ligands that bind domain 2 also bind domain 1 (and in the case of stargazin also domain 3), obtaining a clear output for these cellular assays implies turning off other domains (here by mutation). The alternate assays referred to the FRET competitive assays performed with stargazin and non-mutated PSD-95, in which the competing effect was masked by binding to the other domains, the limited control over the relative expression level of the 3 partners and cellular variability. Hence, we have used these mutations when we worked with the model system based on stargazin (Fig 8d and e) and introduced as a last experiment a system (Fig 8f) in which binding of the PSD-95 partner is occurring on domain 2 (and possibly also domain 1) and that did not necessitate these mutations.

The manuscript states that “Although epitopes were only mapped for three of the clones, it seems likely that given their specificity most bind to regions outside of the conserved binding grooves.” Is screening 3 isolates sufficient to make this claim? Is this statement necessary?

We have modified the sentence.

The discussion refers to these reagents as “imaging tools.” This is not clear to me. They may be useful modulators for many kinds of experiments but, given that they are unlabeled, how are they specifically of benefit to imaging?

The use of Xph clones (and not the competitors Xph-ETWV or -Stg) as imaging tools is the scope of a manuscript that will be submitted shortly after this one. The tools rely on the minimal invasiveness of Xph15/20 clones (and not the engineered competitors) and further engineering to allow direct fluorescence detection. Examples of such applications are provided in the two articles discussed in the previous sentences (Fukata et al., J Cell Biol, 2013; Gross et al., Neuron, 2013).

The extent of the Supplementary Information is excessive for a paper of this length. There is 4 times the number of SI figures than main text figures while the only figure in the introduction is in the SI. I would suggest condensing the NMR figures similar to that shown in SI figure 8. My opinion is that SI Figs. 1, 6, 7 and 31 may not be necessary as these are not major points that affect the narrative.

We have tried to reduce as much as possible the number of pages, however in the interest of optimal transparency for our data and analysis we believe that the extra supplementary information provides an important resource. In particular for the NMR spectra, the fact that a complete chemical shift assignment procedure was required for all complexes to determine and validate the chemical shift perturbation necessitates the inclusion of the fully annotated spectra. Such information would unfortunately be difficult to provide with reduced-size spectra.

MINOR POINTS

The Xph constructs are referred to as having “small size.” This seems incorrect given that they are nearly equal in size to the PDZ domains. *We were referring to antibodies, we have clarified this in the text.*

AP peptide used in biotin experiments is not defined. *The sequence is defined in the Material and Methods, in the Vectors paragraph.*

TYPOS

P5 “All PDZ domains inhibitors that have been reported to date” should read “All PDZ domain inhibitors” *Corrected*

SI Fig 26 Legend: “(derived from the last 15 residues of stargazing)” should read “stargazin”
Corrected

The panel numbering and legend are off for SI Fig. 2.
Corrected

Reviewer #2 (Remarks to the Author):

The authors present here a strategy to block intracellular Protein-Protein interaction by combining engineered FN3 domains with peptides selective to the archetypical PDZ domain PSD95. Overall the experiments are well executed and also provide the necessary information to follow the major conclusions of the manuscript. The overall idea is an extension or new twist to the concept of affinity clamps introduced by the Koide lab in 2009 where an engineered FN3 domain is combined with a peptide recognition domain to bind selectively C-terminal peptides in cells. The authors stress in their paper that is required to selectively target individual PDZ domains in multi-PDZ scaffold proteins to better understand their biological roles in cells. In general, I think this is a good approach but I have several major concerns:

1. More care should be given to the fact that the selected FN3 domain are already specific to the PSD95 domain without counter selection. In practice this would allow to generate also FN3 binders that directly target the binding site of the PDZ and avoids the combination with peptides.

We do not fully understand this remark. Clones that rely on the PDZ domain binding sites would indeed prevent the use of peptide but would bind to conserved PDZ domain residues and therefore suffer from poor selectivity if any. This is something that we have observed experimentally with more recent experiments.

2. If the technology is such a great advance why didn't the authors provide engineered FN3 domains with selectivity to all three PDZs in the PSD95 assembly and show then that the strategy allows to target individual PDZ in the array. I am sure that by using phage display the authors can select in parallel several PDZ specific FN3 binders.

Selections can indeed easily be done in parallel however proper characterization of the resulting clones still remains a sizeable amount of work that cannot be easily performed in parallel. Our choice here has been to focus on the in-depth characterization, validation and presentation of the approach rather than immediately searching to target various PDZ domains at once. Implementation of the approach to other domains is underway.

3. Can the authors provide a better assessment of intracellular specificity? For example, it is fairly common by others that are generating intracellular tools to check intracellular specificity by coupled co-IP mass spectrometry. Does the FN3-peptide bind endogenous PSD95 in cells?

Xph clones by themselves do bind endogenous PSD-95 in neuronal cells (experimental evidence of this property is part of a manuscript in preparation focused on their use as imaging tools). For the Xph20-ETWV construct, we have added pull-down and co-IP experiments that demonstrate the binding to endogenous PSD-95. A set of pull-down experiments were performed on a rat brain lysate using a biotinylated Xph20-ETWV, while another set of co-IP experiments were performed on primary rat hippocampal neuron culture with a TAT-derivative of Xph20-ETWV.

4. It is confusing that the peptide apparently only binds to domain 2 although the domain 1 is recognized by the FN3. Is this a function of linker length? Have several linkers been tried?

In order to address these questions, we have performed two sets of additional photocrosslinking experiments (SI33 and 34). In the first, we show Xph20-ETWV can interact with the binding groove of PDZ domain 1 when using this domain alone, which indicates that Xph20 binding is not preventing interaction of binding motifs to domain 1. In the second, we show that by replacing the binding motif with one more prone to interact with domain 1 (the C-terminus of the GluK2 subunit), we obtain here a mix of fragments from both the binding grooves of domain 1 and 2. Overall, these results suggest that the initial preferential binding to domain 2 is a result of both the relative affinity for each domain and conformation(s) of the tandem domains that would favor in our context (Xph20+linker+binding motif) interactions with domain 2. Linker length is not a parameter that was investigated in this study as the initial ones that we designed performed well (the ETWV and Stg linkers are of different length, but not significant difference was observed), nonetheless our results suggest that beyond the overall binding efficiency they may contribute to the domain targeting.*

5. Is there a cellular phenotype when the interaction of PSD95 is blocked that can be linked to the role of PSD95 in cells?

As no comparable molecular tool exists and genetic approaches suffer from compensation mechanisms, no phenotype has been reported yet for the blocking of PSD-95 PDZ domain 2. This is currently being investigated.

In general, the approach is a nice idea however I believe that it hasn't met the high standards of Nature Communication and should be published in a more specialized journal.

Reviewer #3 (Remarks to the Author):

The manuscript by Rimbault et al represents a huge piece of work. It includes a very thoughtful planning, several demanding experimental steps very carefully executed and analyzed, and rock-solid results that appear to fulfill all aspects of the planned aims. A very wide panel of techniques have been used, justifying the important list of authors that contributed different expertises. Finally, the paper is extremely well written; the reader is very well guided through the impressive amount of data.

The initial question was, how can we make selective inhibitors of close paralog proteins comprising very similar, yet not identical, peptide-binding domains? The "paralog model" is here the DLG protein family. Four very similar proteins, containing three PDZ domains, among which two are closely associated within a tandem unit. All these PDZ domains are known to bind C-terminal PDZ-Binding Motifs ("PBMs") with overlapping binding specificities. How to target one paralog protein and not the three other ones? The proposed strategy has been to build a library with a scaffold domain called "10FN3" bearing engineered variable loops. This generated a library of "10FN3bodies" which were then screened by Phage Display to identify specific binders of one particular paralog (here PSD-95). Several clones were selected, that bound selectively to PSD-95 in the submicromolar-nanomolar range while binding only weakly, if at all, to the other paralogs. The authors then built protein fusion constructs combining one of the selected binders to PBM peptides able to bind to one of the PDZ domains of PSD-95. This resulted in a bivalent protein inhibitor of PSD-95 that binds in the subnanomolar - picomolar range. The authors then developed an impressive series of approaches to characterize the thermodynamic, kinetic and structural aspects of binding, including NMR, SPR-BIAcore, ITC, cross-linking, mass spectrometry, semi-experimental

structural modeling and pull-down and cellular assays. Among other findings, they could show that the "10FN3bodies" bound to regions distal from the "PBM-binding grooves of the PDZ domains. They could map these regions quite precisely thanks to NMR experiments.

Comments

1) Nomenclature.

The authors should also mention that these four proteins are also very frequently called DLG-1,2, 3 and 4. Actually, it is somehow confusing that in the manuscript the authors use the same name, i.e. PSD-95, to name both the entire four-member family and the particular member of that family that they focus on. Why not call it the DLG family rather than the PSD-95-like" family? Then the confusion between PSD-95 as a protein or as a family would be avoided and the reader would be less confused with all these names...

We have followed reviewer #3 advice and modified the text accordingly.

2) Main data vs supplemental data.

In principle, the figures or tables in the main text should focus on those results that are most convincing, reliable and essential to the main line of the paper. Supplemental figures should rather deal with additional "details" (or repeats of an observation shown in the main figures). Furthermore, in such a solid experimental paper, any model or hypothesis figure should be rather kept for the discussion at the end of the paper to clearly differentiate the data from hypothesis-models in the information flow.

In this regard the steps of the phage display, a very essential part of this work, might deserve to be further described in the main text, either in a figure panel or a table.

By contrast: being derived from modeling, do the molecular details shown in fig. 4 really bring much to the work, considering their uncertain nature ? To take an instance, there is no doubt from the NMR data that the Phe residue is involved and that it contacts the indicated regions of the 10FN3bodies. However, the exact conformation of the Phe side chain, and its precise position relative to the bound domain, cannot be known with the atomic precision shown on the plots. Resolving experimentally the conformation of the Phe would require structure solving by crystallography or NMR. Resolving experimentally the relative positions of the two proteins in solution would also require such approaches, or (less reliably) SAXS and/or analytical ultracentrifugation. In the absence of such data, the surface imprints on the DLG tandem shown in Fig 3 are sufficient to describe all the information provided by the NMR data. This figure actually lacks the complementary plots, which would show the surface imprints of the 10FN3bodies (currently shown in suppl fig 17 and 18). Conversely, the plots shown in Fig 4 would be more appropriate in the supplementals.

It is not only that figure 4, from the "experimentally supported reality" viewpoint, may represent the weakest result of this otherwise extremely careful experimental article. It is also that it does not bring crucial information to the story. The critical information is that the phe is strongly involved. The conformation of the Phe in the complex is not a critical information. Same remark for the relative spatial positions of the two protein constructs: what matters for this story is their interfaces, not their precise relative positions.

Therefore my proposed option for these models would either to show them in the supplemental figures, or to use them in a conclusive figure supporting the discussion.

Our primary goal with the models is to provide a basis by which to judge how the binders will associate with the tandem PDZ1-PDZ2 domains from PSD-95. In addition to defining binding

regions on each component, the models help to identify any potential steric concerns that could hinder the second step in our approach (i.e. the addition of appended PDZ peptide ligands), and to allow for distance measurement between the association interface and the canonical PDZ ligand binding site to ensure sufficient number of residues that link the appended peptide to the Xph binders. To emphasize these aspects of the models – and to avoid over-interpretation of any high-resolution details – we have provided a new main text Fig. 4 that better illustrates these models. What we now highlight is that both Xph15 and Xph20 bind in similar ways to only PDZ1, leaving PDZ2 free to move, and to show that the PDZ peptide binding regions are fully accessible to ligands. For Xph18, we highlight that the PDZ1 and PDZ2 domains are now locked into a complex in which PDZ1 and PDZ2 have very limited mobility. This situation is supported by the fact that Xph18 binds to regions on both PDZ1 and PDZ2, and importantly that a single main series of residual dipolar coupling data defines the interdomain orientation between PDZ1 and PDZ2. As with Xph15 and Xph20, the Xph18-bound complex also does not inhibit access to the canonical peptide binding regions on PDZ1 or PDZ2.

3) Bifunctional inhibitors - avidity & affinity - Ceff.

Combining two low affinity binders that interact with distinct regions of the target protein to build up an inhibitor displaying higher affinity and specificity has been used in diverse occasions in the PDZ field. The authors could cite more papers that used this strategy in the PDZ field. See for instance papers with the following PMIDS:

26014966 ; 25797137 ; 22343531.

These articles also cite, in turn, further examples of multivalent ligands displaying increased affinity and specificity.

In one of these papers (ref 26014966) a very simple mathematical formula was used to compute and discuss the "effective concentration" "Ceff" created by tethering the two sites A and B. The higher this "effective concentration", the more efficient the avidity effect created by the combination of the two binding sites on the single inhibitory molecule. Whenever one creates a bifunctional molecule (here, a 10FN3-PBM fusion) the Ceff is a useful quantitative parameter to describe the efficiency of the "tether" (here, the linker sequence) connecting the two functional moieties. It would be interesting if the authors would apply here the same formula to their own data, to compute for each Xph-PBM fusion the Ceff parameter and discuss it.

We have inserted the reference for targeting HPV E6, divalent ligands including those from Stromgaard group have been cited earlier.

We have done the required measurements (SI Fig 28c and d) to be able to calculate the effective concentration (Ceff, SI table 8).

4) Dual conformations of 10FN3bodies

Page 9

"During the analysis, we also noted that there were two populations in the NMR spectra, but only for the unbound state."

That is quite interesting. Indeed the text is not 100% clear whether this is observed equally for the three constructs. Can the authors discuss this point, propose possible theoretical explanations and also the possible impact of this behavior on the the binding mode ? Indeed, if there are distinct conformations in the unbound state, the binding model may be complex. The derived values of KD values and kinetic constants are more likely to be incorrect. The best way to keep cautious is to describe the constants as "tentative values assuming a simple binding model". Then the reader is warned that these values are to be taken with caution.

We were indeed surprised by the degree to which the Xph molecules displayed dual conformations. In addition to the population details provided within Supplementary Figures 17-20, we now include an additional Supplementary Figure 21 that illustrates the extra data that we have used to characterize these two populations. We know that changes in temperature from 283 K to 308 K have no effect on the balance between these two populations (Supplementary Fig. 21a). We also know that the ratio between the two populations are maintained during titration with PSD-95-12 (Supplementary Fig. 21b,c). This result shows that the populations are maintained in equilibrium and it is not clear if one particular population could bind preferentially to PSD-95-12. For the analysis of chemical shifts between the two populations, it is possible only for Xph20, since the two populations differ by a ratio of ~3:1 and thus their assignment is unambiguous. The secondary chemical shifts for ¹H α or ¹³C α shows that the main difference between the conformations of Xph20 are in beta-strand G and the preceding FG loop (Supplementary Fig. 21d). Finally, we have found that the interconversion between the two conformations is not overly rapid, with no observation of exchange at 1 or 2 seconds by using 2D ¹⁵N-HSQC-based exchange-resolved NMR spectra (Supplementary Fig. 21e). As for an impact on binding mode, it appears that the two populations are thus in equilibrium that is unaffected by temperature, and although requiring more than a second to exchange, only a single bound conformation is observed and both populations decrease equally during binding within the timescale of NMR spectroscopy experiments. Therefore we have added more details about these two populations within the text. We also explicitly state that we use a 1:1 model as NMR titrations all indicate a 1:1 interaction and that the use of another model (with more variables) will always lead to much better fit of any dataset but would be meaningless if we cannot demonstrate its existence by another experimental approach.

5) SPR sensograms profiles

Fig 5. Maybe the authors should comment more extensively and precisely the differences of shape of the different sensorgrams. These shape differences are very striking visually, so any reader will spot them. Explaining and interpreting these shape differences in more detail would be useful (and pedagogic) for readers not experienced with SPR, and will probably also satisfy the experienced users...

To clarify this point we have added a Rate Plane with Isoaffinity Diagonals (RaPID) plot, SI Fig 24, and commented on the impact of different binding rate constant values on the shape of sensorgrams in the legend.

6) "Natural" range of protein-protein binding affinities

Page 10

"Affinities were overall on the lower end of natural binders such as antibodies with dissociation constants ranging from 10 μ M to 100 nM and relatively fast dissociation rate constants (half-lives shorter than 1 min)".

Are antibodies the best instance of "natural binders" ?... Antibodies are binders of a very specific nature and tend to bind in the subnano-picomolar range: they belong to "another planet" as compared to the typical biological binders, as most biological interactions are generally reversible... Indeed, domain-motif interactions (such as PDZ-PBM interaction) are typically in the KD range mentioned by the authors (from 10 μ M to 100 nM with fast dissociation rate). Therefore it rather seems that "10FN3bodies" are already on the "higher end" of natural binders, if one excludes antibodies.

"Xph20 presented a faster k_a and a slower k_d , leading to nanomolar dissociation constants of 330 nM and 67 nM by SPR and ITC respectively."

330nM is rather "submicromolar" (a bit lower than 1 μ M) than nanomolar (around 1-10nM). 67 nM is intermediate. By contrast, later in the manuscript, the affinities of the Xph-ETWV fusions are described as "subnanomolar" when some of them almost approach the picomolar range. The fact, that adding appropriate PBMs to the tail of the Xph binder allows to multiply by almost 10.000 fold the affinity is impressive. An additional reason to calculate the "Ceff" for each bivalent construct as proposed before.

We have followed the reviewer's advices and modified the text accordingly.

7) Full amino acid sequence details of main constructs

Since this article is mainly describing the interaction between protein constructs (DLG fragments and Xph derivatives) it would be very useful if the authors would provide an additional suppl table displaying the full aminoacid sequences of all DLG constructs, Xph (at least 15, 18 and 20) and Xph-PBM fusions. At the moment, to get these sequences it is necessary to look for them in different places, or sometimes try to reconstruct them from fragmental information in different parts of the ms.

In particular, concerning the two "ultimate inhibitors" Xph20-ETWV and Xph20-stg, it would be fantastic to know exactly their sequences. Luckily, I did find the sequence of Xph20-ETWV (suppl figure 27). By contrast I searched everywhere for the exact amino-acid sequence of the Xph20-stg fusion and could not find it. To understand how this construct was built I had to go to the following paragraph (mat&meth p 21-22):

"The fragment corresponding to a 26 amino acid-linker followed by Stargazin 13 last amino acids was synthesized (Genscript) and inserted into pIG-Xph15 after Xph15 sequence with a classical ligation using the XhoI restriction site. This plasmid was used to insert the other Xph sequences between the BamHI and XhoI sites. The constructs were then PCR amplified and transferred into pBIG using the BamHI and BlnI sites. For the amber suppression approach, the TAG amber codon was introduced in the competitors by site-directed mutagenesis at the -5 position. An arginine residue was also introduced by site-directed mutagenesis at -11 and -12 positions for Xph20-ETWV and Xph20-Stg constructs respectively, in order to facilitate proteomics experiments (smaller resulting trypsin-digested fragments)."

Sure, one can try to reconstitute the construct using these explanations, provided that one can find the right stargazin sequence on the web, make the proper mutation at the -11 and -12 positions, reconstitute properly the sequence of the linker etc... But this is really complex when instead the authors can provide the sequence of Xph20-Stg in a common table, together with that of Xph20-ETWV, those of Xph15, 18, 20, and those of the main DLG constructs.

We have added SI table 10 with the requested sequences.

8) Discussion.

-Considering the huge team effort and human and material resources that have obviously been invested in this work, it might be worth discussing at least in a small paragraph the possible strategies that now may be envisioned to allow the in vivo targetting of such inhibitor in the animal's or person's body at the right localization where it should perform its inhibitory action. Indeed, in our own team we have previously generated, a recombinant bifunctional fusion protein that acted as a potent inhibitor of a viral oncoprotein. However, when trying to transfer our finding, it appeared that the targeting of a recombinant protein inhibitor in an intracellular

context remains very difficult to develop. Maybe the situation is evolving and the authors are aware of innovative strategies ? This would be interesting to discuss.

We have modified a paragraph (p20) in the discussion where we discuss possible ways of delivering the competitors. We have also produced and used on culture neurons a fusion of the competitor with a cell permeable peptide (TAT) on p13 and fig 7. The primary application of the tools is to address fundamental questions with respect to the molecular mechanisms involved in the synapse function. In that context, the delivery approaches and requirements are slightly different from what may be needed when considering therapeutic applications. In particular, local application or sparse transfection or infection in culture or tissue models may be preferable over high yield widespread delivery in more integrated biological models.

-Discussion page 15. "Although epitopes were only mapped for three of the clones, it seems likely that given their specificity most bind to regions outside of the conserved binding grooves.". This is a very interesting remark. It would be tempting to speculate, that the fact that PBM-binding groove is not a good "attractor" for "10FN3bodies" is related to the fact that the PBM-binding groove does not bind strongly and specifically to PBMs either. In other words, the stereochemical properties of the PBM binding groove would not make it a candidate for strong and/or specific protein-protein interactions, in general. And, by the way, this weak capacity for high-affinity and/or high binding specificity might well result from evolution and thereby respond to a particular requirement in the cell. It is well known that PDZ-PBM interactions are rather weak and promiscuous.

The comment is interesting on many levels but it is difficult to integrate more elements in the discussion on this remark as we have only identified the epitopes for 3 clones. Indeed, while we can justify our claim that most isolated clones likely bind outside the binding groove, we do lack additional elements (in particular structural and binding data) that would allow us to compare ¹⁰FN3 clones that bind outside of the binding groove (which we estimate to be the majority of the ones we isolated), clones that rely on the PDZ domain binding groove for interaction and PBMs. The fact that we did not apparently isolate such clones in our selection does not necessarily means that strong or comparable binders cannot be generated. Considering that specificity was the prime motivation of our selection and that the binding grooves are conserved, we did not investigate clones that clearly lacked specificity (Xph19 or 24 for instance) or orient the selection for PBM-binding groove binders. We currently interpret the results of our selection more as the consequence of the existence of a strong hot-spot on the "back" of PDZ domain 1. We anticipate that clones that would rely on the binding groove would engage one of their diversified loops in parts or totality of the PDZ domain binding groove (which would result in low micromolar interactions with fast kinetics as would be expected from a PBM) and could use the rest of their surface, and in particular the second diversified loop, to generate additional interactions that would stabilize the overall interaction in comparison to canonical PBM. Consequently, the binding modes of a canonical PBM and one resulting from a small domain on which parts of the surface were diversified are likely to significantly differ both in terms of affinity and binding kinetics. However, in the absence of experimental evidence that remains highly speculative.

9) Detected typos (very few)

-Page 3

"One of its main functionS" *Corrected*

-page 15

"one of the main synaptic scaffold proteinS." *Corrected*

-Page 17 "at the price of an acceptable loss of the competitor affinity" -> "at the expense of an acceptable loss of the competitor affinity" *Corrected*

Reviewer #4 (Remarks to the Author):

The manuscript of Rimbault and colleagues reports on the design of specific ligands to specifically target a PDZ domain tandem of PSD95. For this purpose, the authors combined two molecules within a single bivalent chimera: one has been selected by phage display using library built with the so-called 10FN3 scaffold domain; the second molecule is a peptide containing a PDZ-binding motif capable to bind to one of the PDZ domains of PSD95.

The authors accumulated a huge amount of data using highly diverse approaches (phage display, NMR, SPR, ITC, MS, pull-down, ...) to characterize the binding of their molecules.

The manuscript shows a very beautiful and original work, is highly pleasant to read and the supplementary file section, although very large (it contains 32 suppl. figures, 6 tables and 2 notes) is highly useful for readers who are interested to go deeper in the details. Results are very interesting, not only in the PDZ topic, but also more generally in inter-domain interaction research area.

I strongly support the publication of this manuscript if the authors can address the four Main aspects detailed below.

M1/ Specificity vs selectivity.

These two words are often employed in the manuscript. But it would be a plus whether the authors would define the meaning of these two words in the manuscript, specially the difference between the two. Indeed, there are several ways to define specificity or selectivity depending on the scientific context (chemistry, biology, by comparing to a few members of a family, or to the full list of the family members, ...).

We have added a note in the SI to clarify the definition of the two terms.

M2/ Page 7: "Based on the specificity evaluation, none of the final clones clearly stood out and therefore we selected three representative clones (Xph15, Xph18 and Xph20)".

I do not fully agree with that sentence and the reasoning proposed by the authors to justify the choice of Xph20. To me, several clones clearly stand out based on the Phage-ELISA (2.c) and pull-down assays (2.d): Xph15, Xph18, Xph21 and Xph25. Indeed, more than the high intensity of the ratios calculated from the pull-down assay, I consider the absence of signal for the 3 other PDZ tandems (dark blue) compared to PSD95-12 as more important and relevant of the specificity; this way of thinking would exclude the Xph20 clone.

Furthermore, by looking at the 1H-15N HSQC (Supp. Fig. 7), I also have some concerns regarding the Xph20 clone since the superimposition of the HSQC spectra for SAP-97 shows quite significant changes as it is for PSD-95 (in both cases, blue and green peaks become visible). This observation calls into question the specificity of this clone.

Therefore, can the authors explain why they select Xph20, rather than Xph21 or ph25?

We have modified the text to clarify the reason of our choice. Xph21 was excluded as its binding strength appears lower than most clones, Xph25 performed similarly as Xph15, and Xph20 was chosen as despite a slight sign of binding to another tandem (only in the pull-down assay) it constituted one of the best binders in terms of relative "affinity". Xph18 was chosen as an intermediate clone between 15 and 20.

In the NMR experiments (SI Fig 8) and conditions (> 80 μ M), some level of interaction between SAP97 and Xph20 can be indeed observed. We have added a SPR experiment (SI 23b) to estimate the binding constant and report a $K_D > 150 \mu$ M. These results were commented in the main text.

M3/ SPR concerns

Usually SPR people decide to use single cycle experiments when they face some troubles to regenerate their surface, or when regeneration is detrimental to the ligand.

Regeneration might be an issue specially when the dissociation rate constant is very slow. However, this doesn't seem to be the case in the present work according to the KD values that the authors have reported at least at the beginning of the manuscript.

We routinely use capture systems to avoid having to both deal with regenerations, which in some cases can compromise the ligand's activity, or loss of activity of the ligand (anchored to the chip). The capture system allows to renew the ligand after each cycle.

All together, these information would suggest that the authors faced some difficulties to develop this assay. In this regard, the authors should check several issues and discuss them in the manuscript:

* in figure 5a, it is striking to me that the overall shapes for Xph15 and Xph18 recorded by single cycle SPR experiment are so different when the affinities are so similar (4.3 uM compared to 2.6 uM). What does it mean?

The shapes of the sensorgrams are different because the association and dissociation rate constants are different for the various Xph. As the dissociation constant is a ratio of the two rate constants, different sets can lead to similar values. To clarify this point we have added a Rate Plane with Isoaffinity Diagonals (RaPID) plot, SI Fig 24.

* in the same figure 5a, the experimental plateau observed for the highest analyte concentration is significantly different from sample to sample (from ~10 RU for Xph15 to about 35 for Xph18), although the highest concentrations for Xph15 and Xph18 are the same and the affinities for Xph15 and Xph18 differ only by a factor of 2 (4.3 uM compared to 2.6 uM). Is it possible for the authors to provide some explanations?

The density of Xph15 and Xph18 used for the experiments was slightly different (captured Xph15, 18 and 20 about 37, 74 and 24 RU respectively) and led to the difference in response intensity. We have specified these densities in the legend.

* Rmax describes the maximum binding capacity, and can be calculated knowing the level of immobilized ligand at the surface as well as the molecular weights of analyte and ligand. It would be good to compare the theoretical maximal value with the experimental one obtained during the runs.

We have specified the amount of immobilized ligand in all experiments where that variable was key.

* Is there any explanation for the biphasic behavior observed when the ligand is one of the PDZ tandem (Supp. fig. 22), whereas a mono-exponential behavior is observed for Xph clones as a ligand (Fig. 5)?

The common explanations for a different behavior when the analyte and ligand are interchanged is either heterogeneity in one of the partner or surface-related effects. Heterogeneity implies that there is an additional specie in one of the analyte solution that leads to another type of binding event. This binding event may occur either on the ligand or on the chip surface. If the dissociation rate constant of this event is slow, this will lead to dramatic effects on the sensorgram curve, even more so if this occurs on the chip surface. The additional species may be a different conformer or partially unfolded analyte, an oligomer or aggregate, a truncated analyte or a contaminant. From all of our analysis and various experiments, we can rule out oligomer or aggregates. The dual conformation of the three Xph observed by NMR may be at the origin of this effect.

M4/Kinetic and steady-state analysis

In link with the previous remarks, another point would deserve to be addressed: as said before, the authors observed a mono-exponential or a biphasic behavior depending on the choice of the ligand for SPR experiment. They also observed by NMR that free Xph20 and Xph15 displayed two populations (not equally distributed between the two samples), likely arising from two slowly exchanging populations.

Despite this, they used 1:1 kinetics or 1:1 steady-state model to analyze their data, more likely because more complicated models would fit the data anyway.

Taking all these observations together, I would rather temper the analysis and the conclusions regarding numerical values of binding kinetic and affinity constants since the fitted constants are probably biased in some way.

This point should at least be discussed in the manuscript.

In response to similar comments by other reviewers, we have added sentences in the discussion. With respect to the binding model choice, we use a 1:1 model as NMR titrations all indicate a 1:1 interaction and as the reviewer points out the use of another model (with more variables) will always lead to much better fit of any dataset but would be meaningless if we cannot demonstrate its existence by another experimental approach.

Next, some other Points are given below that can be easily addressed by the authors:

P1/ Sup. fig. 10:

According to the legend, crosspeaks that shift upon addition of Xph15 are annotated. However, I have the feeling that several annotations are missing (for instance the peaks at 10.3;123 ppm or 7.8;111 ppm).

We have now included additional crosspeak annotation to Supplementary Figure 10.

P2/ An other way to represent the shift upon binding would be to plot the $\Delta\delta(H,N)$ vs. the sequence (as actually the authors did in suppl. fig. 14 to 20). It would seem appropriate to systematically provide this kind of plot for every figure displaying superimposed NMR data.

The three plots for the PSD-95-12 chemical shift perturbation ($\Delta\delta(H,N)$ vs. the PSD-95-12 sequence) are already provided for Xph15, Xph18 and Xph20 within the main text Fig. 3a,b,c. We have therefore not repeated them in the Supplementary Figures

P3/By the way, it would be valuable, in order to help the reader, to superimpose on these $\Delta\delta(H,N)$ plots not only the position of the BC, DE and FG loops, but more generally information about secondary structures elements (as strands beta A to D).

We agree that this would be a useful addition, and we have now updated Supplementary Figures 17,18,19,20,21 with explicit annotation of the beta strands.

P4/ PSD95-12 and addition of the different Xph by NMR (Suppl. figures S10-S12): why a surface representation to map the residues affected upon binding is not always provided as it is in Suppl. figures S14-S20?

The three plots for the PSD-95-12 chemical shift perturbation ($\Delta\delta(H,N)$ vs. the PSD-95-12 sequence) are already provided for Xph15, Xph18 and Xph20 within the main text Fig. 3a,b,c. We have therefore not repeated them in the Supplementary Figures.

P5/ Fig. 5: why aren't enthalpy changes (ΔH) and entropy changes ($-T\Delta S$) provided as usually done for ITC data?

We have added them in Fig 5.

Finally, a few Remarks on the text:

R1/ In the M&M section for SPR, a reference is missing for the sentence " biotinylated peptide was synthesized as previously described ".

Meanwhile, there is no information about what is attached to the reference flow cell.

We have added the reference and specified the state of the reference flow cell.

R2/ To avoid any confusion between affinity (KD) and kinetic rates (kd), I would suggest to use k_{on} and k_{off} rather than k_a and k_d for kinetics association and dissociation rate constants, respectively.

As suggested, we have replaced k_a and k_d by k_{on} and k_{off} .

R3/ Similarly to Xphxx-ETWV constructs, I guess that Xphxx_Stg corresponds to a construct containing the Xph clone followed by a linker and the 13 last residues of stargazin. However, unless I missed it, there is no clear definition of Xph-Stg in the text.

We have added the sequences of the main constructs in SI table 10.

R4/ Supp. fig. 26, panel b: in order to improve the visibility of the plot showing the equilibrium analysis, I would suggest to use a log scale on X axis rather than the actual linear scale. Differences between the curves will be easier to observe.

We have repeated the titration (now SI28) and extended the concentration range.

A few Typos:

T1/ Figure 7, legend of panel d/: "minimum and maximum OF all individual data points".
Corrected

T2/ Page 11: "some level(S?) of interaction is(ARE?) here observed ..."
Corrected

T3/ End of page 14: "In contrast, expression of Xph20-ETWV lead(S) to more than a 50% decrease ..."
Corrected

T4/ Page 17: "The fusion of moderate to weak binders (100 nM-10 uM) thus resulted in strong binders with affinities in the pM range arising from slower dissociation rate constants, Kd." Is it not kd rather than Kd? And therefore koff according to one of my previous comments.
Corrected

T5/ Page 23: "Cultures were grown in sterile glass vessel(S) 5 h at 37 °C at 250 rpm(.)" The period is missing.
Corrected

Reviewers' comments:

Reviewer #1 (Remarks to the Author):

The revised manuscript by Rimbault et al. includes new experiments, new analysis and added clarification regarding the specificity and affinity of their designed inhibitors. It is now made clear that their inhibitor is selective for PSD-95 within the DLG family. The new fluorescence polarization measurements have better established the binding constants. The Response to Reviewers was well crafted and satisfactorily answered my questions regarding the modeling and structure of the PDZ tandem and the experimental details of SPR. I also thank the authors for helping me to locate specific pieces of data that I was having trouble finding in this very data-dense manuscript. This study has described a novel way to target one domain in a multidomain protein and use that to leverage specificity within a highly homologous protein family. All my concerns have been address and I support publication of the revised manuscript.

Reviewer #2 (Remarks to the Author):

The authors have submitted a revised version of their manuscript: "Engineering selective competitors for the discrimination of highly conserved protein-protein interaction modules". Although some of my original criticism has been addressed, I am still not convinced that the paper represents a clear step forward. While the approach is innovative, I still think that there some issues that cannot be ignored:

1. The issue of stoichiometry has not been adequately addressed. The ITC experiments in Figure 5 show that the binding does not correspond to a 1:1 stoichiometry ($N=1$). This should be discussed in greater detail in the manuscript. For example even the strongest binder Xph20 does not really show a 1:1 binding. The argument that conformational changes are required to establish binding is weak because the NMR chemical shift perturbation analysis shows a relative narrow epitope on the 1st PDZ domain for Xph15 and Xph20. Also, it should be critically discussed how the stoichiometry effects the models shown in Figure 4.

2. The problem with specificity is not addressed adequately. The pull-down experiments shown in Figure 7 are not sufficient to establish intracellular specificity. They only establish that (in case the authors have validated that the band is indeed PSD95) PSD95 is bound. Other proteins with a similar PDZ binding motif may also be Co-Iped here. For example, all class I PDZs have to some extent the

ability to bind the peptide ETWV (Tonikian 2008 Plos Bio). Therefore, I suggest to perform a Co-IP Massspec experiment where the engineered bi-valent binder is expressed in cells fused to a Flag-Tag and after Co-IP the resulting supernatant is analyzed using mass spectrometry.

The data show in Figure 6 c would be IC50 and should not be confused with an inhibitor constant K_i . Also densitometry is less than ideal to establish an IC50. I would therefore suggest some form of competition ELISA experiment.

Furthermore, the authors use quite frequently selective binding or selective blocking although it is not clearly established if the binders are indeed selective / specific. This should be phrased more carefully.

Reviewer #3 (Remarks to the Author):

The authors have addressed all my points therefore I advise to publish the manuscript. I have no additional comment

Reviewer #4 (Remarks to the Author):

All the points raised in the first submission have been satisfactorily addressed.

Reviewers' comments:

Reviewer #1 (Remarks to the Author):

The revised manuscript by Rimbault et al. includes new experiments, new analysis and added clarification regarding the specificity and affinity of their designed inhibitors. It is now made clear that their inhibitor is selective for PSD-95 within the DLG family. The new fluorescence polarization measurements have better established the binding constants. The Response to Reviewers was well crafted and satisfactorily answered my questions regarding the modeling and structure of the PDZ tandem and the experimental details of SPR. I also thank the authors for helping me to locate specific pieces of data that I was having trouble finding in this very data-dense manuscript. This study has described a novel way to target one domain in a multidomain protein and use that to leverage specificity within a highly homologous protein family. All my concerns have been address and I support publication of the revised manuscript.

We thank the reviewers for their constructive comments and help in improving the manuscript.

Reviewer #2 (Remarks to the Author):

The authors have submitted a revised version of their manuscript: "Engineering selective competitors for the discrimination of highly conserved protein-protein interaction modules". Although some of my original criticism has been addressed, I am still not convinced that the paper represents a clear step forward. While the approach is innovative, I still think that there some issues that cannot be ignored:

1. The issue of stoichiometry has not been adequately addressed. The ITC experiments in Figure 5 show that the binding does not correspond to a 1:1 stoichiometry ($N=1$). This should be discussed in greater detail in the manuscript. For example even the strongest binder Xph20 does not really show a 1:1 binding. The argument that conformational changes are required to establish binding is weak because the NMR chemical shift perturbation analysis shows a relative narrow epitope on the 1st PDZ domain for Xph15 and Xph20. Also, it should be critically discussed how the stoichiometry effects the models shown in Figure 4.

The ITC experiments were performed to confirm the affinity results obtained by SPR, which they do by providing similar binding constants for Xph15, 18 and 20. In this revised version, we have repeated the ITC titration for Xph18 (Fig 5b) and consequently obtained an increased N with a similar K_D value (as the experiment was performed with a different Xph18 construct we have not averaged the values with the previous ones). We have also added new comments in the Fig 3 legend and in the ITC method section regarding stoichiometry.

We agree that the results obtained by ITC for Xph15, 18 and 20 do not correspond to a strict 1:1 stoichiometry and note that such deviations are not uncommon when investigating interactions between two proteins. Determining whether these deviations actually reflect a different stoichiometry necessarily requires supporting experimental evidence.

The deviation in the obtained values may in part be due to any difference in the measured concentrations and the actual active concentrations of any two protein partners. In that regard, we note that Xph18, which is the one that deviates the most from the 1:1 stoichiometry in the ITC experiments, has consistently been more complicated than the other clones to obtain at the high concentrations (50-100 μ M or higher) required for implementing the titrations. It could also be due to a different model than the simple 1:1 interaction (but still with a 1:1 stoichiometry). This more complex model would be consistent with the fact that Xph18 shows binding only to the isolated PDZ1 domain by NMR spectroscopy and not to PDZ2, and thus

the secondary contact with PDZ2 likely occurs either by conformational selection or induced fit of the PDZ domain arrangement in the tandem construct. We have also discussed this general possibility in the text in light of the differences observed in SPR signal depending on the assay orientation (which is not an uncommon situation) and on the observed dual conformation of the clones by NMR. We agree that such findings could in theory also be the consequence of different stoichiometry. However, we have no evidence for a 2:1 or a 1:2 model in any of the complement approaches, and on the contrary have two other experimental approaches that are consistent with a 1:1 model (SPR and NMR). In particular, for the three clones, titrations followed by NMR spectroscopy do not show modification or appearance of new signals after 1:1 stoichiometry is reached when titrating in the Xph and the reverse titration leads in each case also leads to a single conformer and population of Xph15, 18 or 20. In the preparation of every complex for multidimensional NMR spectra, a slight molar excess of 1.2 was enough to fully saturate a single bound conformation. Additionally, the compactness of the epitopes for Xph15 and 20 reasonably excludes other models with respect to stoichiometry. As previously mentioned, the situation is more complex for Xph18 as the epitope is more spread out. It has been one of our major efforts in the revision process to provide a model for Xph18 that satisfies all our experimental data. While a 2:1 binding model with two Xph18 for one tandem PDZ domain could explain this distributed epitope, it would not comply with SPR, NMR titrations and the RDC experiments. The latter RDC results, in particular, show a constrained arrangement of the two PDZ domains to each other in complex with 1.2 molar equivalents of Xph18, supporting a single binding event. The same arguments as outlined above would apply to the opposite 1:2 model.

As described in the text, the models shown on Fig 4 and SI Fig 22 were generated using a 1:1 model of interaction.

Depending on what the reviewer is referring to with conformational change, we note that:

-the argument that conformational changes of PSD-95 may be required for binding was only used for Xph18 and is suggested by markedly slower K_{on} by SPR and the size of the epitope, which in a 1:1 model would impose a constrained conformation of the tandem PDZ domain.

-Unbound Xph15, 18 and 20 all display two main conformations and therefore conformational changes are required for binding of a subpopulation of Xph15, 18 and 20 for which a single conformation is observed when bound.

2. The problem with specificity is not addressed adequately. The pull-down experiments shown in Figure 7 are not sufficient to establish intracellular specificity. They only establish that (in case the authors have validated that the band is indeed PSD95) PSD95 is bound. Other proteins with a similar PDZ binding motif may also be Co-IPed here. For example, all class I PDZs have to some extent the ability to bind the peptide ETWV (Tonikian 2008 Plos Bio). Therefore, I suggest to perform a Co-IP Massspec experiment where the engineered bi-valent binder is expressed in cells fused to a Flag-Tag and after Co-IP the resulting supernatant is analyzed using mass spectrometry.

In the manuscript, we have characterized the newly isolated clones (Xph15, 18 and 20), which according to our data bind exclusively and specifically to PSD-95 in a large concentration range, as well as the engineered competitors and in particular Xph20-ETWV. The competitors were all designed with the same goal to readdress a non-selective canonical binding motif (either derived from stargazing or ETWV) by the fusion to a specific binder. We are therefore fully aware of the intrinsic capacity of each such competitor to bind to any class I PDZ domain given the presence of a valine and a threonine at the 0 and -2 positions respectively. Our characterization effort has consequently been focused on determining how efficient was the readdressing by systematically comparing the binding and competing properties between PSD-95 and the most relevant potential off-targets (PSD-93, SAP97 and SAP102). Our data is consistent with what could be expected from the design, i.e., while all the competitors possess a basal affinity for all type I PDZ domain based on the nature of the competing moiety,

they display a significantly improved affinity and competing capacity for PSD-95, which we refer to as selectivity.

In addition to the previous characterizations, we have followed the reviewer request and performed a proteomic analysis of the cellular targets of Xph20-ETWV. We have therefore added in the revised manuscript LC-MS/MS analyses of pull-downs conducted with biotinylated Xph20-ETWV and Xph0-ETWV on rat brain lysates (Fig 7c and d). The results now clearly demonstrate that the fusion of Xph20 and the ETWV motif allow selective binding and enrichment of PSD-95 over other PDZ domain-containing proteins that can interact with the ETWV motif.

The data show in Figure 6 c would be IC50 and should not be confused with an inhibitor constant Ki. Also densitometry is less than ideal to establish an IC50. I would therefore suggest some form of competition ELISA experiment.

The data shown in Fig6c actually correspond to binding constants and not concentrations. The "One site – Fit Ki" model that we used provided with the GraphPad software is the following:

$$\log EC50 = \log(10^{\log Ki} * (1 + Lig/LigKd))$$
$$Y = Bottom + (Top - Bottom) / (1 + 10^{-(X - \log EC50)})$$

where,

Lig is the concentration of labeled ligand in nM. A single concentration of ligand is used for the entire experiment (constrained value)

LigKd is the equilibrium dissociation constant of the ligand in nM (constrained value)

Top and Bottom are plateaus in the units of Y axis.

logKi is the log of the molar equilibrium dissociation constant of competitor.

Ki is the equilibrium dissociation constant in Molar.

Our understanding of the reviewer comment is that densitometry performed on colloidal stained gels may be an issue for its sensitivity compared to enzyme-based measurements for lower concentrations.

When setting up the assay, western blotting was also considered as a method to measure binding. Considering that western blotting and colloidal blue staining provided similar results, we opted for the latter, which presents the advantages of allowing to see all protein material (and confirm that equal amounts of beads were used) and of being faster to implement.

Considering our data, using a more sensitive detection technique might change the final Ki values but it would do it uniformly on all sample and not change the fold difference between the different conditions, which in our opinion is the key information to be extracted from that experiment.

ELISA-based competition experiments are certainly a possible alternative format for our competition assay but we are not convinced that setting up a new assay format and repeating the whole experiment is justified considering that the conclusions would be the same even if the absolute Ki values were modified.

Furthermore, the authors use quite frequently selective binding or selective blocking although it is not clearly established if the binders are indeed selective / specific. This should be phrased more carefully.

We have been careful in trying to clarify our use of "selective" and "specific" and have rewrote the supplementary note on that point (supplementary note 1) to make it as clear as possible.

We feel that we have provided numerous and sufficient experimental elements to support the claims that

-Xph15,18 and 20 are specific for PSD-95 vs paralogous proteins (SAP97, SAP102 and PSD-93) in the sense that over a broad range of concentrations only binding to PSD-95 is observed.

-Xph20-ETWV (and other similar ligands to the extent that we have characterized them) is selective for PSD-95 vs paralogous proteins (SAP97, SAP102 and PSD-93) in the sense that

it presents a significantly increased affinity for PSD-95 compared to the other proteins. We can reasonably extend that property to other type-I PDZ domain-containing proteins as their interaction would solely result for the binding of the ETWV or Stg motif (as is the case for paralogs) and as indicated by the proteomic analysis (Fig 7c and d).

Reviewer #3 (Remarks to the Author):

The authors have addressed all my points therefore I advise to publish the manuscript. I have no additional comment

We thank the reviewers for their constructive comments and help in improving the manuscript.

Reviewer #4 (Remarks to the Author):

All the points raised in the first submission have been satisfactorily addressed.

We thank the reviewers for their constructive comments and help in improving the manuscript.